# Convergence Analysis of Gradient Descent under Coordinate-wise Gradient Dominance

## Abstract

We consider the optimization problem of finding Nash Equilibrium (NE) for a nonconvex function $f(x) = f(x_1, ..., x_n)$, where $x_i \in \mathbb{R}^{d_i}$ denotes the $i$-th block of the variables. Our focus is on investigating first-order gradient-based algorithms and their variations such as the block coordinate descent (BCD) algorithm for tackling this problem. We introduce a set of conditions, termed the $n$-sided PL condition, which extends the well-established gradient dominance condition a.k.a Polyak-Łojasiewicz (PL) condition and the concept of multi-convexity. This condition, satisfied by various classes of non-convex functions, allows us to analyze the convergence of various gradient descent (GD) algorithms. Moreover, our study delves into scenarios where the objective function only has strict saddle points, and normal gradient descent methods fail to converge to NE. In such cases, we propose adapted variants of GD that converge towards NE and analyze their convergence rates.

## 1 Introduction

Optimization problems with nonconvex objectives appear in many applications from computer science to economics (Intriligator, 2002) and more recently, in machine learning (Jain et al., 2017), such as training deep neural networks (Goodfellow et al., 2016) or policy optimization in reinforcement learning (Silver et al., 2014). On the other hand, the Gradient Descent (GD) algorithm and its variants are driving the practical success of many machine learning approaches. Naturally, understanding the limits of such GD-based algorithms in the nonconvex setting has become an important avenue of research in recent years (Jin et al., 2021; Zhou et al., 2024; Jordan et al., 2023). Along this line of research, we are interested in finding Nash Equilibrium $x^\star = (x_1^\star, \cdots, x_n^\star)$ for the nonconvex optimization $f(x)$, i.e.

$$f(x_i^\star; x_{-i}^\star) \leq f(y_i; x_{-i}^\star), \forall y_i \in \mathbb{R}^{d_i}, \tag{1}$$

where $f$ is a continuously differentiable but possibly nonconvex function. The variable $x$ can be partitioned into $n$ blocks $(x_1, ..., x_n)$, where $x_i \in \mathbb{R}^{d_i}$ is the $i$-th block and $\sum_{i=1}^n d_i = d$. This optimization problem can be viewed as a potential game between $n$ players. The objective of $i$-th player is to minimize the function $f(x_i, x_{-i})$ when other players' parameters are denoted by $x_{-i}$.

From a game-theoretic perspective, this is a multi-agent potential game where the potential function $f$ captures the aggregate impact of all agents' strategies $\{x_i\}_{i=1}^n$ Monderer & Shapley (1996). Each agent minimizes $f$ over its variables $x_i$, assuming others' strategies are fixed. However, privacy concerns arise as strategies may reveal sensitive information. In decentralized settings, such as network routing Candogan et al. (2010) or resource allocation (Zhang et al., 2021), agents optimize independently without full knowledge of $f$ or others' strategies. Furthermore, convergence to an NE is not always stable (Carmona, 2013), as gradient descent may diverge.

For a general nonconvex differentiable function $f : \mathbb{R}^d \to \mathbb{R}$, finding its NE is PPAD-complete (Daskalakis et al., 2009). A straightforward approach to tackle this problem is to introduce additional structural assumptions to achieve convergence guarantees. Within this scope, various relaxations of convexity have been proposed, for example, weak strong convexity (Liu et al., 2014), restricted secant inequality (Zhang & Yin, 2013), error bound (Cannelli et al., 2020), quadratic growth (Cui et al., 2017), etc. Recently, there has been a surge of interest in analyzing nonconvex functions with block structure. Multiple assumptions have been analyzed which is correlated to each block when other blocks are fixed, for example, PL-strongly-concave (Guo et al., 2023), nonconvex-PL

(Sanjabi et al., 2018), PL-PL (Daskalakis et al., 2020; Yang et al., 2020; Chen et al., 2022) and multi-convex (Xu & Yin, 2013; Shen et al., 2017; Wang et al., 2019a; 2022b). For instance, the multi-convexity assumes the convexity of the function concerning each block (coordinate) when the remaining blocks are fixed.On the other hand, the other aforementioned conditions are tailored for objective functions comprising only two blocks. They are particularly defined for min-max type optimizations rather than minimization tasks.

The nonconvex optimization realm has seen a growing interest in the gradient dominance condition a.k.a. Polyak-Łojasiewicz (PL) condition. For instance, in analyzing linear quadratic games (Fazel et al., 2018), matrix decomposition (Li et al., 2018), robust phase retrieval (Sun et al., 2018) and training neural networks (Hardt & Ma, 2017; Charles & Papailiopoulos, 2018; Liu et al., 2022). This is due to its ability to enable sharp convergence analysis of both deterministic GD and stochastic GD algorithms while being satisfied by a wide range of nonconvex functions. More formally, a function $f$ satisfies the PL condition if there exists a constant $\mu > 0$ such that

$$\|\nabla f(x)\|^2 \geq 2\mu(f(x) - \min_{y \in \mathbb{R}^d} f(y)), \forall x \in \mathbb{R}^d. \tag{2}$$

This was first introduced by Polyak (1963); Lojasiewicz (1963), who analyzed the convergence of the GD algorithm under the PL condition and showed its linear convergence to the global minimum. This condition can be perceived as a relaxation of strong convexity and as discussed in (Karimi et al., 2016), it is closely related to conditions such as weak-strong convexity(Necoara et al., 2019), restricted secant inequality(Zhang & Yin, 2013) and error bound(Luo & Tseng, 1993).

As mentioned, the PL condition has been extended and applied to optimization problems with multiple coordinates. This extension is analogous to generalizing the concept of convexity (concavity) to convex-concavity. For instance, the two-sided PL condition was introduced in (Yang et al., 2020) for analyzing deterministic and stochastic alternating gradient descent ascent (AGDA) in *min-max games*. It is noteworthy that most literature requires convexity or PL condition to establish the last-iterate convergence rate to the NE (Scutari et al., 2010; Sohrabi & Azgomi, 2020; Jordan et al., 2024). This, however, may not hold even if the objective function is quadratic. A considerable relaxation is that the function satisfies strong convexity or PL condition when all variables except one are fixed. Two natural questions arise:

*Can similar results be achieved by extending the two-sided PL condition to accommodate optimization problems in the form of equation 1, where the objective comprises $n$ coordinates? And is there an algorithm to guarantee convergence at a linear rate in such problems?*

Furthermore, as highlighted by Lee et al. (2016); Panageas & Piliouras (2016); Ahn et al. (2022), GD with random initialization almost surely escapes the NE point when it is a strict saddle point. Also, Xu & Yin (2013; 2017) require the potential function to be lower-bounded to approach the NE set rather than diverge to infinity. These prompt us to consider the following questions:

*Is it possible to ensure the convergence to the NE set even though it only contains strict saddle points or the function is not lower bounded by using first-order GD-based algorithms?*

Motivated by the questions above, we introduce the notion of $n$-sided PL[1] condition (definition 2.6), which is an extension to the PL condition and shows that it holds in several well-known nonconvex problems such as $n$-player linear quadratic game, linear residual network, etc. It is noteworthy that unlike the two-sided PL condition, which guarantees to converge to the unique Nash Equilibrium (NE) in min-max optimization (Yang et al., 2020; Chen et al., 2022), functions satisfying the $n$-sided PL (even 2-side PL) condition may have multiple NE points (see section 2.1 for examples). However, as we will discuss, the set of stationary points for such functions is equivalent to their NE points. Moreover, unlike the two-sided PL condition, which ensures linear convergence of the AGDA algorithm to the NE, the BCD algorithm exhibits varying convergence rates for different functions, all satisfying the $n$-sided PL condition. Similar behavior has been observed with multi-convex functions (Xu & Yin, 2017; Wang et al., 2019a). Therefore, additional local or global conditions are required to characterize the convergence rate under the $n$-sided PL condition.

In this work, we study the convergence of first-order GD-based algorithms such as the BCD, and propose different variants of BCD that are more suitable for the class of nonconvex functions satisfying

---

[1]We should emphasize that 2-sided PL and two-sided PL are slightly different conditions as the former is suitable for $\min_{x,y} f(x, y)$ while the latter is for $\min_x \max_y f(x, y)$.

$n$-sided PL condition. We also introduce additional local conditions under which linear convergence can be guaranteed and the convergence to NE still holds even only strict saddle points exist.

### 1.1 Related Work

**Block Coordinate Descent and its variants.** Block coordinate descent (BCD) is an efficient and reliable gradient-based method for optimization problems in 1 which has been used extensively for optimization problems in machine learning (Nesterov, 2012; Allen-Zhu et al., 2016; Zhang & Brand, 2017; Zeng et al., 2019; Nakamura et al., 2021). Numerous existing works have studied the convergence of BCD and its variants for functions. Most of them require the assumptions of convexity, PL condition, and their extensions (Beck & Tetruashvili, 2013; Hong et al., 2017; Lin et al., 2023; Chen et al., 2023; Chorobura & Necoara, 2023). For instance, Xu & Yin (2013; 2017) studied the convergence of BCD for the regularized block multiconvex optimization. They established the last iterate convergence under Kurdyka-Łojasiewicz which might not hold for many functions globally. The authors in (Lin et al., 2023) considered the generalized Minty variational problem and applied cyclic coordinate dual averaging with extrapolation to find its solution. Their algorithm is independent of the dimension of the number of coordinates. However, their results rely on assuming the monotonicity of the operators, which is often hard to satisfy. Cai et al. (2023) considered composite nonconvex optimization and applied cyclic block coordinate descent with PAGE-type variance reduced method. They proved linear and non-asymptotic convergence when the PL condition holds, which is not valid for functions with multiple local minima.

**PL condition in optimization.** The PL condition was originally proposed to relax the strong convexity in the minimization problem sufficient for achieving the global convergence for first-order methods. For example, Karimi et al. (2016) showed that the standard GD algorithm admits a linear convergence to minimize an $L$-smooth and $\mu$-PL function. To be specific, in order to find an $\epsilon$-approximate optimal solution $\hat{x}$ such that $f(\hat{x}) - f^\star \leq \epsilon$, GD requires the computational complexity of the order $O(\frac{L}{\mu} \log \frac{1}{\epsilon})$. Besides this, different proposed methods, such as the heavy ball method and its accelerated version have been analyzed (Danilova et al., 2020; Wang et al., 2022a). The authors in (Yue et al., 2023) proved the optimality of GD by showing that any first-order method requires at least $\Omega(\frac{L}{\mu} \log \frac{1}{\epsilon})$ gradient costs to find an $\epsilon$ approximation of the optimal solution. Furthermore, many studies focus on the sample complexity when the objective function has a finite-sum structure, i.e., $f(x) = \frac{1}{n} \sum_{i=1}^{n} f_i(x)$, e.g., (Lei et al., 2017; Reddi et al., 2016; Li et al., 2021; Wang et al., 2019b; Bai et al., 2024).

In addition to the minimization problem, extensions of the PL condition, such as two-sided conditions, have been proposed to provide convergence guarantees to saddle points for gradient-based algorithms when addressing minimax optimization problems. For example, the two-sided PL holds when both $h_y(x) := f(x, y)$ and $h_x(y) := -f(x, y)$ satisfy the PL condition (Yang et al., 2020; Chen et al., 2022), or one-sided PL condition holds when only $h_y(x)$ satisfies the PL condition (Guo et al., 2023; Yang et al., 2022). Various types of first-order methods have been applied to such problems, for example, SPIDER-GDA (Chen et al., 2022), AGDA (Yang et al., 2020), Multi-step GDA (Sanjabi et al., 2018; Nouiehed et al., 2019). For additional information on the sample complexity of the methods mentioned earlier and their comparisons, see (Chen et al., 2022) and (Bai et al., 2024).

## 2 $n$-sided PL Condition

**Notations:** Throughout this work, we use $\| \cdot \|$ to denote the Euclidean norm and the lowercase letters to denote a column vector. In particular, we use $x_{-i}$ to denote the vector $x$ without its $i$-th block, where $i \in [n] := \{1, ..., n\}$. The partial derivative of $f(x)$ with respect to the variables in its $i$-th block is denoted as $\nabla_i f(x) := \frac{\partial}{\partial x_i} f(x_i, x_{-i})$ and the full gradient is denoted as $\nabla f(x)$ that is $(\nabla_1 f(x), ..., \nabla_n f(x))$. The partial second order derivative with respect to the $i$-th coordinate is denoted as $\nabla_i^2 f(x) := \frac{\partial^2}{\partial^2 x_i} f(x_i, x_{-i})$. The distance between a point $x$ and a closed set $\mathbb{S}$ is given by $dist(x, \mathbb{S}) := \inf_{s \in \mathbb{S}} \|s - x\|$. The uniform sampling between $a$ and $b$ is denoted as $U(a, b)$.

### 2.1 Definitions and Assumptions

Throughout this paper, we assume the function $f(x) : \mathbb{R}^d \to \mathbb{R}$ belongs to $C^1$, i.e., it is continuously differentiable. Furthermore, we assume it has a Lipschitz gradient.

**Assumption 2.1** (Smoothness). *We assume the L-Lipschitz continuity of the derivative $\nabla f(x)$,*

$$\|\nabla f(x) - \nabla f(y)\| \leq L\|x - y\|, \forall x, y$$

*where $L > 0$ is a constant. In this case, $f(x)$ is also called L-smooth.*

A slightly weaker assumption is coordinate-wise smoothness given below. Note that under the Lipschitz gradient assumption, the coordinate-wise smoothness can be deduced.

**Assumption 2.2** (Coordinate-wise Smoothness). *We assume the coordinate-wise $L_c$-Lipschitz continuity of the derivative $\nabla f(x)$,*

$$\|\nabla_i f(x_i, x_{-i}) - \nabla_i f(x'_i, x_{-i})\| \leq L_c\|x_i - x'_i\|, \quad \forall x_i, x'_i, x_{-i}, \forall i \in [n],$$

*where $L_c > 0$ is a constant. In this case, $f(x)$ is also called a coordinate-wise $L_c$-smooth function.*

**Assumption 2.3** (Lower bounded). *The function $f(x)$ is lower bounded, i.e. $\inf_{x \in \mathbb{R}^d} f(x) > -\infty$.*

We now define two notions of optimality for the minimization problem in eq. (1); Nash Equilibrium (NE) and Stationary point.

**Definition 2.4** (Nash Equilibrium (NE)). *Point $x^\star = (x_1^\star, ..., x_n^\star)$ is called a Nash Equilibrium of function $f(x)$ if*

$$f(x_i^\star, x_{-i}^\star) \leq f(x_i, x_{-i}^\star), \forall i \in [n], \forall x_i \in \mathbb{R}^{d_i}.$$

*We denote the set of all Nash equilibrium points of $f(x)$ by $\mathcal{N}(f)$.*

The other notion, stationary point, is related to the first-order condition of optimality and also relevant for studying gradient-based algorithms.

**Definition 2.5** ($\varepsilon$-Stationary point). *Point $\tilde{x} = (\tilde{x}_1, ..., \tilde{x}_n)$ is called an $\varepsilon$-stationary point of $f(x)$ if $\|\nabla f(\tilde{x})\| \leq \varepsilon$. When $\varepsilon = 0$, the point $\tilde{x}$ is called a stationary point. We denote the set of all $\varepsilon$-stationary points and the set of all stationary points of $f(x)$ by $\mathcal{S}_\varepsilon(f)$ and $\mathcal{S}(f)$, respectively.*

For general nonconvex minimization problems, the above two notions are not necessarily equivalent, i.e., a stationary point may not be a NE. Nevertheless, for the remainder of this work, we assume that the objective function $f$ has at least one NE, i.e., $\mathcal{N}_f \neq \emptyset$. We also assume that $\arg\min_{x_i \in \mathbb{R}^{d_i}} f(x_i, x_{-i})$ is non-empty for any $i \in [n]$ and $x_{-i}$, i.e., there exists a best response to every $x_{-i}$. Note that this is not a limiting assumption given that the function is lower bounded. Below, we formally introduce the $n$-sided PL condition for the function $f(x)$.

**Definition 2.6** ($n$-sided PL Condition). *We say a function $f(x) = f(x_1, ..., x_n)$ satisfies $n$-sided $\mu$-PL condition if there exists a positive constant $\mu > 0$ such that*

$$\|\nabla_i f(x_i, x_{-i})\|^2 \geq 2\mu\big(f(x_i, x_{-i}) - f_{x_{-i}}^\star\big), \quad \forall x \in \mathbb{R}^d, \forall i \in [n], \tag{3}$$

*where $f_{x_{-i}}^\star := \min_{y_i} f(y_i, x_{-i})$.*

We say a function $f(x)$ is $n$-sided PL, if it satisfies the $n$-sided $\mu$-PL condition for some $\mu > 0$. It is worth noting that the $n$-sided PL condition does not imply convexity or the gradient dominance (PL) condition. It is an extension to the PL condition, as when $f$ is independent of $x_{-i}$, i.e., $f(x_i, x_{-i}) = \phi(x_i)$ for some function $\phi$ satisfying the PL condition, then $f$ satisfies the PL condition. Moreover, it is considerably weaker than multi-strong convexity.

Next result shows that under the $n$-sided PL condition, the set of stationary points and the NE set are equivalent. All proofs are presented in the Appendix C. For instance, the set of stationary points and the NE set of $f_0$ in Figure 1 is $\{(-1, -1), (1, 1), (0, 0)\}$.

**Lemma 2.7.** *If $f(x) = f(x_1, ..., x_n)$ satisfies the $n$-sided PL condition, then $\mathcal{S}(f) = \mathcal{N}(f)$*

It is also important to emphasize that, unlike the $n$-sided PL, the two-sided PL condition is defined such that the right-hand side of equation 3 is the difference between the function and its minimum for one coordinate while for the other coordinate it is the difference between the function and its maximum. As a consequence, under the two-sided condition, the stationary points are also global minimax points. However, under the $n$-sided PL condition in definition 2.6, it is no longer possible to ensure that the NE are global minimums. In fact, there could be multiple NEs with different function values. For example, consider the functions $f_0(x, y)$ and $f(x, y)$ illustrated in Figure 1. As

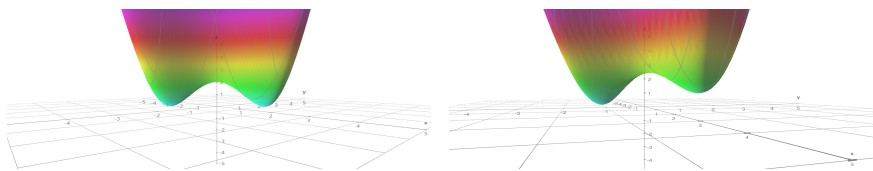

Figure 1: Left is function $f_0(x, y) = (x - 1)^2(y + 1)^2 + (x + 1)^2(y - 1)^2$ and right is function $f(x, y) = f_0(x, y) + \exp(-(y - 1)^2)$.

shown in Appendix B, both functions are 2-sided PL, but their set of NE and the set of minimum points are not equivalent. In particular, both functions have three NE points while, $f_0(x, y)$ has two global minimums and a saddle point, and $f(x, y)$ has a local, a global minimum, and a saddle point.

**Remark 2.8.** *The $n$-sided PL condition is defined coordinated-wise, with the coordinates aligned with the vectors $\{e_1, ..., e_n\}$, where $e_i$ belongs to $\mathbb{R}^d$, such that the entries corresponding to the $i$-th block are one and zero elsewhere. This condition can naturally be extended to $n$-sided directional PL in which the $i$-th inequality is aligned with a designated vector $v_i$. In this extension, the partial gradient and $f^*_{x_{-i}}$ are replaced with their directional variants along vector $v_i$. Note that the results of this work will remain valid in the directional setting, provided that the definitions of NE and the presented algorithms are adjusted to their respective directional variants.*

## 3 ALGORITHMS AND CONVERGENCE ANALYSIS

Within this section, our initial focus is on studying the BCD algorithm for finding a stationary point of equation 1 under the $n$-sided PL condition. Afterward, we propose different variants of BCD algorithms that can provably achieve better convergence rates.

The BCD algorithm is a coordinate-wise approach that iteratively improves its current estimate by updating a selected block coordinate using the first-order partial derivatives until it converges. It is important to note that BCD algorithms typically utilize the partial gradient evaluated at the latest estimated point to update the selected coordinate. Depending on how the coordinates are chosen, various types of BCD algorithms can be devised. For example, coordinates can be selected uniformly at random, *random BCD*, or in a deterministic cyclic sequence, progressing one after another. Algorithm 1 presents the *cyclic BCD* algorithm with learning rates $\{\alpha_i^t\}$. Moreover, to update the $i$-th block at the $t$-th iteration, it employs $\nabla_i f(x_{1:i-1}^t, x_{i:n}^{t-1})$, where $(x_{1:i-1}^t, x_{i:n}^{t-1})$ denotes the latest estimated point and it

---

**Algorithm 1** Cyclic Block Coordinate Descent (BCD)

**Input:** initial point $x^0 = (x_1^0, ..., x_n^0)$, learning rates $\{\alpha_i^t\}$
**for** $t = 1$ **to** $n$ **do**
  **for** $i = 1$ **to** $n$ **do**
    $x_i^t = x_i^{t-1} - \alpha_i^t \nabla_i f(x_{1:i-1}^t, x_{i:n}^{t-1})$
  **end for**
**end for**

---

is $(x_1^t, ..., x_{i-1}^t, x_i^{t-1}, ..., x_n^{t-1})$. Next result shows that when the iterates of the BCD, $\{x^t\}$ are bounded, the output converges to the NE set.

**Theorem 3.1.** *Under the assumption 2.2 and assumption 2.3, if $f(x)$ satisfies $n$-sided PL condition, the iterates $\{x^t\}$ are bounded and the learning rates $\alpha_i^t = \alpha \leq \frac{1}{L_c}$, then $\lim_{t \to +\infty} dist(x^t, \mathcal{N}(f)) = 0$.*

The above result ensures the convergence of BCD to the NE set, but it does not necessarily indicate whether the output converges to a point within the NE set. The convergence to a point within the NE set can be established if further every point in the NE set is isolated, e.g., $f_0$ and $f$ in Figure 1.

**Theorem 3.2.** *Under the assumptions of theorem 3.1, if $\mathcal{N}(f)$ is the union of isolated points, i.e., there exists $\eta > 0$, such that $\min_{\substack{y, z \in \mathcal{N}(f) \\ y \neq z}} \|y - z\| \geq \eta$, then $\{x^t\}$ converges to a point in $\mathcal{N}(f)$.*

It is noteworthy that, following the results of Lee et al. (2016; 2019); Panageas & Piliouras (2016); Ahn et al. (2022), when the function is smooth, and the initial points are chosen randomly, the BCD

algorithm avoids strict saddle points in the NE set almost surely. See the Appendix D for formal statements and proofs.

Although the above results ensure the convergence of BCD when the function is lower bounded and also satisfies the $n$-sided PL, they do not specify the last-iterate convergence rate. Unlike the two-sided PL condition that leads to linear convergence of AGDA to the min-max, the $n$-sided PL condition does not necessarily lead to any specific convergence rate of the BCD. To demonstrate this phenomena, we consider two 2-sided PL functions: $f_1(x, y) = (x + y)^2 + \exp(-1/(x - y)^2)$ for $(x, y) \neq (0, 0)$ and zero otherwise and $f_2(x, y) = (x + y)^2$. We applied the BCD algorithm to both these functions with small enough[2] constant learning rates to find their NE points with different random initializations.

As it is illustrated in Figure 2, the BCD converges linearly for the function $f_2$ while it converges sub-linearly for $f_1$. This example shows that characterizing the convergence rate of the BCD[3] algorithm under the $n$-sided PL condition and the smoothness might not be feasible and further assumptions on the function class are required. In what follows, we study one such assumption that holds for a large class of non-convex functions and characterize the convergence rate of random BCD and GD under this additional assumption.

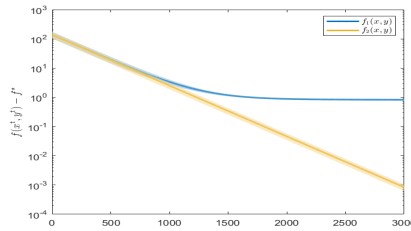

Figure 2: The BCD algorithm applied to functions $f_1(x, y)$ and $f_2(x, y)$. The y-axis is in log scale, thus the BCD demonstrates linear convergence for $f_2$.

### 3.1 CONVERGENCE UNDER AN ADDITIONAL ASSUMPTION

To introduce our additional assumption, we need to define a quantity related to function $f(x)$ denoted by $G_f(x)$ which plays a central role in analyzing the convergence of coordinate-wise algorithms. That is the average of the best responses,

$$G_f(x) := \frac{1}{n} \sum_{i=1}^{n} f(x_i^*(x), x_{-i}), \tag{4}$$

where $x_i^*(x)$ denotes the best response to $x_{-i}$ that is the closest to $x_i$, i.e., $x_i^*(x) \in \arg\min_{y_i}\{\|y_i - x_i\| | f(y_i, x_{-i}) \leq f(z_i, x_{-i}), \forall z_i\}$. It is straightforward to see that $f(x) - G_f(x) \geq 0$ for all $x$. Moreover, if $x^* \in \mathcal{N}_f$, the best response for every block is $x^*$. Conversely, if $f(x^\star) - G_f(x^\star) = 0$, then $f(x^\star) = \min_{x_i} f(x_i, x^\star_{-i}), \forall i$, which implies $x^\star$ is a NE. As a result, we have

**Theorem 3.3.** $x^\star$ *is a NE if and only if* $f(x^\star) - G_f(x^\star) = 0$.

The next result shows that $G_f(x)$ is both differentiable and smooth under the $n$-sided PL condition. See appendix C.4 for a proof.

**Lemma 3.4.** *If* $f(x)$ *satisfies* $n$-sided $\mu$-PL *and satisfies assumption 2.1, then* $\nabla G_f(x)$ *exists and it is* $L'$-Lipschitz, where $L' := L + \frac{L^2}{\mu}$.

Note that if function $f(x)$ is $L$-smooth and $n$-sided $\mu$-PL, then $L \geq \mu$ (see Appendix A). Below, we introduce an additional assumption on $f$ under which the random BCD algorithm achieves a linear convergence rate. This is about how the gradients of $f$ and $G_f$ are aligned

**Assumption 3.5.** *For a given set of points* $\{x^1, x^2, ...\}$, *there exists* $0 \leq \kappa < 1$ *such that for all* $\tau$,

$$\langle \nabla G_f(x^\tau), \nabla f(x^\tau) \rangle \leq \kappa \|\nabla f(x^\tau)\|^2. \tag{5}$$

For instance, the function $f_0(x, y)$ depicted in Figure 1 satisfies this assumption for all points within $\{(x, y) : |x| > 0.75, |y| > 0.75\}$. Note that this set contains both local minimums of $f_0$.

**Theorem 3.6.** *Suppose* $f(x)$ *is* $n$-sided $\mu$-PL *satisfying assumption 2.1 and assumption 3.5 for all the iterates, then random BCD with* $\alpha^t := \alpha \leq \frac{2(1-\kappa)}{2L'+(1+\kappa)L}$ *achieves linear convergence rate, i.e.,*

$$\mathbb{E}[f(x^{t+1}) - G_f(x^{t+1})] \leq \left(1 - \frac{\mu\alpha(1 - \kappa)}{2}\right)\mathbb{E}[f(x^t) - G_f(x^t)].$$

---

[2]Different learning rates were selected, all less than $1/L_c$, where $L_c$ is defined in assumption 2.2.

[3]Similar behavior was also observed from the GD algorithms for these two functions.

*The expectation is taken over the randomness inherent in the procedure for selecting coordinates.*

The GD algorithm, i.e., $x^t = x^{t-1} - \alpha^t \nabla f(x^{t-1})$ can also achieve similar convergence rate.

**Theorem 3.7.** *Suppose $f(x)$ is $n$-sided $\mu$-PL and satisfies assumption 2.1 and assumption 3.5 for all the iterates, then GD with $\alpha^t := \alpha \leq \frac{2(1-\kappa)}{2L' + (1+\kappa)L}$ achieves linear convergence rate, i.e.,*

$$f(x^{t+1}) - G_f(x^{t+1}) \leq \Big(1 - \frac{n\mu\alpha(1-\kappa)}{2}\Big)(f(x^t) - G_f(x^t)).$$

Applying the Cauchy-Schwarz inequality, it is straightforward to see that a stronger assumption than assumption 3.5 is that there exists $0 \leq \kappa < 1$, such that $\|\nabla G_f(x^t)\| \leq \kappa \|\nabla f(x^t)\|$. On the other hand, the following result shows that $\|\nabla G_f\|$ is always bounded from above by $\|\nabla f\|$ for $n$-sided PL function $f$, but with a constant greater than one. Thus, for instance, if the function $f$ is such that this constant is less than one for the iterates of the random BCD algorithm, then linear convergence can be guaranteed by theorem 3.6. This is indeed the case for functions such as $f_0$ and the linear residual network problem (see Section 4). Moreover, as we showed in Appendix F, there exists a neighborhood around every isolated local minimum of smooth functions such that, on average, the condition in equation 5 holds for all iterates of the GD dynamics.

**Lemma 3.8.** *For an $n$-sided $\mu$-PL function $f(x)$ satisfying assumption 2.1, let $C_f := \frac{L}{\sqrt{n}\mu} + 1$, then $\|\nabla G_f(x)\| \leq C_f \|\nabla f(x)\|$, for all $x$.*

### 3.2 CONVERGENCE WITH THE EXACT BEST RESPONSES BUT WITHOUT ADDITIONAL ASSUMPTION

Herein, we study the setting in which assumption 3.5 does not hold. As we discussed earlier, in this setting, the BCD and GD algorithms may demonstrate different convergence rates. Thus, our objective in the remainder of this section is to develop variants of the random BCD and GD algorithms so that close to linear convergence is still achievable. We accomplish this objective, first by designing algorithms equipped with the knowledge of the best responses, $\{x_i^*(x^t)\}$, at each iteration $t$. More precisely, we initially propose algorithms that presume access to the exact values of the best responses at each iteration. Subsequently, we refine this assumption by integrating a sub-routine into the proposed algorithms capable of approximating the best responses. For the sake of simplicity and space, we describe our block coordinate variants here and the GD variants and their convergence analysis are presented in the Appendix G. To present our theoretical result, we need the following definition.

**Definition 3.9** (($\theta, \nu$)-PŁ condition). *The function $f$ with $\min_x f(x) = 0$ satisfies $(\theta, \nu)$-PL condition iff there exists $\theta \in [1, 2)$ and $\nu > 0$ such that $\|\nabla f(x)\|^\theta \geq (2\nu)^{\theta/2} f(x)$.*

It has been proved by Lojasiewicz (1963) that for any $C^1$ analytic function, there exists a neighborhood $U$ around the minimizer where $(\theta, \nu)$-PL condition is satisfied.

Algorithm 2 presents the steps of our modified version of the random BCD. In this algorithm, instead of updating along the direction of $-\nabla_{i^t} f(x)$, where $i^t$ denotes the chosen coordinate at iteration $t$, a linear combination of $\nabla_{i^t} f(x)$ and $\nabla_{i^t} G_f(x)$ is used to refine the updating directions. The coefficient of this linear combination, $k^t$, is adaptively selected based on the current estimated point. It is important to mention that $\nabla G_f(x)$ can be computed using the gradient of $f$ and the best responses.

$$\nabla G_f(x) = \frac{1}{n} \sum_{i=1}^{n} \nabla f\big(x_i^*(x), x_{-i}\big). \tag{6}$$

**Theorem 3.10.** *For $n$-sided $\mu$-PL function $f(x)$ satisfying assumption 2.1, by applying algorithm 2,*

- *in Case 1 with $\alpha \leq \frac{2(1-\gamma)}{2L'+(1+\gamma)L}$, we have $\mathbb{E}[f(x^{t+1}) - G_f(x^{t+1})|x^t] \leq \big(1 - \frac{\mu\alpha(1-\gamma)}{2}\big)(f(x^t) - G_f(x^t))$,*

- *in Case 2 with $\alpha \leq \min\{\frac{1}{2(L+L')}, \frac{C}{2(L+L')}\}$, we have*

$$\mathbb{E}[f(x^{t+1}) - G_f(x^{t+1})|x^t] \leq \Big(1 - \frac{(L+L')\mu\alpha^2}{2}\Big)(f(x^t) - G_f(x^t)),$$

---

**Algorithm 2** Ideal Adaptive Randomized Block Coordinate Descent (IA-RBCD)

---

**Input:** initial point $x^0 = (x_1^0, ..., x_n^0)$, $T$, learning rates $\alpha$, $0 \leq \gamma < 1$ and $C > 0$
**for** $t = 0$ **to** $T - 1$ **do**
    sample $i^t$ uniformly from $\{1, 2, ..., n\}$
    **if** $\langle \nabla G_f(x^t), \nabla f(x^t) \rangle \leq \gamma \|\nabla f(x^t)\|^2$ **then**
        $k^t = 0$                                                     `:Case 1:`
    **else if** $\frac{(\|\nabla G_f(x^t)\|^2 - \langle \nabla f(x^t), \nabla G(x^t) \rangle)^2}{\langle \nabla f(x^t), \nabla G(x^t) \rangle^2} > C$ **then**
        $k^t = -2 + \frac{\langle \nabla f(x^t), \nabla G_f(x^t) \rangle}{\|\nabla G_f(x^t)\|^2}$                                 `:Case 2:`
    **else**
        $k^t = -1$                                                     `:Case 3:`
    **end if**
    $x_{i^t}^{t+1} = x_{i^t}^t - \alpha(\nabla_{i^t} f(x^t) + k^t \nabla_{i^t} G_f(x^t))$,    $x_i^{t+1} = x_i^t$ if $i \neq i^t$
**end for**

---

- *in Case 3 with $\alpha \leq \frac{1}{L+L'}$, $f - G_f$ is non-increasing. Furthermore, if $f - G_f$ satisfies $(\theta, \nu)$-PL condition and case 3 are satisfied from iterates $t$ to $t + k$, we have*

$$\mathbb{E}[f(x^{t+k}) - G_f(x^{t+k})|x^t] \leq \mathcal{O}\Big(\frac{f(x^t) - G_f(x^t)}{k^{\frac{\theta}{2-\theta}}}\Big).$$

*The exact constant terms are provided in the proof.*

According to this result, IA-RBCD in 2 demonstrates linear convergence for two out of three cases. When the third case occurs finitely many times, for instance, if there exists a neighborhood around an isolated NE point such that the third case does not occur (e.g., function $f_0$ in Figure 1), then linear converge is guaranteed by IA-RBCD. Since rigorously verifying these cases is intractable, we empirically verify them for different well-known problems in the next section.

It is crucial to highlight that BCD requires assumption 2.3 to converge to the NE (Xu & Yin, 2013) and almost surely avoids strict saddle points (Lee et al., 2016). However, theorem 3.10 shows that under the specified assumptions, IA-RBCD converges to the NE irrespective of these conditions.

### 3.3 CONVERGENCE WITH APPROXIMATED BEST RESPONSES AND WITHOUT ADDITIONAL ASSUMPTION

Evaluating $G_f$ at a given point requires the knowledge of the best responses at that point. Often, these best responses are not known a priori and they have to be computed at each iteration. Fortunately, since in our study, $f(x)$ satisfies the $n$-sided PL condition, the best responses can be efficiently approximated, by applying GD algorithm with the partial gradients as a sub-routine. Algorithm 4 presents the steps of this sub-routine and Algorithm 3 shows the steps of our adaptive random BCD algorithm. The main difference between algorithms 2 and 3 is that at every iteration, A-RBCD approximates the best response function by gradient descent. This is efficient as it converges to the $G_f$ at a linear rate. And interestingly, the number of steps for approximating $G_f(x)$, $T'$, only depends on the function parameters and it is independent of the final precision of $f - G_f$.

**Theorem 3.11.** *For an $n$-sided $\mu$-PL function $f(x)$ satisfying assumption 2.1, by implementing algorithm 3 with $\beta \leq \frac{1}{L}$ and $T' \geq \log\big(\frac{169nL^2}{\mu^2\gamma^2\alpha^6}\big) / \log(\frac{1}{1-\mu\beta})$,*

- *in Case 1 with $\alpha \leq \frac{2(1-\gamma)}{2L'+(1+\gamma)L}$, we have $\mathbb{E}[f(x^{t+1}) - G_f(x^{t+1})|x^t] \leq \big(1 - \frac{\mu\alpha(1-\gamma)}{2}\big)(f(x^t) - G_f(x^t))$,*

- *in Case 2 with $\alpha \leq \min\big\{\frac{1}{\sqrt{C_f}}, \big(\frac{3C\gamma}{(13+12\gamma)C_f}\big)^{1/2}, \frac{71C\gamma^2}{676(L+L')}, \frac{3\gamma(L+L')\mu}{(13+108\gamma)LC_f^4}, \frac{1}{2(L+L')}\big\}$, we have*

$$\mathbb{E}[f(x^{t+1}) - G_f(x^{t+1})|x^t] \leq \Big(1 - \frac{(L+L')\mu\alpha^2}{4}\Big)(f(x^t) - G_f(x^t)).$$

- *in Case 3 with $\alpha \leq \min\big\{\frac{1}{L+L'}, \big(\frac{13}{12(1+C_f)}\big)^{1/3} \frac{\|\nabla f(x^t) - \nabla G_f(x^t)\|}{\|\nabla f(x^t)\|}\big\}$, $f - G_f$ is non-increasing. Furthermore, if $f - G_f$ satisfies $(\theta, \nu)$-PL condition and case 3 occurs from iterates $t$ to $t + k$, then*

$$\mathbb{E}[f(x^{t+k}) - G_f(x^{t+k})|x^t] \leq \mathcal{O}\Big(\frac{f(x^t) - G_f(x^t)}{k^{\frac{\theta}{2-\theta}}}\Big)$$

---

**Algorithm 3** Adaptive randomized Block Coordinate Descent (A-RBCD)

---

**Input:** initial point $x^0 = (x_1^0, ..., x_n^0)$, $T, T'$, learning rates $\alpha, \beta, 0 < \gamma < 1$ and $C > 0$
**for** $t = 0$ **to** $T - 1$ **do**
  sample $i^t$ uniformly from $\{1, 2, ..., n\}$
  $y^{t,T'} = \text{ABR}(x^t, T', \beta)$                                 :Algorithm 4
  compute $\tilde{\nabla} G_f(x^t) = \frac{1}{n} \sum_{l=1}^{n} \nabla f(y_l^{t,T'}, x_{-l}^t)$
  **if** $\langle \tilde{\nabla} G_f(x^t), \nabla f(x^t) \rangle \leq (\gamma - \gamma \frac{\alpha_t^3}{13}) \|\nabla f(x^t)\|^2$ **then**
    $\tilde{k}^t = 0$                                             :Case 1:
  **else if** $\frac{(\|\tilde{\nabla} G_f(x^t)\|^2 - \langle \nabla f(x^t), \tilde{\nabla} G_f(x^t) \rangle)^2}{\|\tilde{\nabla} G_f(x^t)\|^4} > C$ **then**
    $\tilde{k}^t = -2 + \frac{\langle \nabla f(x^t), \tilde{\nabla} G_f(x^t) \rangle}{\|\tilde{\nabla} G_f(x^t)\|^2}$                       :Case 2:
  **else**
    $\tilde{k}^t = -1$                                            :Case 3:
  **end if**
  $x_{i^t}^{t+1} = x_{i^t}^t - \alpha(\nabla_{i^t} f(x^t) + \tilde{k}^t \tilde{\nabla}_{i^t} G_f(x^t)), \quad x_i^{t+1} = x_i^t, \text{ if } i \neq i^t$
**end for**

---

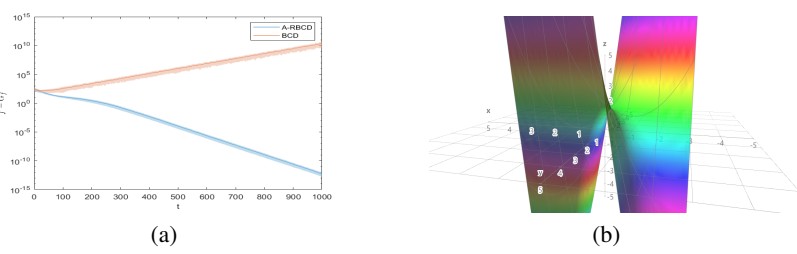

(a)                                    (b)

Figure 3: (a) Performance of A-RBCD (blue) and BCD (red) on function $f(x, y)$ shown in (b).

# 4 APPLICATIONS

Herein, we discuss two well-known nonconvex problems that satisfy the $n$-sided PL condition.

**Function with only strict saddle point:** We consider the quadratic function $f(x, y) = (x - 1)^2 + 4(x + 0.1 \cos(x))y + (y + 0.1 \sin(y))^2$. The problem aims at finding the NE $(x^\star, y^\star)$, i.e.,

$$f(x^\star, y^\star) \leq f(x, y^\star), \forall x, \quad f(x^\star, y^\star) \leq f(x^\star, y), \forall y. \tag{7}$$

Figure 3 represents the convergence results of A-RBCD and BCD with 100 random initialization. The iterates of A-RBCD always converge to the NE at a linear rate while BCD diverges. Note that the NE is a strict saddle point.

**Linear Residual Network:** It aims at learning linear transformation $R : \mathbb{R}^d \to \mathbb{R}^d$, such that $y = Rx + \xi$, where $\xi \sim \mathcal{N}(0, I_d)$ and $I_d$ denotes the identity matrix of dimension $d$. The learned model can be parameterized by a sequence of weight matrices $A_1, ..., A_n \in \mathbb{R}^{d \times d}$, such that $h_0 = x$, $h_j = (I + A_j)h_{j-1}, \hat{y} = h_n$. Thus, the objective function of this problem is given by

$$f(A_1, ..., A_n) := \mathbb{E}[\|\hat{y} - y\|^2] = \mathbb{E}\big[\|(I + A_n)...(I + A_1)x - Rx - \xi\|^2\big].$$

Even though $(I + A_n) \cdots (I + A_1)$ is a linear map, the optimization problem over the factored variables $(A_1, ..., A_n)$ is non-convex (Hardt & Ma, 2017). More precisely, we considered two settings: (1) $d = 3, n = 5$ and (2) $d = 5, n = 10$ with covariance matrices $\Sigma = \mathbb{E}[xx^T] = I_d$, and applied the A-RBCD algorithm to both settings. Figure 4 illustrates the resulting error curves on a log-scaled y-axis, obtained from 100 trials. Each trial is obtained by randomly selecting the diagonals of matrix $R$ according to $U(0.5, 1.5)$ and initializing $A_i$s with random entries according to $U(-0.1, 0.1)$.

**Infinite Horizon $n$-player Linear-quadratic (LQR) Game:** The objective function of this game can be formulated as

$$\mathbb{E}_{x_0 \sim \mathcal{D}} \Big[ \sum_{t=0}^{+\infty} [(x^t)^T Q x^t + \sum_{i=1}^{n} ((u_i^t)^T R_i u_i^t] \Big],$$

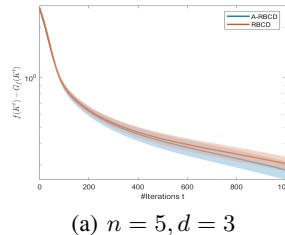 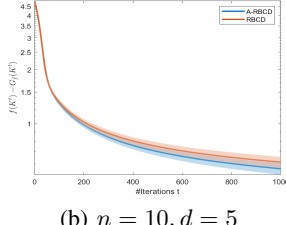

(a) $n = 5, d = 3$        (b) $n = 10, d = 5$

Figure 4: The performance of the A-RBCD and RBCD on linear residual network problems for different network sizes illustrates linear convergence, as advocated by theorem 3.6.

where $x_t$ denotes the state, $u_i^t$ is the input of $i$-th player at time $t$, and $i \in [n]$. The state transition of the system is characterized by $x^{t+1} = Ax^t + \sum_{i=1}^n B_i u_i^t$, where $A \in \mathbb{R}^{k \times k}$ and $B \in \mathbb{R}^{k \times d}$. When players apply linear feedback strategy, i.e., $u_i^t = -K_i x^t$,, the objective function becomes

$$f(K_i, K_{-i}) = \mathbb{E}_{x_0 \sim \mathcal{D}} \Big[ \sum_{t=0}^{+\infty} [(x^t)^T Q x^t + \sum_{i=1}^n ((K_i x_i^t)^T R_i K_i x_i^t)] \Big].$$

If $K_i$s are bounded and $\Sigma_0 = \mathbb{E}_{x^0 \sim \mathcal{D}}[x^0 (x^0)^T]$ is full rank, the objective function $f$ satisfies the $n$-sided PL condition (see appendix E.1 for a proof). However, as it is discussed in Fazel et al. (2018), even the objective of one-player LQR is not convex. Subsequently, the objective function of the $n$-player LQR game is not multi-convex. See appendix E.2 for examples.

We applied our A-RBCD algorithm to this problem when $A \in \mathbb{R}$, $B_i \in \mathbb{R}^{1 \times d}$ and the entries of $B_i$, $Q$ and the diagonal entries of $R_i$ were sampled according to $\frac{1}{nd} U(0, 1)$, $U(0, 1)$ and $U(0, 1)$, respectively. We set the learning rate $\alpha = 0.05$ and random initialization $K_i \sim U(0, 1)^d$ for all $i$. Fig. 5 demonstrates the resulting error curve, $f(K^t) - G_f(K^t)$, and $\rho := \frac{\langle \nabla f(K^t), \nabla G_f(K^t) \rangle}{\|\nabla f(K^t)\|^2}$. This shows that during the updating procedure, the third case did not occur. Plots are obtained from 50 trials.

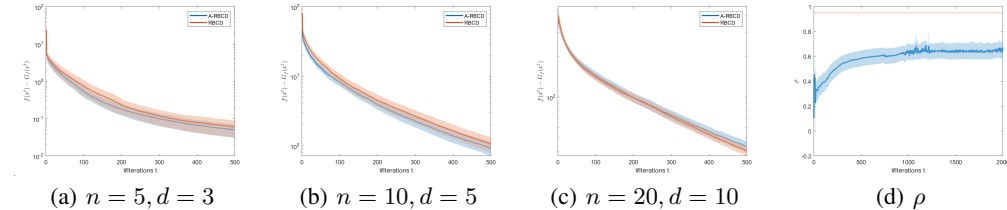

(a) $n = 5, d = 3$    (b) $n = 10, d = 5$    (c) $n = 20, d = 10$    (d) $\rho$

Figure 5: The performance of the A-RBCD and RBCD on $n$-player LQR for different game sizes. The y-axis of (a)-(c) are in the log scale.

## 5   CONCLUSION

In this paper, we identified a subclass of nonconvex functions called $n$-sided PL functions and studied the convergence of GD-based algorithms, particularly the BCD algorithm, for finding their NEs. The $n$-sided PL condition is a reasonable extension of the gradient dominance condition, which holds in various problems. We showed that the convergence rate of such first-order algorithms in this subclass of functions depends on a local relation between the function $f$ and the average of the best responses $G_f$. Subsequently, we proposed two novel algorithms, IA-RBCD and A-RBCD, equipped with $G_f$, that provably converge to the NE set almost surely with random initialization even if the function is not lower bounded and has strict saddle points. We hope this work can shed some light on the understanding of nonconvex optimization.

## 6 REPRODUCIBILITY STATEMENT

We affirm that all the result from this paper are reproducible. The detailed proof of lemma and theorem are given in the appendix. The source code for the applications section is in the supplementary materials.

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

# Appendix

## A   Technical Lemmas

**Lemma A.1.** *Karimi et al. (2016). If $f(\cdot)$ is l-smooth and it satisfies PL with constant $\mu$, then it also satisfies error bound (EB) condition with $\mu$, i.e.*

$$\|\nabla f(x)\| \geq \mu \|x_p - x\|, \forall x,$$

*where $x_p$ is the projection of $x$ onto the optimal set, also it satisfies quadratic growth (QG) condition with $\mu$, i.e.*

$$f(x) - \min_y f(y) \geq \frac{\mu}{2} \|x_p - x\|^2, \forall x.$$

*Conversely, if $f(\cdot)$ is l-smooth and satisfies EB with constant $\mu$, then it satisfies PL with constant $\frac{\mu}{l}$.*

**Lemma A.2.** *If $f(\cdot)$ is L-smooth and it satisfies n-sided $\mu$-PL condition, then $L \geq \mu$.*

*Proof.* From $L$-smoothness, we have

$$\|\nabla_i f(x_i, x_{-i}) - \nabla_i f(y_i, x_{-i})\| \leq \|\nabla f(x_i, x_{-i}) - \nabla f(y_i, x_{-i})\| \leq L\|x_i - y_i\|, \forall x_i, \ y_i.$$

It indicates,

$$f(y_i, x_{-i}) - f(x_i, x_{-i}) \leq \langle \nabla_i f(x_i, x_{-i}), y_i - x_i \rangle + \frac{L}{2} \|x_i - y_i\|^2.$$

Let $y_i = x_i - \nabla_i f(x_i, x_{-i})/L$. This leads to

$$f(x) - f(x_i^*(x_{-i}), x_{-i}) \geq \frac{1}{2L} \|\nabla_i f(x)\|^2.$$

On the other hand, from the $n$-side PL, we get

$$f(x) - f(x_i^*(x_{-i}), x_{-i}) \leq \frac{1}{2\mu} \|\nabla_i f(x)\|^2.$$

Putting the above inequalities together concludes the result.  □

**Lemma A.3.** *If $f(\cdot)$ is L-smooth and it satisfies n-sided $\mu$-PL condition, then*

$$\frac{1}{2nL} \|\nabla f(x)\|^2 \leq f(x) - G_f(x) \leq \frac{1}{2n\mu} \|\nabla f(x)\|^2.$$

*Proof.* This is a direct corollary from the last two inequalities of lemma A.2.  □

## B   Examples and Application

### B.1   Function $f_1(x, y) = (x - 1)^2 (y + 1)^2 + (x + 1)^2 (y - 1)^2$

Due to symmetry, we only show the condition for the first coordinate.

$$\nabla_x f_1(x, y) = 2(x - 1)(y + 1)^2 + 2(x + 1)(y - 1)^2 = 4x(y^2 + 1) - 8y,$$

$$f_y^* = 2(y^2 - 1)^2 / (y^2 + 1),$$

$$G_{f_1}(x, y) = \frac{(x^2 - 1)^2}{x^2 + 1} + \frac{(y^2 - 1)^2}{y^2 + 1},$$

$$\nabla G_{f_1}(x, y) = \left( \frac{2x(x^2 - 1)(x^2 + 3)}{(x^2 + 1)^2}, \frac{2y(y^2 - 1)(y^2 + 3)}{(y^2 + 1)^2} \right)$$

Thus, the 2-sided PL holds iff $\exists \mu > 0$, s.t. for all $x$ and $y$

$$2\Big((x - 1)(y + 1)^2 + (x + 1)(y - 1)^2\Big)^2$$

$$- \mu\Big((x - 1)^2 (y + 1)^2 + (x + 1)^2 (y - 1)^2 - 2\frac{(y^2 - 1)^2}{y^2 + 1}\Big) \geq 0.$$

The left-hand side is a quadratic equation with respect to $x$ and for $\mu = 2$, it is

$$\left((y+1)^2 + (y-1)^2 - 1\right)\left(x^2\left((y+1)^2 + (y-1)^2\right) - 2x\left((y+1)^2 - (y-1)^2\right)\right)$$
$$+ \left((y+1)^2 + (y-1)^2 - 1\right)\left((y+1)^2 + (y-1)^2 - 4\frac{(y-1)^2(y+1)^2}{(y-1)^2 + (y+1)^2}\right).$$

The above expression is positive for all $x$ and $y$.

**Analysis of the origin:** Although, the origin point is a stationary point of $f_1$ since the Hessian at this point is not positive semi-definite, it is not a local minimum. However, it is straightforward to see that $(0, 0)$ is in fact a NE of $f_1(x, y)$. Note that the Hessian at the origin is

$$H_f(0, 0) = \begin{bmatrix} 4 & -8 \\ -8 & 4 \end{bmatrix} \not\succeq 0.$$

### B.2 FUNCTION $f_2(x, y) = (x - 1)^2(y + 1)^2 + (x + 1)^2(y - 1)^2 + \exp{-(y - 1)^2}$

For this function, we have

$$\nabla_x f_2(x, y) = 2(x - 1)(y + 1)^2 + 2(x + 1)(y - 1)^2,$$
$$\nabla_y f_2(x, y) = 2(y - 1)(x + 1)^2 + 2(y + 1)(x - 1)^2 - 2(y - 1)\exp(-(y - 1)^2).$$

and

$$\nabla_x^2 f_2(x, y) = 2(y + 1)^2 + 2(y - 1)^2 \geq 4,$$
$$\nabla_y^2 f_2(x, y) = 2(x + 1)^2 + 2(x - 1)^2 + 4(y - 1)^2 \exp(-(y - 1)^2) - 2\exp(-(y - 1)^2) \geq 2.$$

It is straightforward to see that this function is smooth as the second-order derivatives are upper-bounded. Moreover, since both the second-order derivatives are strictly positive, then it is 2-sided PL. It is noteworthy that $(0, 0)$ is also an NE for this function but it is not a local minimum as the Hessian at the origin is not positive semi-definite.

### B.3 FUNCTION $f(x, y) = x^2 + 4y^2 + 3\sin^2 y + 4\sin^2 x \sin^2 y$

We can derive that $\operatorname{argmin}_x f(x, y) = 0$ and $\operatorname{argmin}_y f(x, y) = 0$. Then compute the gradients:

$$\nabla_x f(x, y) = 2x + 3\sin(2x)\sin^2(y),$$
$$\nabla_x f(x, y) = 8y + 3\sin(2y) + 4\sin^2(x)\sin(2y).$$

and

$$|\nabla_x^2 f(x, y)| = |2 + 6\cos(2x)\sin^2(y)| \leq 8,$$
$$|\nabla_y^2 f(x, y)| = |8 + 6\cos(2y) + 8\sin^2(x)\cos(2y)| \leq 22.$$

so $f(\cdot, y)$ is $L_1$-smooth with $L_1 = 8$ and $f(x, \cdot)$ is $L_2$-smooth with $L_2 = 22$. Then note that

$$\frac{|\nabla_x f(x, y)|}{|x - x^\star(y)|} = \frac{|\nabla_x f(x, y)|}{|x|} = \frac{|2x + 3\sin(2x)\sin^2(y)|}{|x|} \geq \frac{1}{2},$$
$$\frac{|\nabla_x f(x, y)|}{|x - x^\star(y)|} = \frac{|\nabla_y f(x, y)|}{|y|} = \frac{|8y + 3\sin(2y) + 4\sin^2(x)\sin(2y)|}{|y|} \geq \frac{9}{2}.$$

So $f(\cdot, y)$ satisfies EB with $\mu_{EB1} = \frac{1}{2}$ and $f(x, \cdot)$ satisfies EB with $\mu_{EB2} = \frac{9}{2}$. By Lemma lemma A.1, we have $f(\cdot, y)$ satisfies PL with $\mu_1 = \frac{1}{16}$ and $f(x, \cdot)$ satisfies PL with $\mu_2 = \frac{9}{44}$. Moreover, this function satisfies Assumption 3.5 as it is shown in Figure 6. Since $G_f$ is not straightforward to compute for this function, we applied the A-RBCD algorithm, and the error is presented in Figure 6.

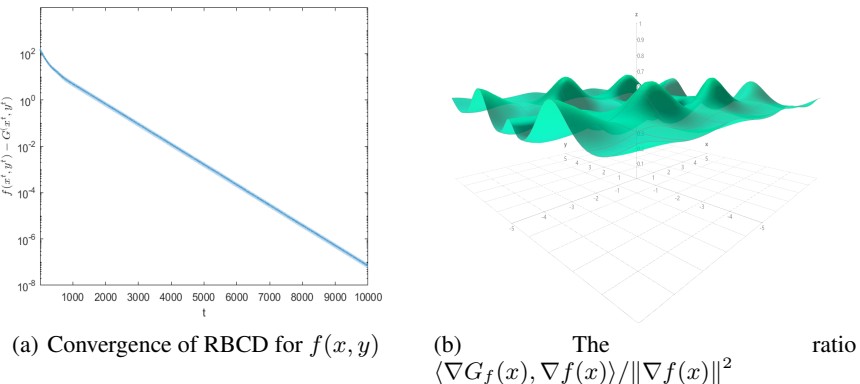

(a) Convergence of RBCD for $f(x, y)$     (b) The ratio $\langle \nabla G_f(x), \nabla f(x) \rangle / \|\nabla f(x)\|^2$

Figure 6: Result of applying random BCD to the $f(x, y) = x^2 + 4y^2 + 3\sin^2 y + 4\sin^2 x \sin^2 y$. Right shows that the ratio is less than one for all points around (0,0), i.e., Assumption 3.5 holds true for this function, and thus by Theorem 3.6, random BCD converges linearly as it is also shown in the left plot.

---

**Algorithm 4** Approximating Best Responses (ABR)

---

   **Input:** Point $x = (x_1, ..., x_n)$, positive number $\beta$ and $T'$
   **for** $j = 1, ..., n$ **do**
      $y_j^0 = x_j$
      **for** $\tau = 0, ..., T' - 1$ **do**
         $y_j^{\tau+1} = y_j^\tau - \beta \nabla_j f(y_j^\tau, x_{-j})$
      **end for**
   **end for**
   **Output:** $y^{T'} = (y_1^{T'}, ..., y_n^{T'})$

---

## C   Technical Proofs

### C.1   Proof of Lemma 2.7

Stationary point $\implies$ Nash Equilibrium: If a point $x$ satisfies $\nabla f(x) = 0$, then the partial derivative $\nabla_{x_i} f(x) = 0, \forall i \in [n]$. From the definition of $n$-sided PL and $f_{x_{-i}}^\star$, we have

$$0 = \nabla_i f(x) \geq 2\mu(f(x_i, x_{-i}) - f_{x_{-i}}^\star) \geq 0, \forall i \in [n],$$
$$\implies f(x_i, x_{-i}) = f_{x_{-i}}^\star = \min_{y_i} f(y_i, x_{-i}), \forall i \in [n],$$
$$\implies f(x_i, x_{-i}) \leq f(\tilde{x}_i, x_{-i}), \forall \tilde{x}_i, \forall i \in [n],$$

which means $x$ satisfies the definition of Nash Equilibrium.

If $f$ is differentiable, then Nash Equilibrium $\implies$ Stationary point: If a point $x$ is a Nash Equilibrium, then $f(x_i, x_{-i}) \leq f(\tilde{x}_i, x_{-i}), \forall \tilde{x}_i, \forall i \in [n]$. Based on the first order optimality condition, we have

$$\nabla_i f(x_i, x_{-i}) = 0, \forall i \in [n],$$

which indicates $\nabla f(x) = 0$.

## C.2 PROOF OF THEOREM 3.1

From the Lipschitz gradient assumption, if $\alpha \leq \frac{1}{L_c}$, we have

$$f(x_{1:i}^t, x_{i+1:n}^{t-1}) - f(x_{1:i-1}^t, x_{i:n}^{t-1}) \leq \langle \nabla_i f(x_{1:i-1}^t, x_{i:n}^{t-1}), x_i^t - x_i^{t-1} \rangle + \frac{L_c}{2} \|x_i^t - x_i^{t-1}\|^2,$$

$$= -(\alpha - \frac{\alpha^2 L_c^2}{2})\|x_i^t - x_i^{t-1}\|^2,$$

$$\leq -\frac{\alpha}{2}\|\nabla_i f(x_{1:i-1}^t, x_{i:n}^{t-1})\|^2.$$

In consequence,

$$f(x_{1:i-1}^t, x_{i:n}^{t-1}) - f(x_{1:i}^t, x_{i+1:n}^{t-1}) \geq \frac{\alpha}{2}\|\nabla_i f(x_{1:i-1}^t, x_{i:n}^{t-1})\|^2 = \frac{\alpha L_c^2}{2}\|x_i^{t-1} - x_i^t\|^2. \quad (8)$$

where the second inequality comes from the quadratic growth of the PL function and the third inequality comes from the Lipschitzness of the gradient. By iterating over all blocks, we have

$$f(x^{t-1}) - f(x^t) = \sum_{i=1}^n f(x_{1:i-1}^t, x_{i:n}^{t-1}) - f(x_{1:i}^t, x_{i+1:n}^{t-1})$$

$$\geq \sum_{i=1}^n \frac{\alpha L_c^2}{2}\|x_i^{t-1} - x_i^t\|^2 = \frac{\alpha L_c^2}{2}\|x^{t-1} - x^t\|^2, \quad (9)$$

where $x^t = \{x_1^t, ..., x_n^t\}$. By iterating overall outer loops, we have

$$f(x^0) - f(x^T) = \sum_{t=1}^T f(x^{t-1}) - f(x^t) \geq \frac{\alpha L_c^2}{2} \sum_{t=1}^T \|x^{t-1} - x^t\|^2.$$

Since $f(x)$ is lower bounded by $\bar{f} = \inf_x f(x)$, we have

$$\sum_{t=1}^T \|x^{t-1} - x^t\|^2 \leq \frac{\alpha L_c^2}{2}(f(x^0) - f(x^T)) \leq \frac{\alpha L_c^2}{2}(f(x^0) - \bar{f}) < +\infty. \quad (10)$$

Since the sequence $\{x^t\}_0^\infty$ is bounded, there exists at least a limit point. For every limit point $\bar{x}$, we denotes $\{x^{k^t}\}$ as its corresponding subsequence such that $\lim_{t\to+\infty} x^{k^t} = \bar{x}$. From eq. (10), we have $\lim_{t\to+\infty} \|x_{t-1} - x_t\| = 0$. As a result, the subsequence $\{x^{k^t+1}\}$ also converge to $\bar{x}$. From the block coordinate gradient descent, we know that

$$x_i^{k^t+1} = x_i^{k^t} - \alpha \nabla_i f(x_{1:i-1}^{k^t+1}, x_{i:n}^{k^t}), \forall i \in [n], \forall t.$$

As $t \to +\infty$, $x_i^{k^t+1} \to \bar{x}_i$ and $x_i^{k^t} \to \bar{x}_i$. We have

$$\bar{x}_i = \bar{x}_i - \alpha \nabla_{x_i} f(\bar{x}), \forall i \in [n], \implies \nabla_i f(\bar{x}) = 0, \forall i \in [n].$$

It implies $\bar{x}$ is a stationary point. From Lemma 2.7, it also implies that $\bar{x}$ is a Nash Equilibrium. As a result, every limit point of $\{x^t\}$ is also a Nash Equilibrium as long as $\{x_t\}$ is bounded.

If we assume that $\{x^t\}$ doesn't converge to Nash Equilibrium, then there exists a positive constant $\epsilon$ a subsequence such that $\text{dist}(x^{k^t}, \mathcal{N}(f)) \geq \epsilon, \forall t$. Since this subsequence is also bounded, then this subsequence must have a limit point $\bar{x} \in \mathcal{N}(f)$, which is a contradiction.

□

## C.3 PROOF OF COROLLARY 3.2

Since $dist(x^t, \mathcal{N}) \to 0$, there exists an integer $T_1 > 0$ such that $x^t \in B(\mathcal{N}, \frac{\eta}{3}), \forall t \geq T_1$, where $B(\mathcal{N}, \frac{\eta}{3}) = \{x | \min_{y \in \mathcal{N}} \|x - y\| < \frac{\eta}{3}\}$. From theorem 3.1, we know that $\lim_{t\to+\infty} \|x_t - x_{t+1}\| = 0$. As a result, there exists an integer $T_2 > 0$ such that $\|x^t - x^{t+1}\| < \frac{\eta}{3}, \forall t \geq T_2$.

We denote $T = \max\{T_1, T_2\}$ and assume $\|x^T - \bar{x}\| \leq \frac{\eta}{3}$, where $\bar{x} \in \mathcal{N}$. Notice that $\bar{x}$ is a unique point at every time $t$, because

$$\|x^t - y\| \geq \|\bar{x} - y\| - \|x^t - \bar{x}\| > \eta - \frac{\eta}{3} = \frac{2\eta}{3} > \frac{\eta}{3},$$

for any $y \in \mathcal{N}$ and $y \neq \bar{x}$. Then,

$$\|x^{t+1} - \bar{x}\| \leq \|x^{t+1} - x^t\| + \|x^t - \bar{x}\| < \frac{2\eta}{3}.$$

For any $y \in \mathcal{N}$ and $y \neq \bar{x}$, we have

$$\|x^{t+1} - y\| \geq \|\bar{x} - y\| - \|x^{t+1} - \bar{x}\| > \eta - \frac{2\eta}{3} = \frac{\eta}{3}.$$

So we always have $\|x^t - \bar{x}\| \leq \frac{\eta}{3}$ for all $t \geq T$ as we have $x^t \in B(\mathcal{N}, \frac{\eta}{3})$. We conclude that $\{x^t\}$ converge to the unique point $\bar{x}$ as $dist(x^t, \mathcal{N}) \to 0$.

$\square$

### C.4 PROOF OF LEMMA 3.4

Based on the Lipschitzness of the $\nabla f$, we have that

$$\|\nabla_i f(x_i^\star(y), x_{-i})\| = \|\nabla_i f(x_i^\star(y), x_{-i}) - \nabla_i f(x_i^\star(y), y_{-i})\| \leq L\|x_{-i} - y_{-i}\|.$$

Also, from $n$-sided PL condition and lemma A.1,

$$\|\nabla_i f(x_i^\star(y), x_{-i})\| \geq \mu \|x_i^\star(y) - x_i^\star(x_i^\star(y), x_{-i})\|.$$

From these two inequalities, we know that

$$\|x_i^\star(y) - x_i^\star(x_i^\star(y), x_{-i})\| \leq \frac{L}{\mu}\|x_{-i} - y_{-i}\|.$$

Then, we can show the smoothness of $g_i(x_{-i}) := \min_{x_i} f(x_i, x_{-i})$.

$$\begin{aligned}
\|\nabla g_i(x_{-i}) - \nabla g_i(y_{-i})\| &= \|\nabla_{-i} f(x_i^\star(x_i^\star(y), x_{-i}), x_{-i}) - \nabla_{-i} f(x_i^\star(y), y_{-i})\|, \\
&= \|\nabla f(x_i^\star(x_i^\star(y), x_{-i}), x_{-i}) - \nabla f(x_i^\star(y), y_{-i})\|, \\
&\leq \|\nabla f(x_i^\star(x_i^\star(y), x_{-i}), x_{-i}) - \nabla f(x_i^\star(x_i^\star(y), x_{-i}), y_{-i})\|, \\
&+ \|\nabla f(x_i^\star(x_i^\star(y), x_{-i}), y_{-i}) - \nabla f(x_i^\star(y), y_{-i})\|, \\
&\leq L\|x_{-i} - y_{-i}\| + L\|x_i^\star(y) - x_i^\star(x_i^\star(y), x_{-i})\|, \\
&\leq \left(L + \frac{L^2}{\mu}\right)\|x_{-i} - y_{-i}\|.
\end{aligned}$$

The first equality is due to Lemma A.5 in Nouiehed et al. (2019). This leads to

$$\begin{aligned}
\|\nabla G_f(x) - \nabla G_f(y)\| &= \|\nabla \frac{1}{n}\sum_{i=1}^n g_i(x_{-i}) - \nabla \frac{1}{n}\sum_{i=1}^n g_i(y_{-i})\| \\
&\leq \frac{1}{n}\sum_{i=1}^n \|\nabla g_i(x_{-i}) - \nabla g_i(y_{-i})\| \\
&\leq \frac{1}{n}\sum_{i=1}^n \left(L + \frac{L^2}{\mu}\right)\|x_{-i} - y_{-i}\| \leq \left(L + \frac{L^2}{\mu}\right)\|x - y\|.
\end{aligned}$$

$\square$

## C.5   PROOF OF THEOREM 3.6

From the $n$-sided PL condition and by noticing that $L$-smoothness indicates the $L$-coordinate-wise smoothness, for $\alpha \leq \frac{1}{L}$, we get

$$f(x^{t+1}) - f(x^t) \leq \langle \nabla_{i^t} f(x^t), x_{i^t}^{t+1} - x_{i^t}^t \rangle + \frac{L}{2} \|x_{i^t}^{t+1} - x_{i^t}^t\|^2,$$

$$= -(\alpha - \frac{L^2\alpha}{2})\|\nabla_i f(x^t)\|^2,$$

$$\leq -\frac{\alpha}{2}\|\nabla_i f(x^t)\|^2,$$

$$\leq -\mu\alpha(f(x^t) - \min_{y_{i^t}} f(y_{i^t}, x_{-i^t}^t)).$$

$$\implies f(x^{t+1}) - \min_{y_{i^t}} f(y_{i^t}, x_{-i^t}^t) \leq (1 - \mu\alpha)(f(x^t) - \min_{y_{i^t}} f(y_{i^t}, x_{-i^t}^t)).$$

By taking the conditional expectation over $i^t$, we get

$$\mathbb{E}[f(x^{t+1}) - \min_{y_{i^t}} f(y_{i^t}, x_{-i^t}^t)|x^t] \leq (1 - \mu\alpha)\mathbb{E}[f(x^t) - \min_{y_{i^t}} f(y_{i^t}, x_{-i^t}^t)|x^t].$$

Then by rearranging terms, we have,

$$\mathbb{E}[f(x^{t+1}) - \min_{y_{i^{t+1}}} f(y_{i^{t+1}}, x_{-i^{t+1}}^{t+1})|x^t]$$

$$\leq (1 - \mu\alpha)\mathbb{E}[f(x^t) - \min_{y_{i^t}} f(y_{i^t}, x_{-i^t}^t)|x^t] + \mathbb{E}[\min_{y_{i^t}} f(y_{i^t}, x_{-i^t}^t) - \min_{y_{i^{t+1}}} f(y_{i^{t+1}}, x_{-i^{t+1}}^{t+1})|x^t].$$

This is equivalent to say

$$\mathbb{E}[f(x^{t+1}) - G_f(x^{t+1})|x^t] \leq (1 - \mu\alpha)(f(x^t) - G_f(x^t)) + \mathbb{E}[G_f(x^t) - G_f(x^{t+1})|x^t].$$

From lemma 3.4, we know $G_f(x)$ has $L' = L + \frac{L^2}{\mu}$-Lipschitz gradient.

$$\mathbb{E}[G_f(x^t) - G_f(x^{t+1})|x^t] \leq \mathbb{E}[-\langle \nabla_{i^t} G_f(x^t), x_{i^t}^{t+1} - x_{i^t}^t \rangle + \frac{L'}{2}\|x_{i^t}^{t+1} - x_{i^t}^t\|^2|x^t]$$

$$= \mathbb{E}[\alpha\langle \nabla_{i^t} G_f(x^t), \nabla_{i^t} f(x^t) \rangle + \frac{\alpha^2 L'}{2}\|\nabla_{i^t} f(x^t)\|^2|x^t]$$

$$= \frac{1}{n}\Big(\alpha\langle \nabla G_f(x^t), \nabla f(x^t) \rangle + \frac{\alpha^2 L'}{2}\|\nabla f(x^t)\|^2\Big).$$

And

$$\mathbb{E}[f(x^t) - f(x^{t+1})] \geq \mathbb{E}[-\langle \nabla_{i^t} f(x^t), x_{i^t}^{t+1} - x_{i^t}^t \rangle - \frac{L}{2}\|x_{i^t}^{t+1} - x_{i^t}^t\|^2|x^t]$$

$$= \mathbb{E}[\alpha\|\nabla_{i^t} f(x^t)\|^2 - \frac{\alpha^2 L}{2}\|\nabla_{i^t} f(x^t)\|^2|x^t]$$

$$= \frac{1}{n}\Big(\alpha\|\nabla f(x^t)\|^2 - \frac{\alpha^2 L}{2}\|\nabla f(x^t)\|^2\Big).$$

If $\langle \nabla G_f(x^t), \nabla f(x^t) \rangle \leq \kappa\|\nabla f(x^t)\|^2$, then by choosing $\alpha \leq \frac{2(1-\kappa)}{2L'+(1+\kappa)L}$, we have

$$\mathbb{E}[G_f(x^t) - G_f(x^{t+1})|x^t] \leq \frac{1}{n}(\alpha\langle \nabla G_f(x^t), \nabla f(x^t) \rangle + \frac{\alpha^2 L'}{2}\|\nabla f(x^t)\|^2)$$

$$\leq \frac{1+\kappa}{2n}\Big(\alpha\|\nabla f(x^t)\|^2 - \frac{\alpha^2 L}{2}\|\nabla f(x^t)\|^2\Big)$$

$$\leq \frac{1+\kappa}{2}\mathbb{E}[f(x^t) - f(x^{t+1})|x^t] = \tilde{\kappa}\mathbb{E}[f(x^t) - f(x^{t+1})|x^t],$$

where $\tilde{\kappa} = \frac{1+\kappa}{2}$. As a result,

$$\mathbb{E}[f(x^{t+1}) - G_f(x^{t+1})|x^t] \leq (1 - \mu\alpha)(f(x^t) - G_f(x^t)) + \tilde{\kappa}\mathbb{E}[f(x^t) - f(x^{t+1})|x^t].$$

To write it differently,

$$(1 + \tilde{\kappa})\mathbb{E}[f(x^{t+1}) - G_f(x^{t+1})|x^t]$$
$$\leq (1 - \mu\alpha)(f(x^t) - G_f(x^t)) + \tilde{\kappa}\mathbb{E}[G_f(x^t) - G_f(x^{t+1})|x^t] + \tilde{\kappa}\mathbb{E}[f(x^t) - G_f(x^t)|x^t]$$
$$= (1 - \mu\alpha + \tilde{\kappa})(f(x^t) - G_f(x^t)) + \tilde{\kappa}\mathbb{E}[G_f(x^t) - G_f(x^{t+1})|x^t]$$
$$\leq (1 - \mu\alpha + \tilde{\kappa})(f(x^t) - G_f(x^t)) + \tilde{\kappa}^2\mathbb{E}[f(x^t) - f(x^{t+1})|x^t].$$

By iterating over this process,

$$\frac{1}{1 - \tilde{\kappa}}\mathbb{E}[f(x^{t+1}) - G_f(x^{t+1})|x^t] \leq \left(\frac{1}{1 - \tilde{\kappa}} - \mu\alpha\right)(f(x^t) - G_f(x^t)),$$
$$\implies \mathbb{E}[f(x^{t+1}) - G_f(x^{t+1})|x^t] \leq (1 - \mu\alpha(1 - \tilde{\kappa}))(f(x^t) - G_f(x^t)),$$
$$= \left(1 - \frac{\mu\alpha(1 - \kappa)}{2}\right)(f(x^t) - G_f(x^t)).$$

$\square$

### C.6   PROOF OF THEOREM 3.7

From the PL condition, the smoothness assumption and $\alpha \leq 1/L$, we get

$$f(x^{t+1}) \leq f(x^t) - \frac{\alpha}{2}\|\nabla f(x^t)\|^2$$
$$\leq f(x^t) - n\mu\alpha(f(x^t) - G_f(x^t)).$$
$$\implies f(x^{t+1}) - G_f(x^t) \leq (1 - n\mu\alpha)(f(x^t) - G_f(x^t)).$$

This is equivalent to say

$$f(x^{t+1}) - G_f(x^{t+1}) \leq (1 - n\mu\alpha)(f(x^t) - G_f(x^t)) + G_f(x^t) - G_f(x^{t+1}).$$

From lemma 3.4, we know $G_f(x)$ has $L' = L + \frac{L^2}{\mu}$-Lipschitz gradient.

$$G_f(x^t) - G_f(x^{t+1}) \leq -\langle\nabla G_f(x^t), x^{t+1} - x^t\rangle + \frac{L'}{2}\|x^{t+1} - x^t\|^2,$$
$$= \alpha\langle\nabla G_f(x^t), \nabla f(x^t)\rangle + \frac{\alpha^2 L'}{2}\|\nabla f(x^t)\|^2.$$

And

$$f(x^t) - f(x^{t+1}) \geq -\langle\nabla f(x^t), x^{t+1} - x^t\rangle - \frac{L}{2}\|x^{t+1} - x^t\|^2$$
$$= \alpha\|\nabla f(x^t)\|^2 - \frac{\alpha^2 L}{2}\|\nabla f(x^t)\|^2$$

If $\langle\nabla G_f(x^t), \nabla f(x^t)\rangle \leq \kappa\|\nabla f(x^t)\|^2$, then by choosing $\alpha \leq \frac{2(1-\kappa)}{2L'+(1+\kappa)L}$, we have

$$G_f(x^t) - G_f(x^{t+1}) \leq \alpha\langle\nabla G_f(x^t), \nabla f(x^t)\rangle + \frac{\alpha^2 L'}{2}\|\nabla f(x^t)\|^2,$$
$$\leq \alpha\kappa\|\nabla f(x^t)\|^2 + \frac{\alpha^2 L'}{2}\|\nabla f(x^t)\|^2,$$
$$\leq \frac{1 + \kappa}{2}\left(\alpha\|\nabla f(x^t)\|^2 - \frac{\alpha^2 L}{2}\|\nabla f(x^t)\|^2\right),$$
$$= \tilde{\kappa}(f(x^t) - f(x^{t+1}))$$

where $\tilde{\kappa} = \frac{1+\kappa}{2}$. As a result,

$$f(x^{t+1}) - G_f(x^{t+1}) \leq (1 - n\mu\alpha)(f(x^t) - G_f(x^t)) + \tilde{\kappa}(f(x^t) - f(x^{t+1}))$$

To write it differently,

$$(1 + \tilde{\kappa})(f(x^{t+1}) - G_f(x^{t+1})) \leq (1 - n\mu\alpha + \tilde{\kappa})(f(x^t) - G_f(x^t)) + \tilde{\kappa}(G_f(x^t) - G_f(x^{t+1}))$$
$$\leq (1 - n\mu\alpha + \tilde{\kappa})(f(x^t) - G_f(x^t)) + \tilde{\kappa}^2(f(x^t) - f(x^{t+1}))$$

By iterating over this process,

$$\frac{1}{1-\tilde{\kappa}}(f(x^{t+1}) - G_f(x^{t+1})) \le \left(\frac{1}{1-\tilde{\kappa}} - n\mu\alpha\right)(f(x^t) - G_f(x^t)),$$

$$\implies f(x^{t+1}) - G_f(x^{t+1}) \le (1 - n\mu\alpha(1-\tilde{\kappa}))(f(x^t) - G_f(x^t)),$$

$$f(x^{t+1}) - G_f(x^{t+1}) \le \left(1 - \frac{n\mu\alpha(1-\kappa)}{2}\right)(f(x^t) - G_f(x^t)).$$

$\square$

### C.7 PROOF OF LEMMA 3.8

We have

$$
\begin{aligned}
\|\nabla G_f(x)\| &= \left\| \frac{1}{n} \sum_{i=1}^{n} \nabla f(x_i^\star(x), x_{-i}) \right\| \\
&\le \left\| \frac{1}{n} \sum_{i=1}^{n} (\nabla f(x_i^\star(x), x_{-i}) - \nabla f(x)) \right\| + \|\nabla f(x)\| \\
&\le \frac{1}{n} \sum_{i=1}^{n} \|\nabla f(x_i^\star(x), x_{-i}) - \nabla f(x)\| + \|\nabla f(x)\| \\
&\le \frac{L}{n} \sum_{i=1}^{n} \|x_i^\star(x) - x_i\| + \|\nabla f(x)\| \\
&\le \frac{L}{\sqrt{n}} \sqrt{\sum_{i=1}^{n} \|x_i^\star(x) - x_i\|^2} + \|\nabla f(x)\| \\
&\le \frac{L}{\mu\sqrt{n}} \sqrt{\sum_{i=1}^{n} \|\nabla_i f(x)\|^2} + \|\nabla f(x)\| = \left(\frac{L}{\mu\sqrt{n}} + 1\right)\|\nabla f(x)\|.
\end{aligned}
$$

The fifth line comes from Cauchy-Schwartz inequality and the sixth line comes from the error bound property. $\square$

### C.8 PROOF OF THEOREM 3.10

**Case 1:** This is analogous to the proof of Theorem 3.6.

**Case 2:** From the smoothness of the function, we get

$$
\begin{aligned}
f(x^{t+1}) \le & f(x^t) + \langle \nabla_{i^t} f(x^t), x_{i^t}^{t+1} - x_{i^t}^t \rangle + \frac{L}{2}\|x_{i^t}^{t+1} - x_{i^t}^t\|^2 \\
= & f(x^t) - \alpha\langle \nabla_{i^t} f(x^t), \nabla_{i^t} f(x^t) + k^t \nabla_{i^t} G_f(x^t) \rangle + \frac{L\alpha^2}{2}\|\nabla_{i^t} f(x^t) + k^t \nabla_{i^t} G_f(x^t)\|^2 \\
= & f(x^t) - (\alpha - \frac{L\alpha^2}{2})\|\nabla_{i^t} f(x^t)\|^2 - (\alpha k^t - L\alpha^2 k^t)\langle \nabla_{i^t} f(x^t), \nabla_{i^t} G_f(x^t)\rangle \\
& + \frac{L\alpha^2(k^t)^2}{2}\|\nabla_{i^t} G_f(x^t)\|^2.
\end{aligned}
$$

Taking the expectation over $i^t$, we have

$$
\begin{aligned}
\mathbb{E}[f(x^{t+1}) - G_f(x^{t+1})|x^t] \le & f(x^t) - G_f(x^t) - \frac{1}{n}(\alpha - \frac{L\alpha^2}{2})\|\nabla f(x^t)\|^2 \\
& - \frac{1}{n}(\alpha k^t - L\alpha^2 k^t)\langle \nabla f(x^t), \nabla G_f(x^t)\rangle \\
& + \frac{L\alpha^2(k^t)^2}{2n}\|\nabla G_f(x^t)\|^2 + \mathbb{E}[G_f(x^t) - G_f(x^{t+1})].
\end{aligned}
$$

For $G_f(x)$, we have

$$
\begin{aligned}
G_f(x^t) \leq & G_f(x^{t+1}) - \langle \nabla_{i^t} G_f(x^t), x^{t+1}_{i^t} - x^t_{i^t} \rangle + \frac{L'}{2} \| x^{t+1}_{i^t} - x^t_{i^t} \|^2 \\
= & G_f(x^{t+1}) + \alpha \langle \nabla_{i^t} G_f(x^t), \nabla_{i^t} f(x^t) + k^t \nabla_{i^t} G_f(x^t) \rangle \\
& + \frac{L'\alpha^2}{2} \| \nabla_{i^t} f(x^t) + k^t \nabla_{i^t} G_f(x^t) \|^2 \\
= & G_f(x^{t+1}) + \alpha(k^t) \| \nabla_{i^t} G_f(x^t) \|^2 + (\alpha + L'\alpha^2 k^t) \langle \nabla_{i^t} G_f(x^t), \nabla_{i^t} f(x^t) \rangle \\
& + \frac{L'\alpha^2}{2} \| \nabla_{i^t} f(x^t) \|^2 + \frac{L'\alpha^2 (k^t)^2}{2} \| \nabla_{i^t} G_f(x^t) \|^2.
\end{aligned}
$$

Taking the expectation over $i^t$ yields

$$
\begin{aligned}
\mathbb{E}[G_f(x^t) - G_f(x^{t+1})|x^t] \leq & \frac{\alpha k^t}{n} \| \nabla G_f(x^t) \|^2 + \frac{\alpha + L'\alpha^2 k^t}{n} \langle \nabla G_f(x^t), \nabla f(x^t) \rangle \\
& + \frac{L'\alpha^2}{2n} \| \nabla f(x^t) \|^2 + \frac{L'\alpha^2 (k^t)^2}{2n} \| \nabla G_f(x^t) \|^2.
\end{aligned}
$$

As a result, we get

$$
\begin{aligned}
\mathbb{E}[f(x^{t+1}) - G_f(x^{t+1})|x^t] \leq & f(x^t) - G_f(x^t) - \frac{1}{n} (\alpha - \frac{L\alpha^2}{2} - \frac{L'\alpha^2}{2}) \| \nabla f(x^t) \|^2 \\
& - \frac{1}{n} (\alpha k^t - L\alpha^2 k^t - \alpha - L'\alpha^2 k^t) \langle \nabla f(x^t), \nabla G_f(x^t) \rangle \quad (11) \\
& + \frac{1}{2n} ((L' + L)\alpha^2 (k^t)^2 + 2\alpha k^t) \| \nabla G_f(x^t) \|^2.
\end{aligned}
$$

Now, we define

$$
\begin{aligned}
h(k^t) := & -\frac{1}{n} (\alpha k^t - L\alpha^2 k^t - \alpha - L'\alpha^2 k^t) \langle \nabla f(x^t), \nabla G_f(x^t) \rangle \\
& + \frac{1}{2n} ((L' + L)\alpha^2 (k^t)^2 + 2\alpha k^t) \| \nabla G_f(x^t) \|^2,
\end{aligned}
$$

which is a convex function. Therefore, we have

$$
\begin{aligned}
h(-1) = & -\frac{2\alpha - (L+L')\alpha^2}{2n} \| \nabla f(x^t) - \nabla G_f(x^t) \|^2 + \frac{1}{n} \left( \alpha - \frac{L\alpha^2}{2} - \frac{L'\alpha^2}{2} \right) \| \nabla f(x^t) \|^2 \\
\leq & \frac{1}{n} \left( \alpha - \frac{L\alpha^2}{2} - \frac{L'\alpha^2}{2} \right) \| \nabla f(x^t) \|^2.
\end{aligned}
$$

The function value $h(k^t)$ at minimizer $k^t = k^\star = -\frac{((L+L')\alpha - 1)\langle \nabla f, \nabla G_f \rangle + \|\nabla G_f\|^2}{(L+L')\alpha \|\nabla G_f\|^2}$ is less or equals to zero if

$$
(L + L')^2 \langle \nabla f, \nabla G_f \rangle^2 \alpha^2 - 2(L + L') \langle \nabla f, \nabla G_f \rangle^2 \alpha + (\| \nabla G_f \|^2 - \langle \nabla f, \nabla G_f \rangle)^2 \geq 0.
$$

which is satisfied if

$$
\alpha \leq \frac{1}{2(L + L')} \frac{(\| \nabla G_f \|^2 - \langle \nabla f, \nabla G_f \rangle)^2}{\langle \nabla f, \nabla G_f \rangle^2}. \quad (12)
$$

Since in this case $\frac{(\| \nabla G_f \|^2 - \langle \nabla f, \nabla G_f \rangle)^2}{\langle \nabla f, \nabla G_f \rangle^2} \geq C$, eq. (12) is satisfied if

$$
\alpha \leq \frac{C}{2(L + L')}.
$$

In consequence, if $\alpha \leq \frac{1}{2C(L+L')}$, $\forall \lambda \in [0, 1]$, we have

$$
h(-\lambda + (1 - \lambda)k^\star) \leq \lambda h(-1) + (1 - \lambda)h(k^\star) \leq \frac{\lambda}{n} \left( \alpha - \frac{L\alpha^2}{2} - \frac{L'\alpha^2}{2} \right) \| \nabla f(x^t) \|^2
$$

By setting $k^t = -1 + \frac{\langle \nabla f(x^t), \nabla G_f(x^t)\rangle - \|\nabla G_f(x^t)\|^2}{\|\nabla G_f(x^t)\|^2} = -\lambda + (1-\lambda)k^\star$, we have

$$0 \le \lambda = 1 - \frac{(L+L')\alpha(k^t+1)\|\nabla G_f\|^2}{(1-(L+L')\alpha)(\langle \nabla f, \nabla G_f\rangle - \|\nabla G_f\|^2)} = 1 - \frac{(L+L')\alpha}{1-(L+L')\alpha} < 1.$$

and

$$h(k^t) = h(-\lambda + (1-\lambda)k^\star) \le \frac{1}{n}\Big(1 - \frac{(L+L')\alpha}{1-(L+L')\alpha}\Big)\Big(\alpha - \frac{L\alpha^2}{2} - \frac{L'\alpha^2}{2}\Big)\|\nabla f(x^t)\|^2.$$

As a result,

$$\mathbb{E}[f(x^{t+1}) - G_f(x^{t+1})|x^t]$$

$$\le f(x^t) - G_f(x^t) - \frac{1}{n}\Big(\alpha - \frac{L\alpha^2}{2} - \frac{L'\alpha^2}{2}\Big)\|\nabla f(x^t)\|^2 + h(k^t)$$

$$\le f(x^t) - G_f(x^t) - \frac{1}{n}\frac{(L+L')\alpha}{1-(L+L')\alpha}\Big(\alpha - \frac{L\alpha^2}{2} - \frac{L'\alpha^2}{2}\Big)\|\nabla f(x^t)\|^2$$

$$\le f(x^t) - G_f(x^t) - \frac{1}{2n}\frac{(L+L')\alpha^2}{1-(L+L')\alpha}\|\nabla f(x^t)\|^2$$

$$\le \Big(1 - \frac{(L+L')\mu\alpha^2}{1-(L+L')\alpha}\Big)(f(x^t) - G_f(x^t))$$

$$\le \Big(1 - \frac{(L+L')\mu\alpha^2}{2}\Big)(f(x^t) - G_f(x^t)).$$

**Case 3:** In this case, notice that $f - G_f$ is $L + L'$-smooth,

$$\mathbb{E}[f(x^{t+1}) - G_f(x^{t+1})|x^t],$$

$$\le f(x^t) - G_f(x^t) + \mathbb{E}[\langle \nabla_{i^t}f(x^t) - \nabla_{i^t}G(x^t), x^{t+1}_{i^t} - x^t_{i^t}\rangle + \frac{L+L'}{2}\|x^{t+1}_{i^t} - x^t_{i^t}\|^2],$$

$$= f(x^t) - G_f(x^t) - (\alpha - \frac{L\alpha^2}{2})\mathbb{E}[\|\nabla_{i^t}f(x^t) - \nabla_{i^t}G(x^t)\|^2],$$

$$\le f(x^t) - G_f(x^t) - \frac{1}{2}\alpha\mathbb{E}[\|\nabla_{i^t}f(x^t) - \nabla_{i^t}G(x^t)\|^2],$$

$$\le f(x^t) - G_f(x^t) - \frac{1}{2n}\alpha\|\nabla f(x^t) - \nabla G(x^t)\|^2,$$

$$\le f(x^t) - G_f(x^t) - \frac{\alpha\nu}{n}(f(x^t) - G_f(x^t))^{\frac{2}{\theta}}.$$

From the Lemma 6 of Fatkhullin et al. (2022), we have

$$\mathbb{E}[f(x^{t+k}) - G_f(x^{t+k})|x^t] \le \frac{(2n)^{\frac{\theta}{2-\theta}}\frac{2-\theta}{\theta}^{-\frac{\theta+2}{2-\theta}} + n^{\frac{\theta}{2-\theta}}\theta^{-\frac{\theta}{2-\theta}} + (\nu\alpha)^{\frac{\theta}{2-\theta}}(f(x^t) - G_f(x^t))}{(\nu\alpha(k+1))^{\frac{\theta}{2-\theta}}}$$

$$\square$$

## C.9 Proof of Theorem 3.11

To approximate $G_f(x^t)$, we need to estimate the best response of i-th block $x_i^\star(x^t)$ when other blocks are fixed. As the function $f(x^t)$ satisfies $n$-sided PL condition, the function $f_i(x_i) = f(x_i, x^t_{-i})$ satisfies strong PL condition. Therefore by applying the gradient descent with partial gradient $\nabla_i f(x_i, x^t_{-i})$, the best response can be approximated efficiently. For any $\delta > 0$,

$$\|x_i^\star(x^t) - y_i^{t,T'}\|^2 \le \frac{2}{\mu}(f(y_i^{t,T'}, x^t_{-i}) - \min_{x_i} f(x_i, x^t_{-i}))$$

$$\le \frac{2}{\mu}(1 - \mu\beta)^{T'}(f(x^t) - \min_{x_i} f(x_i, x^t_{-i})) \tag{13}$$

$$\le \frac{1}{\mu^2}(1 - \mu\beta)^{T'}\|\nabla_i f(x^t)\|^2 \le \frac{\delta^2}{nL^2}\|\nabla_i f(x^t)\|^2.$$

if $T' \geq \frac{1}{\log(\frac{1}{1-\mu\beta})} \log(\frac{nL^2}{\mu^2\delta^2})$ and $\beta \leq \frac{1}{L}$. The first inequality comes from the quadratic growth properties of the function $f_i(x_i) = f(x_i, x^t_{-i})$ since it satisfies the strong PL condition. The second inequality comes from the convergence of gradient descent under the PL condition. The third inequality comes from the definition of the n-sided PL condition.

$$
\begin{aligned}
\|\nabla G_f(x^t) - \tilde{\nabla} G_f(x^t)\| &= \left\| \sum_{i=1}^{n} \frac{1}{n} \nabla f(x^\star_i(x^t), x_{-i}) - \sum_{i=1}^{n} \frac{1}{n} \nabla f(y^{t,T'}_i, x_{-i}) \right\| \\
&\leq \frac{1}{n} \sum_{i=1}^{n} \left\| \nabla f(x^\star_i(x^t), x_{-i}) - \nabla f(y^{t,T'}_i, x_{-i}) \right\| \\
&\leq \frac{L}{n} \sum_{i=1}^{n} \left\| x^\star_i(x^t) - y^{t,T'}_i \right\| \\
&\leq \frac{\delta}{\sqrt{n}} \sum_{i=1}^{n} \|\nabla_i f(x^t)\| \leq \delta \|\nabla f(x^t)\|.
\end{aligned}
\tag{14}
$$

In the fourth line, we apply the eq. (13). In the last line, we apply Cauchy-Schwartz inequality.

The second line comes from triangle inequality and the third line comes from the $L$-Lipschitz continuity of $\nabla f(x^t)$. Then, we denotes $\bar{x}^{t+1}$ as the iterates in the ideal case, i.e.

$$
\bar{x}^{t+1}_i = \begin{cases} x^t_i - \alpha(\nabla_i f(x^t) + k^t \nabla_i G(x^t)), & \text{if } i = i^t, \\ x^{t+1}_i, & \text{if } i \neq i^t. \end{cases}
\tag{15}
$$

Next, by choosing $\delta = \gamma \frac{\alpha^3}{13}$ we show the convergence of $f(x^t) - G_f(x^t)$. To do so, we break it into different cases.

**Case 1:** If $\langle \tilde{\nabla} G_f(x^t), \nabla f(x^t) \rangle \leq (\gamma - \gamma \frac{\alpha^3}{13}) \|\nabla f(x^t)\|^2$, we have

$$
\begin{aligned}
&\langle \nabla G_f(x^t), \nabla f(x^t) \rangle \\
&= \langle \nabla G_f(x^t) - \tilde{\nabla} G_f(x^t), \nabla f(x^t) \rangle + \langle \tilde{\nabla} G_f(x^t), \nabla f(x^t) \rangle \\
&\leq \|\nabla G_f(x^t) - \tilde{\nabla} G_f(x^t)\| \|\nabla f(x^t)\| + \langle \tilde{\nabla} G_f(x^t), \nabla f(x^t) \rangle \\
&\leq \gamma \frac{\alpha^3}{13} \|\nabla f(x^t)\|^2 + \langle \tilde{\nabla} G_f(x^t), \nabla f(x^t) \rangle \leq \gamma \|\nabla f(x^t)\|^2.
\end{aligned}
$$

By choosing $k^t = 0$, from theorem 3.6, we have

$$
\begin{aligned}
\mathbb{E}[f(x^{t+1}) - G_f(x^{t+1})|x^t] &= \mathbb{E}[f(\bar{x}^{t+1}) - G_f(\bar{x}^{t+1})|x^t] \\
&\leq \left(1 - \frac{\mu\alpha(1-\gamma)}{2}\right)(f(x^t) - G_f(x^t)).
\end{aligned}
$$

**Case 2:** $\left(\frac{\|\tilde{\nabla} G_f(x^t)\|^2}{\langle \nabla f(x^t), \tilde{\nabla} G_f(x^t) \rangle} - 1\right)^2 \geq C$ and $\langle \tilde{\nabla} G_f(x^t), \nabla f(x^t) \rangle \geq \left(\gamma - \gamma \frac{\alpha^3}{13}\right) \|\nabla f(x^t)\|^2$. We firstly bound the difference of $\nabla G_f(x^t)$ and $\tilde{\nabla} G_f(x^t)$. From the assumption of case 2, we have

$$
\langle \tilde{\nabla} G_f(x^t), \nabla f(x^t) \rangle \geq \left(\gamma - \gamma \frac{\alpha^3}{13}\right) \|\nabla f(x^t)\|^2, \implies \|\tilde{\nabla} G_f(x^t)\| \geq \left(\gamma - \gamma \frac{\alpha^3}{13}\right) \|\nabla f(x^t)\|.
$$

This indicates

$$
\begin{aligned}
\left| \|\nabla G_f(x^t)\| - \|\tilde{\nabla} G_f(x^t)\| \right| &\leq \|\nabla G_f(x^t) - \tilde{\nabla} G_f(x^t)\| \leq \delta \|\nabla f(x^t)\| \\
&\leq \frac{\delta}{\gamma - \gamma \frac{\alpha^3}{13}} \|\tilde{\nabla} G_f(x^t)\| \leq \frac{1}{2} \|\tilde{\nabla} G_f(x^t)\|.
\end{aligned}
$$

In the last line, we apply $\alpha \leq (C_f)^{-1/2} < 1$ and $\delta = \frac{\gamma\alpha^3}{13} \leq \frac{\gamma - \gamma \frac{\alpha^3}{13}}{2}$. As a result,

$$
\left| \frac{\|\tilde{\nabla} G_f(x^t)\|}{\|\nabla G_f(x^t)\|} - 1 \right| \leq \frac{\delta}{\gamma - \gamma \frac{\alpha^3}{13}} \cdot \frac{\|\tilde{\nabla} G_f(x^t)\|}{\|\nabla G_f(x^t)\|},
$$

and $\frac{\|\tilde{\nabla}G_f(x^t)\|}{\|\nabla G_f(x^t)\|} \leq 2$. These two inequalities imply

$$
\begin{aligned}
\Big| \frac{\|\tilde{\nabla}G_f(x^t)\|^2}{\|\nabla G_f(x^t)\|^2} - 1 \Big| &= \Big( \frac{\|\tilde{\nabla}G_f(x^t)\|}{\|\nabla G_f(x^t)\|} + 1 \Big) \Big| \frac{\|\tilde{\nabla}G_f(x^t)\|}{\|\nabla G_f(x^t)\|} - 1 \Big| \\
&\leq \Big( \frac{\|\tilde{\nabla}G_f(x^t)\|}{\|\nabla G_f(x^t)\|} + 1 \Big) \frac{\delta}{\gamma - \gamma\frac{\alpha^3}{13}} \frac{\|\tilde{\nabla}G_f(x^t)\|}{\|\nabla G_f(x^t)\|} \leq \frac{6\delta}{\gamma - \gamma\frac{\alpha^3}{13}} \leq \frac{12\delta}{\gamma}.
\end{aligned}
\tag{16}
$$

In the last inequality, we applied $\alpha \leq (C_f)^{-1/2} < 1$. Then we can bound the difference between $k^t$ and $\tilde{k}^t$.

$$
\begin{aligned}
|k^t - \tilde{k}^t| &= \Big| \frac{\langle \nabla f(x^t), \nabla G_f(x^t) \rangle}{\|\nabla G_f(x^t)\|^2} - \frac{\langle \nabla f(x^t), \tilde{\nabla}G_f(x^t) \rangle}{\|\tilde{\nabla}G_f(x^t)\|^2} \Big| \\
&\leq \Big| \frac{\langle \nabla f(x^t), \nabla G_f(x^t) \rangle}{\|\nabla G_f(x^t)\|^2} - \frac{\langle \nabla f(x^t), \nabla G_f(x^t) \rangle}{\|\tilde{\nabla}G_f(x^t)\|^2} \Big| \\
&\quad + \Big| \frac{\langle \nabla f(x^t), \nabla G_f(x^t) \rangle}{\|\tilde{\nabla}G_f(x^t)\|^2} - \frac{\langle \nabla f(x^t), \tilde{\nabla}G_f(x^t) \rangle}{\|\tilde{\nabla}G_f(x^t)\|^2} \Big| \\
&\leq \|\nabla f(x^t)\| \|\nabla G_f(x^t)\| \Big| \frac{1}{\|\tilde{\nabla}G_f(x^t)\|^2} - \frac{1}{\|\nabla G_f(x^t)\|^2} \Big| \\
&\quad + \|\nabla f(x^t)\| \|\nabla G_f(x^t) - \tilde{\nabla}G_f(x^t)\| \frac{1}{\|\tilde{\nabla}G_f(x^t)\|^2} \\
&= \|\nabla f(x^t)\| \|\nabla G_f(x^t)\| \frac{1}{\|\tilde{\nabla}G_f(x^t)\|^2} \Big| \frac{\|\tilde{\nabla}G_f(x^t)\|^2}{\|\nabla G_f(x^t)\|^2} - 1 \Big| \\
&\quad + \|\nabla f(x^t)\| \|\nabla G_f(x^t) - \tilde{\nabla}G_f(x^t)\| \frac{1}{\|\tilde{\nabla}G_f(x^t)\|^2}, \\
&\leq \frac{12\delta}{\gamma} \|\nabla f(x^t)\| \|\nabla G_f(x^t)\| \frac{1}{\|\tilde{\nabla}G_f(x^t)\|^2} \\
&\quad + \|\nabla f(x^t)\| \|\nabla G_f(x^t) - \tilde{\nabla}G_f(x^t)\| \frac{1}{\|\tilde{\nabla}G_f(x^t)\|^2} \\
&\leq \frac{12\delta C_f}{\gamma} \frac{\|\nabla f(x^t)\|^2}{\|\tilde{\nabla}G_f(x^t)\|^2} + \|\nabla f(x^t)\| \|\nabla G_f(x^t) - \tilde{\nabla}G_f(x^t)\| \frac{1}{\|\tilde{\nabla}G_f(x^t)\|^2} \\
&\leq \Big( \frac{12\delta C_f}{\gamma} + \delta \Big) \frac{\|\nabla f(x)\|^2}{\|\tilde{\nabla}G_f(x^t)\|^2} \leq \frac{12\delta C_f}{\gamma\alpha} + \frac{\delta}{\alpha} \leq \frac{13\delta C_f}{\gamma\alpha} \leq C_f \alpha^2 \leq 1.
\end{aligned}
\tag{17}
$$

where $C_f = \frac{L}{\sqrt{n}\mu} + 1$. The fourth line comes from Cauchy-Schwartz inequality. The eighth line comes from eq. (16). The sixth line comes from lemma 3.8. The ninth line comes from eq. (14). The last line comes from $\delta = \frac{\gamma\alpha^3}{13}$ and $\alpha \leq (C_f)^{-1/2}$. Also, the absolute value of $k^t$ and $\tilde{k}^t$ can be bounded.

$$
|\tilde{k}^t| = \Big| -2 + \frac{\langle \nabla f(x^t), \tilde{\nabla}G_f(x^t) \rangle}{\|\tilde{\nabla}G_f(x^t)\|^2} \Big| \leq 2 + \frac{\|\nabla f(x^t)\|}{\|\tilde{\nabla}G_f(x^t)\|} \leq 2 + \Big( \gamma - \gamma\frac{\alpha^3}{13} \Big)^{-1} \leq 2 + \frac{13}{12\gamma}, \tag{18}
$$

and

$$
|k^t| = |k^t - \tilde{k}^t + \tilde{k}^t| \leq |k^t - \tilde{k}^t| + |\tilde{k}^t| \leq 3 + \frac{13}{12\gamma}. \tag{19}
$$

As a result,

$$
\begin{aligned}
\|k^t\nabla G_f(x^t) - \tilde{k}^t\tilde{\nabla}G_f(x^t)\| &= \|k^t\nabla G_f(x^t) - \tilde{k}^t\nabla G_f(x^t) + \tilde{k}^t\nabla G_f(x^t) - \tilde{k}^t\tilde{\nabla}G_f(x^t)\| \\
&\leq \|k^t\nabla G_f(x^t) - \tilde{k}^t\nabla G_f(x^t)\| + \|\tilde{k}^t\nabla G_f(x^t) - \tilde{k}^t\tilde{\nabla}G_f(x^t)\| \\
&\leq |k^t - \tilde{k}^t|\|\nabla G_f(x^t)\| + |\tilde{k}^t|\|\nabla G_f(x^t) - \tilde{\nabla}G_f(x^t)\| \\
&\leq C_f\alpha^2\|\nabla G_f(x^t)\| + \Big(2 + \frac{13}{12\gamma}\Big)\|\nabla G_f(x^t) - \tilde{\nabla}G_f(x^t)\|, \\
&\leq C_f^2\alpha^2\|\nabla f(x^t)\| + \Big(2 + \frac{13}{12\gamma}\Big)\delta\|\nabla f(x^t)\| \\
&\leq C_f^2\alpha^2\|\nabla f(x^t)\| + \frac{(2\gamma + \frac{13}{12})\alpha^3}{13}\|\nabla f(x^t)\| \\
&\leq C_f^2\alpha^2\|\nabla f(x^t)\| + \frac{37\alpha^3}{156}\|\nabla f(x^t)\| \leq 2C_f^2\alpha^2\|\nabla f(x^t)\|.
\end{aligned}
$$
(20)

The fourth line is from eq. (17) and eq. (18). The fifth and sixth lines come from eq. (14) and $\delta = \frac{\gamma\alpha^3}{13}$, respectively.

In the case of one of ideal settings, we need $\alpha$ to satisfy eq. (12). However, we only have the estimation $\tilde{\nabla}G_f(x^t)$. Next, we show that eq. (12) is satisfied if $\alpha$ is small enough. Then we can make sure the linear convergence of the ideal case and further bound the difference of $f - G_f$ between the ideal case and the practical case.

$$
\begin{aligned}
&\Big(\frac{\langle\nabla f(x^t), \nabla G_f(x^t)\rangle}{\|\nabla G_f(x^t)\|^2} - 1\Big)^2 \\
=&\Big(\frac{\langle\nabla f(x^t), \tilde{\nabla}G_f(x^t)\rangle}{\|\tilde{\nabla}G_f(x^t)\|^2} - 1 + \frac{\langle\nabla f(x^t), \nabla G_f(x^t)\rangle}{\|\nabla G_f(x^t)\|^2} - \frac{\langle\nabla f(x^t), \tilde{\nabla}G_f(x^t)\rangle}{\|\tilde{\nabla}G_f(x^t)\|^2}\Big)^2 \\
\geq&\Big(\frac{\langle\nabla f(x^t), \tilde{\nabla}G_f(x^t)\rangle}{\|\tilde{\nabla}G_f(x^t)\|^2} - 1\Big)^2 \\
&- 2\Big|\frac{\langle\nabla f(x^t), \tilde{\nabla}G_f(x^t)\rangle}{\|\tilde{\nabla}G_f(x^t)\|^2} - 1\Big| \cdot \Big|\frac{\langle\nabla f(x^t), \nabla G_f(x^t)\rangle}{\|\nabla G_f(x^t)\|^2} - \frac{\langle\nabla f(x^t), \tilde{\nabla}G_f(x^t)\rangle}{\|\tilde{\nabla}G_f(x^t)\|^2}\Big| \\
\geq&\Big(\frac{\langle\nabla f(x^t), \tilde{\nabla}G_f(x^t)\rangle}{\|\tilde{\nabla}G_f(x^t)\|^2} - 1\Big)^2 - 2C_f\alpha^2\Big|\frac{\langle\nabla f(x^t), \tilde{\nabla}G_f(x^t)\rangle}{\|\tilde{\nabla}G_f(x^t)\|^2} - 1\Big| \\
\geq&\Big(\frac{\langle\nabla f(x^t), \tilde{\nabla}G_f(x^t)\rangle}{\|\tilde{\nabla}G_f(x^t)\|^2} - 1\Big)^2 - 2C_f\alpha^2\Big(\frac{\|\nabla f(x^t)\|}{\|\tilde{\nabla}G_f(x^t)\|} + 1\Big) \\
\geq&\Big(\frac{\langle\nabla f(x^t), \tilde{\nabla}G_f(x^t)\rangle}{\|\tilde{\nabla}G_f(x^t)\|^2} - 1\Big)^2 - 2C_f\alpha^2\Big(\frac{13}{12\gamma} + 1\Big) \geq C - 2C_f\alpha^2\Big(\frac{13}{12\gamma} + 1\Big) \geq \frac{C}{2}.
\end{aligned}
$$

In the fifth line, we applies eq. (17). In the last line, we used $\alpha^2 \leq \frac{3C\gamma}{(13+12\gamma)C_f}$ and

$$
\|\tilde{\nabla}G_f(x^t)\| \geq \big(\gamma - \gamma\frac{\alpha^3}{13}\big)\|\nabla f(x^t)\| \geq \frac{12\gamma}{13}
$$

As a result, we obtain

$$
\begin{aligned}
\Big(\frac{\|\nabla G_f(x^t)\|^2}{\langle\nabla f(x^t), \nabla G_f(x^t)\rangle} - 1\Big)^2 &= \Big(\frac{\langle\nabla f(x^t), \nabla G_f(x^t)\rangle}{\|\nabla G_f(x^t)\|^2} - 1\Big)^2 \Big(\frac{\|\nabla G_f(x^t)\|^2}{\langle\nabla f(x^t), \nabla G_f(x^t)\rangle}\Big)^2 \\
&\geq \frac{C}{2}\Big(\frac{\|\nabla G_f(x^t)\|}{\|\nabla f(x^t)\|}\Big)^2 \geq \frac{C}{2}\frac{\|\tilde{\nabla}G_f(x^t)\|^2 - 2\|\nabla G_f(x^t) - \tilde{\nabla}G_f(x^t)\|^2}{2\|\nabla f(x^t)\|^2} \\
&\geq \frac{C}{2}\Big(\frac{72\gamma^2}{169} - \delta^2\Big) = \frac{C}{2}\Big(\frac{72\gamma^2}{169} - \frac{\gamma^2\alpha^6}{169}\Big) \\
&\geq \frac{71C\gamma^2}{338} \geq 2(L + L')\alpha.
\end{aligned}
$$

In the second line, we applied $\|x\|^2 \geq \frac{1}{2}\|y\|^2 - \|x-y\|^2$, $\forall x, y \in \mathbb{R}^d$. In the third line, we used the fact that $\|\tilde{\nabla}G_f(x^t)\| \geq \frac{\gamma}{2}\|\nabla f(x^t)\|$ and $\|\nabla G_f(x^t) - \tilde{\nabla}G_f(x^t)\| \leq \delta\|\nabla f(x^t)\|$ and applied $\delta = \frac{\gamma\alpha^3}{13}$. The last line comes from $\alpha \leq \frac{71C\gamma^2}{676(L+L')}$. Since eq. (12) is satisfied, it indicates $h(k^\star) \leq 0$. And we can apply the result from the ideal case. From lemma 3.4 and eq. (15), we have

$$\mathbb{E}[f(x^{t+1}) - G_f(x^{t+1})|x^t] - \mathbb{E}[f(\bar{x}^{t+1}) - G_f(\bar{x}^{t+1})|x^t]$$

$$\leq \mathbb{E}[\langle \nabla_{i^t} f(\bar{x}^{t+1}) - \nabla_{i^t} G_f(\bar{x}^{t+1}), x_{i^t}^{t+1} - \bar{x}_{i^t}^{t+1}\rangle + \frac{L+L'}{2}\|x_{i^t}^{t+1} - \bar{x}_{i^t}^{t+1}\|^2 |x^t]$$

$$= \mathbb{E}[\langle \nabla_{i^t} f(\bar{x}^{t+1}) - \nabla_{i^t} G_f(\bar{x}^{t+1}), \alpha(\tilde{k}^t \tilde{\nabla}_{i^t} G_f(x^t) - k^t \nabla_{i^t} G_f(x^t))\rangle |x^t]$$

$$\quad + \mathbb{E}[\frac{L+L'}{2}\|\alpha(\tilde{k}^t \tilde{\nabla}_{i^t} G_f(x^t) - k^t \nabla_{i^t} G_f(x^t))\|^2 |x^t]$$

$$= \mathbb{E}[\langle \nabla_{i^t} f(x^t) - \nabla_{i^t} G_f(x^t), \alpha(\tilde{k}^t \tilde{\nabla}_{i^t} G_f(x^t) - k^t \nabla_{i^t} G_f(x^t))\rangle |x^t]$$

$$\quad + \mathbb{E}[\langle \nabla_{i^t} f(\bar{x}^{t+1}) - \nabla_{i^t} f(x^t) - \nabla_{i^t} G_f(\bar{x}^{t+1}) + \nabla_{i^t} G_f(x^t), \alpha(\tilde{k}^t \tilde{\nabla}_{i^t} G_f(x^t) - k^t \nabla_{i^t} G_f(x^t))\rangle |x^t]$$

$$\quad + \mathbb{E}[\frac{L+L'}{2}\|\alpha(\tilde{k}^t \tilde{\nabla}_{i^t} G_f(x^t) - k^t \nabla_{i^t} G_f(x^t))\|^2 |x^t].$$

The first term is

$$\mathbb{E}[\langle \nabla_{i^t} f(x^t) - \nabla_{i^t} G_f(x^t), \alpha(\tilde{k}^t \tilde{\nabla}_{i^t} G_f(x^t) - k^t \nabla_{i^t} G_f(x^t))\rangle |x^t]$$

$$= \frac{1}{n}\langle \nabla f(x^t) - \nabla G_f(x^t), \alpha(\tilde{k}^t \tilde{\nabla} G_f(x^t) - k^t \nabla G_f(x^t))\rangle$$

$$\leq \frac{\alpha}{n}\|\nabla f(x^t) - \nabla G_f(x^t)\|\|k^t \nabla G_f(x^t) - \tilde{k}^t \tilde{\nabla} G_f(x^t)\|$$

$$\leq \frac{\alpha}{n}(\|\nabla f(x^t)\| + \|\nabla G_f(x^t)\|)2C_f^2\alpha^2\|\nabla f(x^t)\|$$

$$\leq \frac{1}{n}2C_f^2(1+C_f)\alpha^3\|\nabla f(x^t)\|^2.$$

In the fourth line, we apply the triangle inequality and the eq. (20). The second term is

$$\mathbb{E}[\langle \nabla_{i^t} f(\bar{x}^{t+1}) - \nabla_{i^t} f(x^t) - \nabla_{i^t} G_f(\bar{x}^{t+1}) + \nabla_{i^t} G_f(x^t), \alpha(\tilde{k}^t \tilde{\nabla}_{i^t} G_f(x^t) - k^t \nabla_{i^t} G_f(x^t))\rangle |x^t]$$

$$\leq \mathbb{E}[\|\nabla_{i^t} f(\bar{x}^{t+1}) - \nabla_{i^t} f(x^t) - \nabla_{i^t} G_f(\bar{x}^{t+1}) + \nabla_{i^t} G_f(x^t)\|\|\alpha(\tilde{k}^t \tilde{\nabla}_{i^t} G_f(x^t) - k^t \nabla_{i^t} G_f(x^t))\||x^t]$$

$$\leq \mathbb{E}[(L+L')\alpha\|\bar{x}_{i^t}^{t+1} - x_{i^t}^t\|\|\tilde{k}^t \tilde{\nabla}_{i^t} G_f(x^t) - k^t \nabla_{i^t} G_f(x^t)\||x^t]$$

$$\leq \mathbb{E}[(L+L')\alpha^2\|\nabla_{i^t} f(x^t) + k^t \nabla_{i^t} G_f(x^t)\|\|\tilde{k}^t \tilde{\nabla}_{i^t} G_f(x^t) - k^t \nabla_{i^t} G_f(x^t)\||x^t]$$

$$= \frac{1}{n}\sum_{i=1}^{n}[(L+L')\alpha^2\|\nabla_i f(x^t) + k^t \nabla_i G_f(x^t)\|\|\tilde{k}^t \tilde{\nabla}_i G_f(x^t) - k^t \nabla_i G_f(x^t)\|]$$

$$\leq \frac{1}{n}(L+L')\alpha^2\|\nabla f(x^t) + k^t \nabla G_f(x^t)\|\|\tilde{k}^t \tilde{\nabla} G_f(x^t) - k^t \nabla G_f(x^t)\|$$

$$\leq \frac{1}{n}(L+L')\alpha^2(\|\nabla f(x^t)\| + |k^t|\|\nabla G_f(x^t)\|)\|\tilde{k}^t \tilde{\nabla} G_f(x^t) - k^t \nabla G_f(x^t)\|$$

$$\leq \frac{1}{n}(L+L')\alpha^2\Big(1 + \Big(3 + \frac{13}{12\gamma}\Big)C_f\Big)\|\nabla f(x^t)\|\|\tilde{k}^t \tilde{\nabla} G_f(x^t) - k^t \nabla G_f(x^t)\|$$

$$\leq \frac{1}{n}2(L+L')C_f^2\alpha^4\Big(1 + \Big(3 + \frac{13}{12\gamma}\Big)C_f\Big)\|\nabla f(x^t)\|^2$$

$$\leq \frac{1}{n}C_f^2\alpha^3\Big(1 + \Big(3 + \frac{13}{12\gamma}\Big)C_f\Big)\|\nabla f(x^t)\|^2$$

$$\leq \frac{1}{n}C_f^2\Big(\frac{13}{12\gamma} + 4C_f\Big)\alpha^3\|\nabla f(x^t)\|^2.$$

In the sixth line, we apply Cauchy-Schwartz inequality. The eighth line comes from eq. (19). The ninth line comes from eq. (20). The third term is

$$\mathbb{E}[\frac{L+L'}{2}\|\alpha(\tilde{k}^t\tilde{\nabla}_{i^t}G_f(x^t) - k^t\nabla_{i^t}G_f(x^t))\|^2|x^t]$$

$$= \frac{1}{n}\frac{L+L'}{2}\|\alpha(\tilde{k}^t\tilde{\nabla}G_f(x^t) - k^t\nabla G_f(x^t))\|^2$$

$$\leq \frac{1}{n}\frac{L+L'}{2}(2C_f^2\alpha^3\|\nabla f(x^t)\|)^2$$

$$= \frac{1}{n}2(L+L')C_f^4\alpha^6\|\nabla f(x^t)\|^2 \leq \frac{1}{n}C_f^4\alpha^5\|\nabla f(x^t)\|^2.$$

In the third line, we apply eq. (20). In conclusion,

$$\mathbb{E}[f(x^{t+1}) - G_f(x^{t+1})|x^t] - \mathbb{E}[f(\bar{x}^{t+1}) - G_f(\bar{x}^{t+1})|x^t]$$

$$\leq \frac{1}{n}2C_f^2(1+C_f)\alpha^3\|\nabla f(x^t)\|^2 + \frac{1}{n}C_f^2\Big(\frac{13}{12\gamma} + 4C_f\Big)\alpha^3\|\nabla f(x^t)\|^2$$

$$+ \frac{1}{n}C_f^4\alpha^5\|\nabla f(x^t)\|^2 \tag{21}$$

$$\leq \frac{1}{n}\Big(\Big(2 + \frac{13}{12\gamma}\Big)C_f^2 + 6C_f^3 + C_f^4\Big)\alpha^3\|\nabla f(x^t)\|^2$$

$$\leq \frac{1}{n}\Big(9 + \frac{13}{12\gamma}\Big)C_f^4\alpha^3\|\nabla f(x^t)\|^2.$$

and

$$\mathbb{E}[f(x^{t+1}) - G_f(x^{t+1})|x^t]$$

$$= \mathbb{E}[f(\bar{x}^{t+1}) - G_f(\bar{x}^{t+1})|x^t] + \mathbb{E}[f(x^{t+1}) - G_f(x^{t+1})|x^t] - \mathbb{E}[f(\bar{x}^{t+1}) - G_f(\bar{x}^{t+1})|x^t]$$

$$\leq \Big(1 - \frac{(L+L')\mu\alpha^2}{2}\Big)(f(x^t) - G_f(x^t)) + \frac{1}{n}\Big(9 + \frac{13}{12\gamma}\Big)C_f^4\alpha^3\|\nabla f(x^t)\|^2$$

$$\leq \Big(1 - \frac{(L+L')\mu\alpha^2}{2}\Big)(f(x^t) - G_f(x^t)) + \Big(18 + \frac{13}{6\gamma}\Big)LC_f^4\alpha^3(f(x^t) - G_f(x^t))$$

$$\leq \Big(1 - \frac{(L+L')\mu\alpha^2}{4}\Big)(f(x^t) - G_f(x^t)).$$

In the second line, we apply theorem 3.10 and eq. (21). In the last line we apply $\alpha \leq \frac{3\gamma(L+L')\mu}{(13+108\gamma)LC_f^4}$.

**Case 3:** From eq. (11) and eq. (15) with $k^t = -1$, we know that

$$\mathbb{E}[f(\bar{x}^{t+1}) - G_f(\bar{x}^{t+1})|x^t] \leq f(x^t) - G_f(x^t) - \frac{1}{n}\Big(\alpha - \frac{L\alpha^2}{2} - \frac{L'\alpha^2}{2}\Big)\|\nabla f(x^t) - \nabla G_f(x^t)\|^2$$

$$\leq f(x^t) - G_f(x^t) - \frac{\alpha}{2n}\|\nabla f(x^t) - \nabla G_f(x^t)\|^2. \tag{22}$$

The second line comes from $\alpha \leq \frac{1}{L+L'}$. From lemma 3.4, we have

$$\mathbb{E}[f(x^{t+1}) - G_f(x^{t+1})|x^t] - \mathbb{E}[f(\bar{x}^{t+1}) - G_f(\bar{x}^{t+1})|x^t]$$

$$\leq \mathbb{E}[\langle\nabla_{i^t}f(\bar{x}^{t+1}) - \nabla_{i^t}G_f(\bar{x}^{t+1}), x_{i^t}^{t+1} - \bar{x}_{i^t}^{t+1}\rangle + \frac{L+L'}{2}\|x_{i^t}^{t+1} - \bar{x}_{i^t}^{t+1}\|^2|x^t]$$

$$= \mathbb{E}[\langle\nabla_{i^t}f(\bar{x}^{t+1}) - \nabla_{i^t}G_f(\bar{x}^{t+1}), \alpha(\tilde{\nabla}_{i^t}G_f(x^t) - \nabla_{i^t}G_f(x^t))\rangle|x^t]$$

$$+ \mathbb{E}[\frac{L+L'}{2}\|\alpha(\tilde{\nabla}_{i^t}G_f(x^t) - \nabla_{i^t}G_f(x^t))\|^2|x^t]$$

$$= \mathbb{E}[\langle\nabla_{i^t}f(x^t) - \nabla_{i^t}G_f(x^t), \alpha(\tilde{\nabla}_{i^t}G_f(x^t) - \nabla_{i^t}G_f(x^t))\rangle|x^t]$$

$$+ \mathbb{E}[\langle\nabla_{i^t}f(\bar{x}^{t+1}) - \nabla_{i^t}f(x^t) - \nabla_{i^t}G_f(\bar{x}^{t+1}) + \nabla_{i^t}G_f(x^t), \alpha(\tilde{\nabla}_{i^t}G_f(x^t) - \nabla_{i^t}G_f(x^t))\rangle|x^t]$$

$$+ \mathbb{E}[\frac{L+L'}{2}\|\alpha(\tilde{\nabla}_{i^t}G_f(x^t) - \nabla_{i^t}G_f(x^t))\|^2|x^t].$$

The first term is

$$\mathbb{E}[\langle \nabla_{i^t} f(x^t) - \nabla_{i^t} G_f(x^t), \alpha(\tilde{\nabla}_{i^t} G_f(x^t) - \nabla_{i^t} G_f(x^t)) \rangle | x^t]$$

$$= \frac{1}{n} \langle \nabla f(x^t) - \nabla G_f(x^t), \alpha(\tilde{\nabla} G_f(x^t) - \nabla G_f(x^t)) \rangle$$

$$\leq \frac{1}{n} \alpha \| \nabla f(x^t) - \nabla G_f(x^t) \| \| \tilde{\nabla} G_f(x^t) - \nabla G_f(x^t) \|$$

$$\leq \frac{1}{n} \alpha (\| \nabla f(x^t) \| + \| \nabla G_f(x^t) \|) \| \tilde{\nabla} G_f(x^t) - \nabla G_f(x^t) \|$$

$$\leq \frac{1}{n} (1 + C_f) \alpha \| \nabla f(x^t) \| \| \tilde{\nabla} G_f(x^t) - \nabla G_f(x^t) \| \leq \frac{1}{13n} \gamma (1 + C_f) \alpha^4 \| \nabla f(x^t) \|^2.$$

In the last line, we apply eq. (14) and $\delta = \frac{\gamma \alpha^3}{13}$. The second term is

$$\mathbb{E}[\langle \nabla_{i^t} f(\bar{x}^{t+1}) - \nabla_{i^t} f(x^t) - \nabla_{i^t} G_f(\bar{x}^{t+1}) + \nabla_{i^t} G_f(x^t), \alpha(\tilde{\nabla}_{i^t} G_f(x^t) - \nabla_{i^t} G_f(x^t)) \rangle | x^t]$$

$$\leq \mathbb{E}[\| \nabla_{i^t} f(\bar{x}^{t+1}) - \nabla_{i^t} f(x^t) - \nabla_{i^t} G_f(\bar{x}^{t+1}) + \nabla_{i^t} G_f(x^t) \| \| \alpha(\tilde{\nabla}_{i^t} G_f(x^t) - \nabla_{i^t} G_f(x^t)) \| | x^t]$$

$$\leq \mathbb{E}[(L + L')\alpha \| \bar{x}_{i^t}^{t+1} - x_{i^t}^t \| \| \tilde{\nabla}_{i^t} G_f(x^t) - \nabla_{i^t} G_f(x^t) \| | x^t]$$

$$\leq \mathbb{E}[(L + L')\alpha^2 \| \nabla_{i^t} f(x^t) - \nabla_{i^t} G_f(x^t) \| \| \tilde{\nabla}_{i^t} G_f(x^t) - \nabla_{i^t} G_f(x^t) \| | x^t]$$

$$\leq \frac{1}{n} \sum_{i=1}^{n} [(L + L')\alpha^2 \| \nabla_i f(x^t) - \nabla_i G_f(x^t) \| \| \tilde{\nabla}_i G_f(x^t) - \nabla_i G_f(x^t) \|]$$

$$\leq \frac{1}{n} (L + L')\alpha^2 \| \nabla f(x^t) - \nabla G_f(x^t) \| \| \tilde{\nabla} G_f(x^t) - \nabla G_f(x^t) \|$$

$$\leq \frac{1}{n} (L + L')\alpha^2 (\| \nabla f(x^t) \| + \| \nabla G_f(x^t) \|) \| \tilde{\nabla} G_f(x^t) - \nabla G_f(x^t) \|$$

$$\leq \frac{1}{n} (L + L')(1 + C_f)\alpha^2 \| \nabla f(x^t) \| \| \tilde{\nabla} G_f(x^t) - \nabla G_f(x^t) \|$$

$$\leq \frac{1}{13n} \gamma (L + L')(1 + C_f)\alpha^5 \| \nabla f(x^t) \|^2 \leq \frac{1}{13n} \gamma (1 + C_f)\alpha^4 \| \nabla f(x^t) \|^2.$$

In the sixth line, we apply Cauchy-Schwartz inequality. In the ninth line, we apply eq. (14) and $\delta = \frac{\gamma \alpha^3}{13}$. The third term is

$$\mathbb{E}[\frac{L + L'}{2} \| \alpha(\tilde{\nabla}_{i^t} G_f(x^t) - \nabla_{i^t} G_f(x^t)) \|^2 | x^t]$$

$$= \frac{L + L'}{2n} \| \alpha(\tilde{\nabla} G_f(x^t) - \nabla G_f(x^t)) \|^2 \leq \frac{L + L'}{338n} \gamma^2 \alpha^8 \| \nabla f(x^t) \|^2 \leq \frac{1}{338n} \gamma^2 \alpha^7 \| \nabla f(x^t) \|^2.$$

In the second line, we applied eq. (14) and $\delta = \frac{\gamma \alpha^3}{13}$. Overall, we obtain

$$\mathbb{E}[f(x^{t+1}) - G_f(x^{t+1}) | x^t] - \mathbb{E}[f(\bar{x}^{t+1}) - G_f(\bar{x}^{t+1}) | x^t]$$

$$\leq \frac{2}{13n} \gamma (1 + C_f)\alpha^4 \| \nabla f(x^t) \|^2 + \frac{1}{338n} \gamma^2 \alpha^7 \| \nabla f(x^t) \|^2 \leq \frac{3}{13n} \gamma (1 + C_f)\alpha^4 \| \nabla f(x^t) \|^2.$$

and,

$$\mathbb{E}[f(x^{t+1}) - G_f(x^{t+1}) | x^t]$$

$$= \mathbb{E}[f(\bar{x}^{t+1}) - G_f(\bar{x}^{t+1}) | x^t] + \mathbb{E}[f(x^{t+1}) - G_f(x^{t+1}) | x^t] - \mathbb{E}[f(\bar{x}^{t+1}) - G_f(\bar{x}^{t+1}) | x^t]$$

$$\leq f(x^t) - G_f(x^t) - \frac{1}{2n} \alpha \| \nabla f(x^t) - \nabla G_f(x^t) \|^2 + \frac{3}{13n} \gamma (1 + C_f)\alpha^4 \| \nabla f(x^t) \|^2$$

$$\leq f(x^t) - G_f(x^t) - \frac{\alpha}{4n} \| \nabla f(x^t) - \nabla G_f(x^t) \|^2,$$

$$\leq f(x^t) - G_f(x^t) - \frac{\alpha \nu}{2n} (f(x^t) - G_f(x^t))^{\frac{2}{\theta}}.$$

In the last two line, we apply eq. (22) and $\alpha \leq (\frac{13}{12(1+C_f)})^{1/3} \frac{\| \nabla f(x^t) - \nabla G_f(x^t) \|}{\| \nabla f(x^t) \|}$.

From Lemma 6 of Fatkhullin et al. (2022), we have

$$\mathbb{E}[f(x^{t+k}) - G_f(x^{t+k})|x^t] \leq \frac{(4n)^{\frac{\theta}{2-\theta}} \frac{2-\theta}{\theta}^{-\frac{\theta+2}{2-\theta}} + (2n)^{\frac{\theta}{2-\theta}} \theta^{-\frac{\theta}{2-\theta}} + (\nu\alpha)^{\frac{\theta}{2-\theta}} (f(x^t) - G_f(x^t))}{(\nu\alpha(k+1))^{\frac{\theta}{2-\theta}}}.$$

$\square$

## D  ALMOST SURELY CONVERGENCE TO LOCAL MINIMUM

Let the function $g, g_1, ..., g_n$ to be $(x'_i, x'_{-i}) = g_i(x_i, x_{-i}) = (x_i - \alpha\nabla_i f(x_i, x_{-i}), x_{-i})$ and $g = g_n \circ g_{n-1} \circ \cdots \circ g_1$. Then, we have $x^{t+1} = g(x^t)$.

**Theorem D.1.** *Under assumption 2.2, if $f$ is twice continuously differentiable, $g$ is locally diffeomorphism for $\alpha < \frac{1}{L_c}$.*

*Proof.* To show $g$ is bijective, we only need to show $g_i$ is bijective for all $i$. We firstly show $g_i$ is injective for $\alpha < \frac{1}{L_c}$. If $g_i(x_i, x_{-i}) = g_i(y_i, y_{-i})$, we must have $x_{-i} = y_{-i}$ from the definition of $g_i$. Then, $\|x_i - y_i\| = \alpha\|\nabla_i f(x_i, x_{-i}) - \nabla_i f(y_i, y_{-i})\| = \alpha\|\nabla_i f(x_i, x_{-i}) - \nabla_i f(y_i, x_{-i})\| \leq \alpha L_c\|x_i - y_i\|$. As $\alpha < \frac{1}{L}$, we have $x_i = y_i$.

To show $g$ is surjective, we consider the following problem,

$$\min[\frac{1}{2}\|x_i - y_i\|^2 - \alpha f(x_i, x_{-i})].$$

For $\alpha < \frac{1}{L}$, this function is strongly convex when $x_{-i}$ are fixed. So there is a unique minimizer $x_{y_i}$ such that $y_i = x_{y_i} - \alpha\nabla_i f(x_{y_i}, x_{-i})$ for all $x_{-i}$. By setting $x_{-i} = y_{-i}$, we would have $y = g_i(x_y)$ where the j-th block of $x_y$ is $x_{y_i}$ if $j = i$ and is $y_j$ if $j \neq i$. We have already shown $g_i$ is bijective. Because $g = g_n \circ g_{n-1} \circ \cdots \circ g_1$, $g$ is also bijective and also invertible.

As $f$ is twice continuously differentiable, $g_i$ is continuously differentiable. Because the composition of continuously differentiable functions is continuously differentiable, $g$ is continuously differentiable. From the definition of $g$, the Jacobian of $g$ is

$$Dg(x) = Dg_n(g_{n-1:1}(x))Dg_{n-1}(g_{n-2:1}(x))\dots Dg_2(g_1(x))Dg_1(x).$$

and the Jacobian of $g_i$ is

$$Dg_i(x) = I - E_i\nabla^2 f(x)$$

where the $i$-th diagonal block of $E_i = I^{d_i \times d_i}$ and 0 elsewhere. It can be easily observed that the fixed point of $g$ is equivalent to the Nash Equilibrium point of $f$. For any Nash Equilibrium point $x^\star$ with $\lambda_{min}[\nabla^2 f(x^\star)] < 0$, we can represent $Dg(x^\star)$

$$Dg(x^\star) = (I - \alpha E_n\nabla^2 f(g_{n-1:1}(x^\star)))(I - \alpha E_{n-1}\nabla^2 f(g_{n-2:1}(x^\star)))\cdots$$
$$\cdots (I - \alpha E_2\nabla^2 f(g_1(x^\star)))(I - \alpha E_1\nabla^2 f(x^\star)),$$
$$= (I - \alpha E_n\nabla^2 f(x^\star))(I - \alpha E_{n-1}\nabla^2 f(x^\star))\dots(I - \alpha E_2\nabla^2 f(x^\star))(I - \alpha E_1\nabla^2 f(x^\star)).$$

Since $\alpha < \frac{1}{L}$ and $I - \alpha\nabla^2_{i,i}f(x^\star) > 0$, $det(I - \alpha E_i\nabla^2 f(x^\star)) = det|I - \alpha\nabla^2_{i,i}f(x^\star)| \neq 0$. As a result, $(I - \alpha E_i\nabla^2 f(x^\star))$ is invertible for all $i$. So $Dg(x^\star)$ is also invertible. Overall $g$ is locally diffeomorphism. $\square$

**Theorem D.2.** *Let $C$ be the set of strict saddle points, i.e., $\lambda_{min} < 0$. If $C$ has at most countably infinite cardinality and $\alpha < \frac{1}{L_c}$ under BCD and $f$ is twice continuously differentiable, then*

$$Pr(\lim_t x^t \in C) = 0.$$

*Proof.* Since $\lambda_{min}[\nabla^2 f(x^\star)] < 0$ and the set $W^{cs}_{loc}$ is a manifold equal to the number of non-negative eigenvalues of $\nabla^2 f(x^\star)$, this manifold has measure zero. Let $B$ be the neighborhood of $x^\star$. If $x^t$ converge to the $x^\star$, then there exists a $T$ such that $g^t(x) \in B$ for all $t \geq T$. This means that $g^t(x) \in \bigcap_{k=0}^\infty g^{-k}(B) \subseteq W^{cs}_{loc}$. Then we have the global stable set of $W^s(x^\star)$ satisfies

$$W^s(x^\star) \subseteq \bigcup_{k=0}^\infty g^{-k}(W^{cs}_{loc}).$$

which indicates $W^s(x^\star)$ also has measure zero. And for the set $C$,

$$Pr(\lim_t x^t \in C) = \sum_{x^\star \in C} Pr(\lim_t x^t = x^\star) = 0.$$

□

# E    PROOFS OF THE APPLICATION SECTION

## E.1    PROOF OF $N$-SIDED PL CONDITION FOR MULTI-PLAYER LINEAR QUADRATIC GAME

The system can be written down as

$$x^{t+1} = Ax^t + \sum_{i=1}^N B_i u_i^t = Ax^t + \sum_{i=1}^N B_i K_i x^t = (A - \sum_{j \neq l} B_j K_j)x^t + B_l K_l x^t,$$

and the system can be written down as

$$f(K_l, K_{-l}) = \mathbb{E}_{x_0 \sim \mathcal{D}}[\sum_{t=0}^{+\infty}[(x^t)^T Q x^t + \sum_{i=1}^N ((x_i^t)^T K_i^T R_i K_i x_i^t]]$$

$$= \mathbb{E}_{x^0 \sim \mathcal{D}}[\sum_{t=0}^{+\infty}[(x^t)^T (Q + \sum_{j \neq l} K_j^T R_j K_j)x^t + (x_l^t)^T K_l^T R_l K_l x_l^t]].$$

Define $\Sigma_K$ as the state correlation matrix, i.e.

$$\Sigma_K = \mathbb{E}_{x^0 \sim \mathcal{D}} \sum_{t=0}^\infty x^t (x^t)^T.$$

From the Corollary 5 of Fazel et al. (2018), we have

$$f(K_l, K_{-l}) - \min_{K_l'} f(K_l', K_{-l}) \leq \frac{\left\|\Sigma_{K_{l,K_{-l}}^\star, K_{-l}}\right\|}{\sigma_{min}(\Sigma_0)^2 \sigma_{min}(R_l)} \|\nabla_{K_l} f(K_l, K_{-l})\|_F^2, \forall l$$

where $K_{l,K_{-l}}^\star \in \operatorname{argmin}_{K_l'} f(K_l', K_{-l})$. Since $K$ is bounded and $\sigma_{min}(\Sigma_0) > 0$, then $0 < \kappa < +\infty$, and $f$ satisfies $N$-sided PL condition.

## E.2    COUNTEREXAMPLE OF MULTI-CONVEXITY FOR $N$-PLAYER LINEAR-QUADRATIC GAME

Here, we only need to prove that there exists $K_1$, $K_1'$ and $K_2$ such that

$$f(K_1, K_2) + f(K_1', K_2) \leq 2f(\frac{K_1 + K_1'}{2}, K_2).$$

where $f(K_1, K_2)$ is the objective function of the 2-player potential quadratic game. We denote $A$ and $B$ to be $3 \times 3$ identity matrix and

$$K_1 = \begin{bmatrix} 0 & 0 & -10 \\ -1 & 0 & 0 \\ 0 & 0 & 0 \end{bmatrix} \text{ and } K_1' = \begin{bmatrix} 0 & -10 & 0 \\ 0 & 0 & 0 \\ -1 & 0 & 0 \end{bmatrix} \text{ and } K_2 = \begin{bmatrix} 1 & 0 & 0 \\ 0 & 1 & 0 \\ 0 & 0 & 1 \end{bmatrix}.$$

The matrices $A - B(K_1 + K_2)$ and $A - B(K_1' + K_2)$ are both stable, however, the matrix $A - B(\frac{K_1 + K_2}{2})$ is unstable. As a result, the objective function $f(K_1, K_2), f(K_1', K_2) < +\infty$ and $f(\frac{K_1 + K_1'}{2}, K_2) = +\infty$.

## E.3    PROOF OF PL CONDITION FOR LINEAR RESIDUAL NETWORKS

From Hardt & Ma (2017), we have

$$f(A) = \|E\Sigma^{1/2}\|_F^2 + C,$$

and

$$\|\frac{\partial f(A)}{\partial A_i}\|_F^2 = \|(I + A_{i+1}^T) \cdots (I + A_l^T) E\Sigma(I + A_1^T)(I + A_{i-1}^T)\|_F^2$$

$$\geq 4(1-\tau)^{2(l-1)}\sigma_{\min}(\Sigma)\|E\Sigma^{1/2}\|_F^2.$$

where $\Sigma = \mathbb{E}[xx^T]$, $E = (I + A_l)...(I + A_1) - R$, $\tau = \max_i \|A_i\| < 1$ and $C$ is a constant. Then, we have

$$\|\frac{\partial f(A)}{\partial A_i}\|_F^2 \geq 4(1-\tau)^{2(l-1)}\sigma_{\min}(\Sigma)(f(A) - C)$$

$$\geq 4(1-\tau)^{2(l-1)}\sigma_{\min}(\Sigma)(f(A) - \min_B f(B)) \qquad (23)$$

$$\geq 4(1-\tau)^{2(l-1)}\sigma_{\min}(\Sigma)(f(A) - \min_{B_i} f(B_i, A_{-i})).$$

where the last step comes from $\min_{B_i} f(B_i, A_{-i}) \geq \min f(A) \geq C$, $\forall i$. Notice that $(I + A_i)$ is invertible, therefore the best response of $i$-th weight matrix $A_i^\star(A)$ always exists, where others blocks are fixed to be $A_{-i}$. Because $\frac{\partial f(A_i^\star(A), A_{-i})}{\partial A_i} = \mathbf{0}$, from eq. (23), the function value at best response $f(A_i^\star(A), A_{-i}) = \min_B f(B)$. From the optimality condition, the full gradient

$$\nabla f(A_i^\star(A), A_{-i}) = \mathbf{0}, \forall i.$$

As a result,

$$\nabla G_f(A) = \frac{1}{n}\sum_{i=1}^{n}\nabla f(A_i^\star(A), A_{-i}) = \mathbf{0},$$

which indicates $\langle \nabla G(A), \nabla f(A)\rangle = 0 \leq \kappa\|\nabla f(A)\|_F^2$ by setting $\kappa = 0$.

# F    DISCUSSION ON ASSUMPTION 3.5

We have the following theorem which shows correlation with assumption 3.5 in the continuous dynamic, i.e., there exists a neighborhood around every isolated local minimum of a locally strongly convex and smooth functions such that, on average, the condition in equation 5 holds for all iterates of the GD algorithm.

**Theorem F.1.** *If $x^\star$ is the isolated local minimum in $U$ and $G_f$ exists, then there exists a radius $r > 0$ s.t. $\forall x_0 \in \mathcal{B}(x^\star, r) \subseteq U$, such that by following the dynamics*

$$\begin{aligned} r(0) &= x_0 \in U, \\ \dot{r}(t) &= -\nabla f(x)|_{x=r(t)}, \end{aligned} \qquad (24)$$

*we have*

$$\int_0^{+\infty} \langle G_f(x), \nabla f(x)\rangle|_{x=r(t)}dt \leq \int_0^{+\infty} \|\nabla f(x)\|^2|_{x=r(t)}dt,$$

*if further $\nabla^2 f(x^\star)$ is positive definite, $\nabla^2 f$ is continuous and $f$ is $L$-smooth,*

$$\int_0^{+\infty} \langle G_f(x), \nabla f(x)\rangle|_{x=r(t)}dt \leq \int_0^{+\infty} \left(1 - \frac{\lambda_{min}^2(\nabla^2 f(x^\star))}{2nL^2}\right) \|\nabla f(x)\|^2|_{x=r(t)}dt.$$

*Proof.* Since $x^\star$ is the isolated local minimum in $U$, $f(x)$ is a positive definite function on $U$. As a result,

$$\dot{f}(r(t)) = \langle \nabla f(x)|_{x=r(t)}, \dot{r}(t)\rangle = -\|\nabla f(x)|_{x=r(t)}\|^2 < 0,$$

for all $r(t) \in U$, $r(t) \neq x^\star$. This indicates $x^\star$ is asymptotically stable. Then, there exists a radius $r > 0$ such that $B = \mathcal{B}(x^\star, r) \subseteq U$. And, if $r(0) \in B$, then $\lim_{t\to+\infty} r(t) = x^\star$. Now consider any $r(0) = x \in B$, we have

$$f(x^\star) - f(x_0) = \int_0^{+\infty} \langle \nabla f(x)|_{x=r(t)}, \dot{r}(t)\rangle dt,$$

and

$$G_f(x^\star) - G_f(x_0) = \int_0^{+\infty} \langle \nabla G_f(x)|_{x=r(t)}, \dot{r}(t)\rangle dt.$$

From these two equations, we have

$$G_f(x^\star) - G_f(x_0) - (f(x^\star) - f(x_0)) = f(x_0) - G_f(x_0) \geq \frac{1}{2nL}\|\nabla f(x_0)\|^2 \geq 0.$$

As a result

$$
\begin{aligned}
f(x_0) - G_f(x_0) &= \int_0^{+\infty} \langle \nabla(G_f(x) - f(x))|_{x=r(t)}, \dot{r}(t)\rangle dt, \\
&= \int_0^{+\infty} \langle \nabla(f(x) - G_f(x)), \nabla f(x)\rangle|_{x=r(t)} dt, \\
&= \int_0^{+\infty} (\|\nabla f(x)\|^2 - \langle G_f(x), \nabla f(x)\rangle)|_{x=r(t)} dt \geq 0.
\end{aligned}
\tag{25}
$$

If $\nabla^2 f(x^\star) > 0$, then defines

$$F(x) = f(x) - G_f(x) - \frac{1}{2nL}\|\nabla f(x)\|^2 \geq 0 = F(x^\star).$$

Its Hessian is positive semidefinite at $x^\star$, i.e.

$$\nabla^2 F(x^\star) = \nabla^2 f(x^\star) - \nabla^2 G_f(x^\star) - \frac{1}{nL}(\nabla^2 f(x^\star))^2 \succeq 0,$$

$$\implies \nabla^2 f(x^\star) - \nabla^2 G_f(x^\star) \succeq \frac{1}{nL}(\nabla^2 f(x^\star))^2 \succ 0.$$

In consequence, there exists a radius $r' \leq r$ such that

$$\nabla^2 f(x) - \nabla^2 G_f(x) \succeq \frac{1}{2nL}(\nabla^2 f(x))^2, \forall x \in \mathcal{B}(x^\star, r').$$

So the function $f(x) - G_f(x)$ is locally convex around the neighborhood of $x^\star$. And for $r(0) = x_0 \in \mathcal{B}(x^\star, r')$

$$
\begin{aligned}
f(x_0) - G_f(x_0) &\geq \frac{\lambda_{min}^2(\nabla^2 f(x^\star))}{4nL}\|x_0 - x^\star\|^2, \\
&\geq \frac{\lambda_{min}^2(\nabla^2 f(x^\star))}{2nL^2}(f(x_0) - f(x^\star)), \\
&= -\frac{\lambda_{min}^2(\nabla^2 f(x^\star))}{2nL^2}\int_0^{+\infty}\langle\nabla f(x)|_{x=r(t)}, \dot{r}(t)\rangle dt, \\
&= \frac{\lambda_{min}^2(\nabla^2 f(x^\star))}{2nL^2}\int_0^{+\infty}\|\nabla f(x)\|^2|_{x=r(t)} dt.
\end{aligned}
\tag{26}
$$

From eq. (25) and eq. (26), we have

$$\int_0^{+\infty}(\|\nabla f(x)\|^2 - \langle G_f(x), \nabla f(x)\rangle)|_{x=r(t)} dt \geq \frac{\lambda_{min}^2(\nabla^2 f(x^\star))}{2nL^2}\int_0^{+\infty}\|\nabla f(x)\|^2|_{x=r(t)} dt,$$

$$\implies \int_0^{+\infty}\left(\left(1 - \frac{\lambda_{min}^2(\nabla^2 f(x^\star))}{2nL^2}\right)\|\nabla f(x)\|^2 - \langle G_f(x), \nabla f(x)\rangle\right)|_{x=r(t)} dt \geq 0.$$

$$\square$$

**Theorem F.2.** *If $f(x)$ satisfies the assumption of theorem 3.10, then, by denoting $S(\gamma, C)$ as the set of non-NE points that don't satisfy case 1 and case 2, we have,*

$$\lim_{\gamma\to 1, C\to 0}|S(\gamma, C)| = 0,\tag{27}$$

*where $|S(\gamma, C)|$ is the measure of $S(\gamma, C)$, if $S(\gamma, C)$ is non-empty,*

$$\lim_{\gamma\to 1, C\to 0}\max_{x\in S(\gamma, C)} f(x) - G_f(x) = 0.\tag{28}$$

*Proof.* Suppose case 1 and case 2 don't satisfy, then the iterates satisfy,

$$\langle \nabla f(x^t), \nabla G_f(x^t) > \gamma \|\nabla f(x^t)\|^2,$$
$$\frac{(\|\nabla G_f(x^t)\|^2 - \langle \nabla f(x^t), \nabla G(x^t)\rangle)^2}{\langle \nabla f(x^t), \nabla G(x^t)\rangle^2} < C. \tag{29}$$

By simplifying the second equation and consider $\langle \nabla f(x^t), \nabla G_f(x^t)\rangle > \gamma \|\nabla f(x^t)\|^2 > 0$, we have

$$\langle \nabla f(x^t), \nabla G_f(x^t)\rangle > \gamma \|\nabla f(x^t)\|^2,$$
$$(1 - \sqrt{C})\langle \nabla f(x^t), \nabla G_f(x^t)\rangle < \|\nabla G_f(x^t)\|^2 < (1 + \sqrt{C})\langle \nabla f(x^t), \nabla G_f(x^t)\rangle. \tag{30}$$

In consequence,

$$\|\nabla f(x^t) - \nabla G_f(x^t)\|^2 = \|\nabla f(x^t)\|^2 - 2\langle \nabla f(x^t), \nabla G_f(x^t)\rangle + \|\nabla G_f(x^t)\|^2,$$
$$< (1 + (1 + \sqrt{C})^2)\|\nabla f(x^t)\|^2 - 2\langle \nabla f(x^t), \nabla G_f(x^t)\rangle,$$
$$< (1 + (1 + \sqrt{C})^2 - 2\gamma)\|\nabla f(x^t)\|^2, \tag{31}$$
$$< 2(1 + (1 + \sqrt{C})^2 - 2\gamma)Ln(f(x^t) - G_f(x^t)).$$

and $f(x^t) - G_f(x^t)$ satisfies,

$$f(x^t) - G_f(x^t) < \frac{\|\nabla f(x^t) - \nabla G_f(x^t)\|^\theta}{(2\nu)^{\theta/2}},$$
$$< (\frac{2(1 + (1 + \sqrt{C})^2 - 2\gamma)Ln}{2\nu})^{\theta/2}(f(x^t) - G_f(x^t))^{\theta/2}. \tag{32}$$

The above inequality brings the upper bound for $f(x^t) - G_f(x^t)$ and $\|\nabla f(x^t)\|$,

$$f(x^t) - G_f(x^t) < (\frac{2(1 + (1 + \sqrt{C})^2 - 2\gamma)Ln}{2\nu})^{\frac{\theta}{2-\theta}}, \tag{33}$$

and

$$\|\nabla f(x^t)\|^2 \leq 2Ln(f(x^t) - G_f(x^t)) < 2Ln(\frac{2(1 + (1 + \sqrt{C})^2 - 2\gamma)Ln}{2\nu})^{\frac{\theta}{2-\theta}}. \tag{34}$$

As $C \to 0$ and $\gamma \to 1$, $f(x^t) - G_f(x^t) < \epsilon$, $\forall \epsilon > 0$. Notice that we consider the non-NE point, which implies

$$0 < \|\nabla f(x^t)\|^2 < 2Ln(\frac{2(1 + (1 + \sqrt{C})^2 - 2\gamma)Ln}{2\nu})^{\frac{\theta}{2-\theta}}. \tag{35}$$

As a result, as $C \to 0$ and $\gamma \to 1$, the point that satisfies case 3 has its measure converge to 0. $\qquad\square$

# G  ADAPTIVE GD ALGORITHMS

## G.1  IDEAL ADAPTIVE GRADIENT DESCENT

**Theorem G.1.** *For an $n$-side $\mu$-PL function $f(x)$ satisfying assumption 2.1, by applying algorithm 5,*

- *in Case 1 with $\alpha \leq \frac{2(1-\gamma)}{2L' + (1+\gamma)L}$, we have*

$$f(x^{t+1}) - G_f(x^{t+1}) \leq \Big(1 - \frac{n\mu\alpha(1-\gamma)}{2}\Big)(f(x^t) - G_f(x^t)),$$

- *in Case 2 with $\alpha \leq \min\{\frac{1}{2(L+L')}, \frac{C}{2(L+L')}\}$, we have*

$$f(x^{t+1}) - G_f(x^{t+1}) \leq \Big(1 - \frac{n(L+L')\mu\alpha^2}{2}\Big)(f(x^t) - G_f(x^t)),$$

- *in Case 3 with $\alpha \leq \frac{1}{L+L'}$, $f - G_f$ is non-increasing. Furthermore, if $f - G_f$ satisfies $(\theta, \nu)$-PL condition and case 3 are satisfied from iterates $t$ to $t + k$, we have*

$$f(x^{t+1}) - G_f(x^{t+1}) \leq \frac{(2)^{\frac{\theta}{2-\theta}}\frac{2-\theta}{\theta}^{-\frac{\theta+2}{2-\theta}} + \theta^{-\frac{\theta}{2-\theta}} + (\nu\alpha)^{\frac{\theta}{2-\theta}}(f(x^t) - G_f(x^t))}{(\nu\alpha(k+1))^{\frac{\theta}{2-\theta}}}$$

---

**Algorithm 5** Ideal Adaptive Gradient Descent (IA-GD)

---

**Input:** initial point $x^0 = (x_1^0, ..., x_n^0)$, learning rate $\alpha$, $0 \leq \gamma < 1$ and $C > 0$
**for** $t = 0$ **to** $T - 1$ **do**
   **if** $\langle \nabla G_f(x^t), \nabla f(x^t) \rangle \leq \gamma \|\nabla f(x^t)\|^2$ **then**
     $k^t = 0$
   **else if** $\frac{(\|\nabla G_f(x^t)\|^2 - \langle \nabla f(x^t), \nabla G(x^t) \rangle)^2}{\langle \nabla f(x^t), \nabla G(x^t) \rangle^2} > C$ **then**
     $k^t = -2 + \frac{\langle \nabla f(x^t), \nabla G_f(x^t) \rangle}{\|\nabla G_f(x^t)\|^2}$
   **else**
     $k^t = -1$
   **end if**
   $x^{t+1} = x^t - \alpha(\nabla f(x^t) + k^t \nabla G_f(x^t))$
**end for**

---

*Proof.* **Case 1:** This is analogous to the proof of Theorem 3.7.

**Case 2:** From the smoothness assumption, we get

$$
f(x^{t+1}) \leq f(x^t) + \langle \nabla f(x^t), x^{t+1} - x^t \rangle + \frac{L}{2}\|x^{t+1} - x^t\|^2
$$

$$
= f(x^t) - \alpha \langle \nabla f(x^t), \nabla f(x^t) + k^t \nabla G_f(x^t) \rangle + \frac{L\alpha^2}{2}\|\nabla f(x^t) + k^t \nabla_{i^t} G_f(x^t)\|^2
$$

$$
= f(x^t) - (\alpha - \frac{L\alpha^2}{2})\|\nabla f(x^t)\|^2 - (\alpha k^t - L\alpha^2 k^t)\langle \nabla f(x^t), \nabla G_f(x^t) \rangle
$$

$$
+ \frac{L\alpha^2(k^t)^2}{2}\|\nabla G_f(x^t)\|^2.
$$

For $G_f(x)$, we have

$$
G_f(x^t) \leq G_f(x^{t+1}) - \langle \nabla G_f(x^t), x^{t+1} - x^t \rangle + \frac{L'}{2}\|x^{t+1} - x^t\|^2,
$$

$$
= G_f(x^{t+1}) + \alpha \langle \nabla G_f(x^t), \nabla f(x^t) + k^t \nabla G_f(x^t) \rangle
$$

$$
+ \frac{L'\alpha^2}{2}\|\nabla f(x^t) + k^t \nabla G_f(x^t)\|^2,
$$

$$
= G_f(x^{t+1}) + \alpha k^t \|\nabla G_f(x^t)\|^2 + (\alpha + L'\alpha^2 k^t)\langle \nabla G_f(x^t), \nabla f(x^t) \rangle
$$

$$
+ \frac{L'\alpha^2}{2}\|\nabla f(x^t)\|^2 + \frac{L'\alpha^2(k^t)^2}{2}\|\nabla G_f(x^t)\|^2.
$$

As a result, we get

$$
f(x^{t+1}) - G_f(x^{t+1}) \leq f(x^t) - G_f(x^t) - (\alpha - \frac{L\alpha^2}{2} - \frac{L'\alpha^2}{2})\|\nabla f(x^t)\|^2
$$

$$
- (\alpha k^t - L\alpha^2 k^t - \alpha - L'\alpha^2 k^t)\langle \nabla f(x^t), \nabla G_f(x^t) \rangle \qquad (36)
$$

$$
+ \frac{1}{2}((L' + L)\alpha^2(k^t)^2 + 2\alpha k^t)\|\nabla G_f(x^t)\|^2.
$$

Now, we define

$$
h(k^t) := -(\alpha k^t - L\alpha^2 k^t - \alpha - L'\alpha^2 k^t)\langle \nabla f(x^t), \nabla G_f(x^t) \rangle
$$

$$
+ \frac{1}{2}((L' + L)\alpha^2(k^t)^2 + 2\alpha k^t)\|\nabla G_f(x^t)\|^2,
$$

which is a convex function. We have

$$
h(-1) = -\frac{2\alpha - (L + L')\alpha^2}{2}\|\nabla f(x^t) - \nabla G_f(x^t)\|^2 + \left(\alpha - \frac{L\alpha^2}{2} - \frac{L'\alpha^2}{2}\right)\|\nabla f(x^t)\|^2,
$$

$$
\leq \left(\alpha - \frac{L\alpha^2}{2} - \frac{L'\alpha^2}{2}\right)\|\nabla f(x^t)\|^2.
$$

The function value $h(k^t)$ at minimizer $k^t = k^\star = -\frac{((L+L')\alpha - 1)\langle \nabla f, \nabla G_f\rangle + \|\nabla G_f\|^2}{(L+L')\alpha\|\nabla G_f\|^2}$ is less or equals to zero if

$$(L+L')^2\langle \nabla f, \nabla G_f\rangle^2\alpha^2 - 2(L+L')\langle \nabla f, \nabla G_f\rangle^2\alpha + (\|\nabla G_f\|^2 - \langle \nabla f, \nabla G_f\rangle)^2 \geq 0.$$

$$\alpha \leq \frac{1}{2(L+L')}\frac{(\|\nabla G_f\|^2 - \langle \nabla f, \nabla G_f\rangle)^2}{\langle \nabla f, \nabla G_f\rangle^2}. \tag{37}$$

Since in this case $\frac{(\|\nabla G_f\|^2 - \langle \nabla f, \nabla G_f\rangle)^2}{\langle \nabla f, \nabla G_f\rangle^2} \geq C$, eq. (37) is satisfied if

$$\alpha \leq \frac{C}{2(L+L')}.$$

In consequence, if $\alpha \leq \frac{C}{2(L+L')}$, $\forall \lambda \in [0,1]$, we have

$$h(-\lambda + (1-\lambda)k^\star) \leq \lambda h(-1) + (1-\lambda)h(k^\star) \leq \lambda\Big(\alpha - \frac{L\alpha^2}{2} - \frac{L'\alpha^2}{2}\Big)\|\nabla f(x^t)\|^2$$

By setting $k^t = -1 + \frac{\langle \nabla f(x^t), \nabla G_f(x^t)\rangle - \|\nabla G_f(x^t)\|^2}{\|\nabla G_f(x^t)\|^2} = -\lambda + (1-\lambda)k^\star$ and $\alpha \leq \frac{1}{2(L+L')}$, we have

$$0 \leq \lambda = 1 - \frac{(L+L')\alpha(k^t + 1)\|\nabla G_f\|^2}{(1-(L+L')\alpha)(\langle \nabla f, \nabla G_f\rangle - \|\nabla G_f\|^2)} = 1 - \frac{(L+L')\alpha}{1-(L+L')\alpha} < 1.$$

and

$$h(k^t) = h(-\lambda + (1-\lambda)k^\star) \leq (1 - \frac{(L+L')\alpha}{1-(L+L')\alpha})\Big(\alpha - \frac{L\alpha^2}{2} - \frac{L'\alpha^2}{2}\Big)\|\nabla f(x^t)\|^2.$$

As a result,

$$f(x^{t+1}) - G_f(x^{t+1})$$
$$\leq f(x^t) - G_f(x^t) - \Big(\alpha - \frac{L\alpha^2}{2} - \frac{L'\alpha^2}{2}\Big)\|\nabla f(x^t)\|^2 + h(k^t)$$
$$\leq f(x^t) - G_f(x^t) - \frac{(L+L')\alpha}{1-(L+L')\alpha}\Big(\alpha - \frac{L\alpha^2}{2} - \frac{L'\alpha^2}{2}\Big)\|\nabla f(x^t)\|^2$$
$$\leq f(x^t) - G_f(x^t) - \frac{1}{2}\frac{(L+L')\alpha^2}{1-(L+L')\alpha}\|\nabla f(x^t)\|^2$$
$$\leq \Big(1 - \frac{n(L+L')\mu\alpha^2}{1-(L+L')\alpha}\Big)(f(x^t) - G_f(x^t))$$
$$\leq \Big(1 - \frac{n(L+L')\mu\alpha^2}{2}\Big)(f(x^t) - G_f(x^t)).$$

**Case 3:** In this case, notice that $f - G_f$ is $L + L'$-smooth,

$$f(x^{t+k}) - G_f(x^{t+k})$$
$$\leq f(x^t) - G_f(x^t) + \langle \nabla f(x^t) - \nabla G(x^t), x^{t+1} - x^t\rangle + \frac{L+L'}{2}\|x^{t+1} - x^t\|^2,$$
$$= f(x^t) - G_f(x^t) - (\alpha - \frac{L\alpha^2}{2})\|\nabla f(x^t) - \nabla G(x^t)\|^2,$$
$$\leq f(x^t) - G_f(x^t) - \frac{1}{2}\alpha\|\nabla f(x^t) - \nabla G(x^t)\|^2,$$
$$\leq f(x^t) - G_f(x^t) - \nu\alpha(f(x^t) - \nabla G(x^t))^{2/\theta}$$

The result follows directly from Lemma 6 of Fatkhullin et al. (2022). □

---

**Algorithm 6** Adaptive Gradient Descent (A-GD)

---

**Input:** initial point $x^0 = (x_1^0, ..., x_n^0)$, learning rates $\alpha, \beta, 0 < \gamma < 1$ and $C > 0$
**for** $t = 0$ **to** $T - 1$ **do**
  $y^{t,T'} = $ABR$(x^t, T', \beta)$                                       `:Algorithm 4`
  compute $\tilde{\nabla} G_f(x^t) := \frac{1}{n} \sum_{l=1}^n \nabla f(y_l^{t,T'}, x_{-l}^t)$
  **if** $\langle \tilde{\nabla} G_f(x^t), \nabla f(x^t) \rangle \leq (\gamma - \gamma \frac{\alpha^3}{13}) \|\nabla f(x^t)\|^2$ **then**
    $\tilde{k}^t = 0$
  **else if** $\frac{(\|\tilde{\nabla} G_f(x^t)\|^2 - \langle \nabla f(x^t), \tilde{\nabla} G_f(x^t) \rangle)^2}{\|\tilde{\nabla} G_f(x^t)\|^4} > C$ **then**
    $\tilde{k}^t = -2 + \frac{\langle \nabla f(x^t), \tilde{\nabla} G_f(x^t) \rangle}{\|\tilde{\nabla} G_f(x^t)\|^2}$
  **else**
    $\tilde{k}^t = -1$
  **end if**
  $x^{t+1} = x^t - \alpha(\nabla f(x^t) + \tilde{k}^t \tilde{\nabla} G_f(x^t))$
**end for**

---

### G.2 ADAPTIVE GRADIENT DESCENT

**Theorem G.2.** *For an $n$-sided PL function $f(x)$ satisfying assumption 2.1, by implementing algorithm 6 with $\beta \leq \frac{1}{L}$ and $T' \geq \frac{1}{\log(\frac{1}{1-\mu\beta})} \log\left(\frac{169nL^2}{\mu^2\gamma^2\alpha^6}\right)$,*

- *in Case 1 with $\alpha \leq \frac{2(1-\gamma)}{2L'+(1+\gamma)L}$, we have*

$$f(x^{t+1}) - G_f(x^{t+1}) \leq \left(1 - \frac{n\mu\alpha(1-\gamma)}{2}\right)(f(x^t) - G_f(x^t))$$

- *in Case 2 with $\alpha \leq \min\left\{(C_f)^{-1/2}, \left(\frac{3C\gamma}{(13+12\gamma)C_f}\right)^{1/2}, \frac{71C\gamma^2}{676(L+L')}, \frac{3\gamma(L+L')\mu}{(13+108\gamma)C_f^4}, \frac{1}{2(L+L')}\right\}$, we have*

$$f(x^{t+1}) - G_f(x^{t+1}) \leq \left(1 - \frac{n(L+L')\mu\alpha^2}{4}\right)(f(x^t) - G_f(x^t)).$$

- *in Case 3 with $\alpha \leq \min\left\{\frac{1}{L+L'}, \left(\frac{13}{12(1+C_f)}\right)^{1/3} \frac{\|\nabla f(x^t) - \nabla G_f(x^t)\|}{\|\nabla f(x^t)\|}\right\}$, $f - G_f$ is non-increasing. Furthermore, if $f - G_f$ satisfies $(\theta, \nu)$-PL condition and case 3 occurs from iterates $t$ to $t + k$, then*

$$f(x^{t+1}) - G_f(x^{t+1}) \leq \frac{(4)^{\frac{\theta}{2-\theta}} \frac{2-\theta}{\theta}^{-\frac{\theta+2}{2-\theta}} + (2)^{\frac{\theta}{2-\theta}} \theta^{-\frac{\theta}{2-\theta}} + (\nu\alpha)^{\frac{\theta}{2-\theta}}(f(x^t) - G_f(x^t))}{(\nu\alpha(k+1))^{\frac{\theta}{2-\theta}}}..$$

*Proof.* To approximate $G_f(x^t)$, we need to estimate the best response of i-th block $x_i^\star(x^t)$ when other blocks are fixed. As the function $f(x^t)$ satisfies $n$-sided PL condition, the function $f_i(x_i) = f(x_i, x_{-i}^t)$ satisfies strong PL condition. Therefore by applying the gradient descent with partial gradient $\nabla_i f(x_i, x_{-i}^t)$, the best response can be approximated efficiently. For any $\delta > 0$,

$$\|x_i^\star(x^t) - y_i^{t,T'}\|^2 \leq \frac{2}{\mu}(f(y_i^{t,T'}, x_{-i}^t) - \min_{x_i} f(x_i, x_{-i}^t))$$

$$\leq \frac{2}{\mu}(1 - \mu\beta)^{T'}(f(x^t) - \min_{x_i} f(x_i, x_{-i}^t)) \qquad (38)$$

$$\leq \frac{1}{\mu^2}(1 - \mu\beta)^{T'}\|\nabla_i f(x^t)\|^2 \leq \frac{\delta^2}{nL^2}\|\nabla_i f(x^t)\|^2.$$

if $T' \geq \frac{1}{\log(\frac{1}{1-\mu\beta})} \log(\frac{nL^2}{\mu^2\delta^2})$. The first inequality comes from the quadratic growth properties of the function $f_i(x_i) = f(x_i, x_{-i}^t)$ since it satisfies the strong PL condition. The second inequality comes

from the convergence of gradient descent under the PL condition. The third inequality comes from the definition of the n-sided PL condition.

$$
\begin{aligned}
\|\nabla G_f(x^t) - \tilde{\nabla} G_f(x^t)\| &= \left\| \sum_{i=1}^{n} \frac{1}{n} \nabla f(x_i^\star(x_{-i}), x_{-i}) - \sum_{i=1}^{n} \frac{1}{n} \nabla f(y_i^{t,T'}, x_{-i}) \right\| \\
&\leq \frac{1}{n} \sum_{i=1}^{n} \left\| \nabla f(x_i^\star(x^t), x_{-i}) - \nabla f(y_i^{t,T'}, x_{-i}) \right\| \\
&\leq \frac{L}{n} \sum_{i=1}^{n} \left\| x_i^\star(x^t) - y_i^{t,T'} \right\| \\
&\leq \frac{\delta}{\sqrt{n}} \sum_{i=1}^{n} \|\nabla_i f(x^t)\| \leq \delta \|\nabla f(x^t)\|.
\end{aligned}
\tag{39}
$$

In the fourth line, we apply the eq. (38). In the last line, we apply Cauchy-Schwartz inequality.

The second line comes from triangle inequality and the third line comes from the $L$-Lipschitz continuity of $\nabla f(x^t)$. Then, we denotes $\bar{x}^{t+1}$ as the iterates in the ideal case, i.e.

$$
\bar{x}_i^{t+1} = \begin{cases} x_i^t - \alpha(\nabla_i f(x^t) + k^t \nabla_i G(x^t)), & \text{if } i = i^t, \\ x_i^{t+1}, & \text{if } i \neq i^t. \end{cases}
\tag{40}
$$

Next, by choosing $\delta = \gamma \frac{\alpha^3}{13}$ we show the convergence of $f(x^t) - G_f(x^t)$. To do so, we break it into different cases.

**Case 1:** If $\langle \tilde{\nabla} G_f(x^t), \nabla f(x^t) \rangle \leq (\gamma - \gamma \frac{\alpha^3}{13}) \|\nabla f(x^t)\|^2$, we have

$$
\begin{aligned}
&\langle \nabla G_f(x^t), \nabla f(x^t) \rangle \\
&= \langle \nabla G_f(x^t) - \tilde{\nabla} G_f(x^t), \nabla f(x^t) \rangle + \langle \tilde{\nabla} G_f(x^t), \nabla f(x^t) \rangle \\
&\leq \|\nabla G_f(x^t) - \tilde{\nabla} G_f(x^t)\| \|\nabla f(x^t)\| + \langle \tilde{\nabla} G_f(x^t), \nabla f(x^t) \rangle \\
&\leq \gamma \frac{\alpha^3}{13} \|\nabla f(x^t)\|^2 + \langle \tilde{\nabla} G_f(x^t), \nabla f(x^t) \rangle \leq \gamma \|\nabla f(x^t)\|^2.
\end{aligned}
$$

By choosing $k^t = 0$, from theorem 3.6, we have

$$
\begin{aligned}
f(x^{t+1}) - G_f(x^{t+1}) &= f(\bar{x}^{t+1}) - G_f(\bar{x}^{t+1}) \\
&\leq \left( 1 - \frac{n\mu\alpha(1-\gamma)}{2} \right) (f(x^t) - G_f(x^t)).
\end{aligned}
$$

**Case 2:** $\left( \frac{\|\tilde{\nabla} G_f(x^t)\|^2}{\langle \nabla f(x^t), \tilde{\nabla} G_f(x^t) \rangle} - 1 \right)^2 \geq C$ and $\langle \tilde{\nabla} G_f(x^t), \nabla f(x^t) \rangle \geq \left( \gamma - \gamma \frac{\alpha^3}{13} \right) \|\nabla f(x^t)\|^2$. We firstly bound the difference of $\nabla G_f(x^t)$ and $\tilde{\nabla} G_f(x^t)$. From the assumption of case 2, we have

$$
\langle \tilde{\nabla} G_f(x^t), \nabla f(x^t) \rangle \geq \left( \gamma - \gamma \frac{\alpha^3}{13} \right) \|\nabla f(x^t)\|^2, \implies \|\tilde{\nabla} G_f(x^t)\| \geq \left( \gamma - \gamma \frac{\alpha^3}{13} \right) \|\nabla f(x^t)\|.
$$

This indicates

$$
\begin{aligned}
\left| \|\nabla G_f(x^t)\| - \|\tilde{\nabla} G_f(x^t)\| \right| &\leq \|\nabla G_f(x^t) - \tilde{\nabla} G_f(x^t)\| \leq \delta \|\nabla f(x^t)\| \\
&\leq \frac{\delta}{\gamma - \gamma \frac{\alpha^3}{13}} \|\tilde{\nabla} G_f(x^t)\| \leq \frac{1}{2} \|\tilde{\nabla} G_f(x^t)\|.
\end{aligned}
$$

In the last line, we apply $\delta = \frac{\gamma \alpha^3}{13} \leq \frac{\gamma - \gamma \frac{\alpha^3}{13}}{2}$. As a result,

$$
\left| \frac{\|\tilde{\nabla} G_f(x^t)\|}{\|\nabla G_f(x^t)\|} - 1 \right| \leq \frac{\delta}{\gamma - \gamma \frac{\alpha^3}{13}} \cdot \frac{\|\tilde{\nabla} G_f(x^t)\|}{\|\nabla G_f(x^t)\|},
$$

and $\frac{\|\tilde{\nabla}G_f(x^t)\|}{\|\nabla G_f(x^t)\|} \le 2$. These two inequalities imply

$$
\begin{aligned}
\Big|\frac{\|\tilde{\nabla}G_f(x^t)\|^2}{\|\nabla G_f(x^t)\|^2} - 1\Big| &= \Big(\frac{\|\tilde{\nabla}G_f(x^t)\|}{\|\nabla G_f(x^t)\|} + 1\Big)\Big|\frac{\|\tilde{\nabla}G_f(x^t)\|}{\|\nabla G_f(x^t)\|} - 1\Big| \\
&\le \Big(\frac{\|\tilde{\nabla}G_f(x^t)\|}{\|\nabla G_f(x^t)\|} + 1\Big)\frac{\delta}{\gamma - \gamma\frac{\alpha^3}{13}}\frac{\|\tilde{\nabla}G_f(x^t)\|}{\|\nabla G_f(x^t)\|} \le \frac{6\delta}{\gamma - \gamma\frac{\alpha^3}{13}} \le \frac{12\delta}{\gamma}.
\end{aligned}
\tag{41}
$$

In the last inequality, we applied $\alpha \le (C_f)^{-1/2} < 1$. Then we can bound the difference between $k^t$ and $\tilde{k}^t$.

$$
\begin{aligned}
|k^t - \tilde{k}^t| =& \Big|\frac{\langle\nabla f(x^t), \nabla G_f(x^t)\rangle}{\|\nabla G_f(x^t)\|^2} - \frac{\langle\nabla f(x^t), \tilde{\nabla}G_f(x^t)\rangle}{\|\tilde{\nabla}G_f(x^t)\|^2}\Big| \\
\le& \Big|\frac{\langle\nabla f(x^t), \nabla G_f(x^t)\rangle}{\|\nabla G_f(x^t)\|^2} - \frac{\langle\nabla f(x^t), \nabla G_f(x^t)\rangle}{\|\tilde{\nabla}G_f(x^t)\|^2}\Big| \\
&+ \Big|\frac{\langle\nabla f(x^t), \nabla G_f(x^t)\rangle}{\|\tilde{\nabla}G_f(x^t)\|^2} - \frac{\langle\nabla f(x^t), \tilde{\nabla}G_f(x^t)\rangle}{\|\tilde{\nabla}G_f(x^t)\|^2}\Big| \\
\le& \|\nabla f(x^t)\|\|\nabla G_f(x^t)\|\Big|\frac{1}{\|\tilde{\nabla}G_f(x^t)\|^2} - \frac{1}{\|\nabla G_f(x^t)\|^2}\Big| \\
&+ \|\nabla f(x^t)\|\|\nabla G_f(x^t) - \tilde{\nabla}G_f(x^t)\|\frac{1}{\|\tilde{\nabla}G_f(x^t)\|^2} \\
=& \|\nabla f(x^t)\|\|\nabla G_f(x^t)\|\frac{1}{\|\tilde{\nabla}G_f(x^t)\|^2}\Big|\frac{\|\tilde{\nabla}G_f(x^t)\|^2}{\|\nabla G_f(x^t)\|^2} - 1\Big| \\
&+ \|\nabla f(x^t)\|\|\nabla G_f(x^t) - \tilde{\nabla}G_f(x^t)\|\frac{1}{\|\tilde{\nabla}G_f(x^t)\|^2}, \\
\le& \frac{12\delta}{\gamma}\|\nabla f(x^t)\|\|\nabla G_f(x^t)\|\frac{1}{\|\tilde{\nabla}G_f(x^t)\|^2} \\
&+ \|\nabla f(x^t)\|\|\nabla G_f(x^t) - \tilde{\nabla}G_f(x^t)\|\frac{1}{\|\tilde{\nabla}G_f(x^t)\|^2} \\
\le& \frac{12\delta C_f}{\gamma}\frac{\|\nabla f(x^t)\|^2}{\|\tilde{\nabla}G_f(x^t)\|^2} + \|\nabla f(x^t)\|\|\nabla G_f(x^t) - \tilde{\nabla}G_f(x^t)\|\frac{1}{\|\tilde{\nabla}G_f(x^t)\|^2} \\
\le& \Big(\frac{12\delta C_f}{\gamma} + \delta\Big)\frac{\|\nabla f(x)\|^2}{\|\tilde{\nabla}G_f(x^t)\|^2} \le \frac{12\delta C_f}{\gamma\alpha} + \frac{\delta}{\alpha} \le \frac{13\delta C_f}{\gamma\alpha} \le C_f\alpha^2 \le 1.
\end{aligned}
\tag{42}
$$

where $C_f = \frac{L}{\sqrt{n}\mu} + 1$. The third line comes from Cauchy-Schwartz inequality. The sixth line comes from eq. (41). The eighth line comes from lemma 3.8. The ninth line comes from eq. (39). The last two lines come from $\delta = \frac{\gamma\alpha^3}{13}$ and $\alpha \le (C_f)^{-1/2}$. Also, the absolute value of $k^t$ and $\tilde{k}^t$ can be bounded.

$$
|\tilde{k}^t| = \Big| -2 + \frac{\langle\nabla f(x^t), \tilde{\nabla}G_f(x^t)\rangle}{\|\tilde{\nabla}G_f(x^t)\|^2}\Big| \le 2 + \frac{\|\nabla f(x^t)\|}{\|\tilde{\nabla}G_f(x^t)\|} \le 2 + \Big(\gamma - \gamma\frac{\alpha^3}{13}\Big)^{-1} \le 2 + \frac{13}{12\gamma}, \tag{43}
$$

and

$$
|k^t| = |k^t - \tilde{k}^t + \tilde{k}^t| \le |k^t - \tilde{k}^t| + |\tilde{k}^t| \le 3 + \frac{13}{12\gamma}. \tag{44}
$$

As a result,

$$
\begin{aligned}
\|k^t \nabla G_f(x^t) - \tilde{k}^t \tilde{\nabla} G_f(x^t)\| &= \|k^t \nabla G_f(x^t) - \tilde{k}^t \nabla G_f(x^t) + \tilde{k}^t \nabla G_f(x^t) - \tilde{k}^t \tilde{\nabla} G_f(x^t)\| \\
&\leq \|k^t \nabla G_f(x^t) - \tilde{k}^t \nabla G_f(x^t)\| + \|\tilde{k}^t \nabla G_f(x^t) - \tilde{k}^t \tilde{\nabla} G_f(x^t)\| \\
&\leq |k^t - \tilde{k}^t| \|\nabla G_f(x^t)\| + |\tilde{k}^t| \|\nabla G_f(x^t) - \tilde{\nabla} G_f(x^t)\| \\
&\leq C_f \alpha^2 \|\nabla G_f(x^t)\| + \Big(2 + \frac{13}{12\gamma}\Big) \|\nabla G_f(x^t) - \tilde{\nabla} G_f(x^t)\|, \\
&\leq C_f^2 \alpha^2 \|\nabla f(x^t)\| + \Big(2 + \frac{13}{12\gamma}\Big) \delta \|\nabla f(x^t)\| \\
&\leq C_f^2 \alpha^2 \|\nabla f(x^t)\| + \frac{(2\gamma + \frac{13}{12})\alpha^3}{13} \|\nabla f(x^t)\| \\
&\leq 2C_f^2 \alpha^2 \|\nabla f(x^t)\|.
\end{aligned}
\tag{45}
$$

The fourth line comes from eq. (42) and eq. (43). The fifth line comes from eq. (39). The sixth line comes from $\delta = \frac{\gamma\alpha^3}{13}$.

In the case of one of ideal settings, we need $\alpha$ to satisfy eq. (37). However, we only have the estimation $\tilde{\nabla} G_f(x^t)$. Next, we show that eq. (37) is satisfied if $\alpha$ is small enough. Then we can make sure the linear convergence of the ideal case and further bound the difference of $f - G_f$ between the ideal case and the practical case.

$$
\begin{aligned}
&\Big(\frac{\langle \nabla f(x^t), \nabla G_f(x^t)\rangle}{\|\nabla G_f(x^t)\|^2} - 1\Big)^2 \\
=&\Big(\frac{\langle \nabla f(x^t), \tilde{\nabla} G_f(x^t)\rangle}{\|\tilde{\nabla} G_f(x^t)\|^2} - 1 + \frac{\langle \nabla f(x^t), \nabla G_f(x^t)\rangle}{\|\nabla G_f(x^t)\|^2} - \frac{\langle \nabla f(x^t), \tilde{\nabla} G_f(x^t)\rangle}{\|\tilde{\nabla} G_f(x^t)\|^2}\Big)^2 \\
\geq&\Big(\frac{\langle \nabla f(x^t), \tilde{\nabla} G_f(x^t)\rangle}{\|\tilde{\nabla} G_f(x^t)\|^2} - 1\Big)^2 \\
&- 2\Big|\frac{\langle \nabla f(x^t), \tilde{\nabla} G_f(x^t)\rangle}{\|\tilde{\nabla} G_f(x^t)\|^2} - 1\Big| \cdot \Big|\frac{\langle \nabla f(x^t), \nabla G_f(x^t)\rangle}{\|\nabla G_f(x^t)\|^2} - \frac{\langle \nabla f(x^t), \tilde{\nabla} G_f(x^t)\rangle}{\|\tilde{\nabla} G_f(x^t)\|^2}\Big| \\
\geq&\Big(\frac{\langle \nabla f(x^t), \tilde{\nabla} G_f(x^t)\rangle}{\|\tilde{\nabla} G_f(x^t)\|^2} - 1\Big)^2 - 2C_f \alpha^2 \Big|\frac{\langle \nabla f(x^t), \tilde{\nabla} G_f(x^t)\rangle}{\|\tilde{\nabla} G_f(x^t)\|^2} - 1\Big| \\
\geq&\Big(\frac{\langle \nabla f(x^t), \tilde{\nabla} G_f(x^t)\rangle}{\|\tilde{\nabla} G_f(x^t)\|^2} - 1\Big)^2 - 2C_f \alpha^2 \Big(\frac{\|\nabla f(x^t)\|}{\|\tilde{\nabla} G_f(x^t)\|} + 1\Big) \\
\geq&\Big(\frac{\langle \nabla f(x^t), \tilde{\nabla} G_f(x^t)\rangle}{\|\tilde{\nabla} G_f(x^t)\|^2} - 1\Big)^2 - 2C_f \alpha^2 \Big(\frac{13}{12\gamma} + 1\Big) \geq C - 2C_f \alpha^2 \Big(\frac{13}{12\gamma} + 1\Big) \geq \frac{C}{2}.
\end{aligned}
$$

In the fifth line, we apply eq. (42). In the sixth line, we apply $\|\tilde{\nabla} G_f(x^t)\| \geq \big(\gamma - \gamma\frac{\alpha^3}{13}\big)\|\nabla f(x^t)\|$. In the last line, we apply $\alpha^2 \leq \frac{3C\gamma}{(13+12\gamma)C_f}$. As a result, we obtain

$$
\begin{aligned}
\Big(\frac{\|\nabla G_f(x^t)\|^2}{\langle \nabla f(x^t), \nabla G_f(x^t)\rangle} - 1\Big)^2 &= \Big(\frac{\langle \nabla f(x^t), \nabla G_f(x^t)\rangle}{\|\nabla G_f(x^t)\|^2} - 1\Big)^2 \Big(\frac{\|\nabla G_f(x^t)\|^2}{\langle \nabla f(x^t), \nabla G_f(x^t)\rangle}\Big)^2 \\
&\geq \frac{C}{2}\Big(\frac{\|\nabla G_f(x^t)\|}{\|\nabla f(x^t)\|}\Big)^2 \geq \frac{C}{2} \frac{\|\tilde{\nabla} G_f(x^t)\|^2 - 2\|\nabla G_f(x^t) - \tilde{\nabla} G_f(x^t)\|^2}{2\|\nabla f(x^t)\|^2} \\
&\geq \frac{C}{2}\Big(\frac{72\gamma^2}{169} - \delta^2\Big) \geq \frac{C}{2}\Big(\frac{72\gamma^2}{169} - \frac{\gamma^2\alpha^6}{169}\Big) \\
&\geq \frac{71C\gamma^2}{338} \geq 2(L + L')\alpha.
\end{aligned}
$$

In the second line, we applied $\|x\|^2 \geq \frac{1}{2}\|y\|^2 - \|x - y\|^2$, $\forall x, y \in \mathbb{R}^d$. In the third line, we used the fact that $\|\tilde{\nabla} G_f(x^t)\| \geq \frac{\gamma}{2}\|\nabla f(x^t)\|$ and $\|\nabla G_f(x^t) - \tilde{\nabla} G_f(x^t)\| \leq \delta\|\nabla f(x^t)\|$ and applied

$\delta = \frac{\gamma\alpha^3}{13}$. The last line comes from $\alpha \leq \frac{71C\gamma^2}{676(L+L')}$. Since eq. (37) is satisfied, it indicates $h(k^\star) \leq 0$. And we can apply the result from the ideal case. From lemma 3.4 and eq. (40), we have

$$f(x^{t+1}) - G_f(x^{t+1}) - (f(\bar{x}^{t+1}) - G_f(\bar{x}^{t+1}))$$

$$\leq \langle \nabla f(\bar{x}^{t+1}) - \nabla G_f(\bar{x}^{t+1}), x^{t+1} - \bar{x}^{t+1} \rangle + \frac{L+L'}{2}\|x^{t+1} - \bar{x}^{t+1}\|^2$$

$$= \langle \nabla f(\bar{x}^{t+1}) - \nabla G_f(\bar{x}^{t+1}), \alpha(\tilde{k}^t \tilde{\nabla} G_f(x^t) - k^t \nabla G_f(x^t)) \rangle$$

$$\quad + \frac{L+L'}{2}\|\alpha(\tilde{k}^t \tilde{\nabla} G_f(x^t) - k^t \nabla G_f(x^t))\|^2$$

$$= \langle \nabla f(x^t) - \nabla G_f(x^t), \alpha(\tilde{k}^t \tilde{\nabla} G_f(x^t) - k^t \nabla G_f(x^t)) \rangle$$

$$\quad + \langle \nabla f(\bar{x}^{t+1}) - \nabla f(x^t) - \nabla G_f(\bar{x}^{t+1}) + \nabla G_f(x^t), \alpha(\tilde{k}^t \tilde{\nabla} G_f(x^t) - k^t \nabla G_f(x^t)) \rangle$$

$$\quad + \frac{L+L'}{2}\|\alpha(\tilde{k}^t \tilde{\nabla} G_f(x^t) - k^t \nabla G_f(x^t))\|^2.$$

The first term is

$$\langle \nabla f(x^t) - \nabla G_f(x^t), \alpha(\tilde{k}^t \tilde{\nabla} G_f(x^t) - k^t \nabla G_f(x^t)) \rangle$$

$$\leq \alpha \|\nabla f(x^t) - \nabla G_f(x^t)\|\|k^t \nabla G_f(x^t) - \tilde{k}^t \tilde{\nabla} G_f(x^t)\|$$

$$\leq \alpha(\|\nabla f(x^t)\| + \|\nabla G_f(x^t)\|)2C_f^2\alpha^2\|\nabla f(x^t)\|$$

$$\leq 2C_f^2(1 + C_f)\alpha^3\|\nabla f(x^t)\|^2.$$

In the fourth line, we apply the triangle inequality and the eq. (45). The second term is

$$\langle \nabla f(\bar{x}^{t+1}) - \nabla f(x^t) - \nabla G_f(\bar{x}^{t+1}) + \nabla G_f(x^t), \alpha(\tilde{k}^t \tilde{\nabla} G_f(x^t) - k^t \nabla G_f(x^t)) \rangle$$

$$\leq \|\nabla f(\bar{x}^{t+1}) - \nabla f(x^t) - \nabla G_f(\bar{x}^{t+1}) + \nabla G_f(x^t)\|\|\alpha(\tilde{k}^t \tilde{\nabla} G_f(x^t) - k^t \nabla G_f(x^t))\|$$

$$\leq (L+L')\alpha\|\bar{x}^{t+1} - x^t\|\|\tilde{k}^t \tilde{\nabla} G_f(x^t) - k^t \nabla G_f(x^t)\|$$

$$\leq (L+L')\alpha^2\|\nabla f(x^t) + k^t \nabla G_f(x^t)\|\|\tilde{k}^t \tilde{\nabla} G_f(x^t) - k^t \nabla G_f(x^t)\|$$

$$\leq (L+L')\alpha^2(\|\nabla f(x^t)\| + |k^t|\|\nabla G_f(x^t)\|)\|\tilde{k}^t \tilde{\nabla} G_f(x^t) - k^t \nabla G_f(x^t)\|$$

$$\leq (L+L')\alpha^2\left(1 + \left(3 + \frac{13}{12\gamma}\right)C_f\right)\|\nabla f(x^t)\|\|\tilde{k}^t \tilde{\nabla} G_f(x^t) - k^t \nabla G_f(x^t)\|$$

$$\leq 2(L+L')C_f^2\alpha^4\left(1 + \left(3 + \frac{13}{12\gamma}\right)C_f\right)\|\nabla f(x^t)\|^2$$

$$\leq C_f^2\alpha^3\left(1 + \left(3 + \frac{13}{12\gamma}\right)C_f\right)\|\nabla f(x^t)\|^2$$

$$\leq C_f^2\left(\frac{13}{12\gamma} + 4C_f\right)\alpha^3\|\nabla f(x^t)\|^2.$$

In the sixth line, we apply Cauchy-Schwartz inequality. The eighth line comes from eq. (44). The ninth line comes from eq. (45). The third term is

$$\frac{L+L'}{2}\|\alpha(\tilde{k}^t \tilde{\nabla}_{i^t} G_f(x^t) - k^t \nabla_{i^t} G_f(x^t))\|^2$$

$$\leq \frac{L+L'}{2}(2C_f^2\alpha^3\|\nabla f(x^t)\|)^2$$

$$= 2(L+L')C_f^4\alpha^6\|\nabla f(x^t)\|^2 \leq C_f^4\alpha^5\|\nabla f(x^t)\|^2.$$

In the third line, we apply eq. (45). In conclusion,

$$f(x^{t+1}) - G_f(x^{t+1}) - (f(\bar{x}^{t+1}) - G_f(\bar{x}^{t+1}))$$

$$\leq 2C_f^2(1 + C_f)\alpha^3\|\nabla f(x^t)\|^2 + C_f^2\left(\frac{13}{12\gamma} + 4C_f\right)\alpha^3\|\nabla f(x^t)\|^2$$

$$\quad + C_f^4\alpha^5\|\nabla f(x^t)\|^2 \tag{46}$$

$$\leq \left(\left(2 + \frac{13}{12\gamma}\right)C_f^2 + 6C_f^3 + C_f^4\right)\alpha^3\|\nabla f(x^t)\|^2$$

$$\leq \left(9 + \frac{13}{12\gamma}\right)C_f^4\alpha^3\|\nabla f(x^t)\|^2.$$

and

$$f(x^{t+1}) - G_f(x^{t+1})$$

$$= f(\bar{x}^{t+1}) - G_f(\bar{x}^{t+1}) + f(x^{t+1}) - G_f(x^{t+1}) - (f(\bar{x}^{t+1}) - G_f(\bar{x}^{t+1}))$$

$$\leq \left(1 - \frac{n(L+L')\mu\alpha^2}{2}\right)(f(x^t) - G_f(x^t)) + \left(9 + \frac{13}{12\gamma}\right)C_f^4\alpha^3\|\nabla f(x^t)\|^2$$

$$\leq \left(1 - \frac{n(L+L')\mu\alpha^2}{2}\right)(f(x^t) - G_f(x^t)) + \left(18 + \frac{13}{6\gamma}\right)nLC_f^4\alpha^3(f(x^t) - G_f(x^t))$$

$$\leq \left(1 - \frac{n(L+L')\mu\alpha^2}{4}\right)(f(x^t) - G_f(x^t)).$$

In the second line, we apply theorem 3.10 and eq. (46). In the last line we apply $\alpha \leq \frac{3\gamma(L+L')\mu}{(13+108\gamma)LC_f^4}$.

**Case 3:** From eq. (36) and eq. (40) with $k^t = -1$, we know that

$$f(\bar{x}^{t+1}) - G_f(\bar{x}^{t+1}) \leq f(x^t) - G_f(x^t) - \left(\alpha - \frac{L\alpha^2}{2} - \frac{L'\alpha^2}{2}\right)\|\nabla f(x^t) - \nabla G_f(x^t)\|^2$$

$$\leq f(x^t) - G_f(x^t) - \frac{\alpha}{2}\|\nabla f(x^t) - \nabla G_f(x^t)\|^2. \tag{47}$$

The second line comes from $\alpha \leq \frac{1}{L+L'}$. From lemma 3.4, we have

$$f(x^{t+1}) - G_f(x^{t+1}) - (f(\bar{x}^{t+1}) - G_f(\bar{x}^{t+1}))$$

$$\leq \langle \nabla f(\bar{x}^{t+1}) - \nabla G_f(\bar{x}^{t+1}), x^{t+1} - \bar{x}^{t+1} \rangle + \frac{L+L'}{2}\|x^{t+1} - \bar{x}^{t+1}\|^2$$

$$= \langle \nabla f(\bar{x}^{t+1}) - \nabla G_f(\bar{x}^{t+1}), \alpha(\tilde{\nabla}G_f(x^t) - \nabla G_f(x^t)) \rangle$$

$$+ \frac{L+L'}{2}\|\alpha(\tilde{\nabla}G_f(x^t) - \nabla G_f(x^t))\|^2$$

$$= \langle \nabla f(x^t) - \nabla G_f(x^t), \alpha(\tilde{\nabla}G_f(x^t) - \nabla G_f(x^t)) \rangle$$

$$+ \langle \nabla f(\bar{x}^{t+1}) - \nabla f(x^t) - \nabla G_f(\bar{x}^{t+1}) + \nabla G_f(x^t), \alpha(\tilde{\nabla}G_f(x^t) - \nabla G_f(x^t)) \rangle$$

$$+ \frac{L+L'}{2}\|\alpha(\tilde{\nabla}G_f(x^t) - \nabla G_f(x^t))\|^2.$$

The first term is

$$\langle \nabla f(x^t) - \nabla G_f(x^t), \alpha(\tilde{\nabla}G_f(x^t) - \nabla G_f(x^t)) \rangle$$

$$\leq \alpha\|\nabla f(x^t) - \nabla G_f(x^t)\|\|\tilde{\nabla}G_f(x^t) - \nabla G_f(x^t)\|$$

$$\leq \alpha(\|\nabla f(x^t)\| + \|\nabla G_f(x^t)\|)\|\tilde{\nabla}G_f(x^t) - \nabla G_f(x^t)\|$$

$$\leq (1 + C_f)\alpha\|\nabla f(x^t)\|\|\tilde{\nabla}G_f(x^t) - \nabla G_f(x^t)\| \leq \frac{1}{13}\gamma(1 + C_f)\alpha^4\|\nabla f(x^t)\|^2.$$

In the last line, we apply eq. (39) and $\delta = \frac{\gamma\alpha^3}{13}$. The second term is

$$\langle \nabla f(\bar{x}^{t+1}) - \nabla f(x^t) - \nabla G_f(\bar{x}^{t+1}) + \nabla G_f(x^t), \alpha(\tilde{\nabla}G_f(x^t) - \nabla G_f(x^t)) \rangle$$

$$\leq \|\nabla f(\bar{x}^{t+1}) - \nabla f(x^t) - \nabla G_f(\bar{x}^{t+1}) + \nabla G_f(x^t)\|\|\alpha(\tilde{\nabla}G_f(x^t) - \nabla G_f(x^t))\|$$

$$\leq (L+L')\alpha\|\bar{x}^{t+1} - x^t\|\|\tilde{\nabla}G_f(x^t) - \nabla G_f(x^t)\|$$

$$\leq (L+L')\alpha^2\|\nabla f(x^t) - \nabla G_f(x^t)\|\|\tilde{\nabla}G_f(x^t) - \nabla G_f(x^t)\|$$

$$\leq (L+L')\alpha^2(\|\nabla f(x^t)\| + \|\nabla G_f(x^t)\|)\|\tilde{\nabla}G_f(x^t) - \nabla G_f(x^t)\|$$

$$\leq (L+L')(1 + C_f)\alpha^2\|\nabla f(x^t)\|\|\tilde{\nabla}G_f(x^t) - \nabla G_f(x^t)\|$$

$$\leq \frac{1}{13}\gamma(L+L')(1 + C_f)\alpha^5\|\nabla f(x^t)\|^2 \leq \frac{1}{13}\gamma(1 + C_f)\alpha^4\|\nabla f(x^t)\|^2.$$

In the sixth line, we apply Cauchy-Schwartz inequality. In the ninth line, we apply eq. (39) and $\delta = \frac{\gamma \alpha^3}{13}$. The third term is

$$\frac{L + L'}{2} \|\alpha(\tilde{\nabla} G_f(x^t) - \nabla G_f(x^t))\|^2$$

$$\leq \frac{L + L'}{338} \gamma^2 \alpha^8 \|\nabla f(x^t)\|^2 \leq \frac{1}{338} \gamma^2 \alpha^7 \|\nabla f(x^t)\|^2.$$

In the second line, we applied eq. (39) and $\delta = \frac{\gamma \alpha^3}{13}$. Overall, we obtain

$$f(x^{t+1}) - G_f(x^{t+1}) - (f(\bar{x}^{t+1}) - G_f(\bar{x}^{t+1}))$$

$$\leq \frac{2}{13} \gamma(1 + C_f) \alpha^4 \|\nabla f(x^t)\|^2 + \frac{1}{338} \gamma^2 \alpha^7 \|\nabla f(x^t)\|^2 \leq \frac{3}{13} \gamma(1 + C_f) \alpha^4 \|\nabla f(x^t)\|^2.$$

and,

$$f(x^{t+1}) - G_f(x^{t+1})$$

$$= f(\bar{x}^{t+1}) - G_f(\bar{x}^{t+1}) + f(x^{t+1}) - G_f(x^{t+1}) - (f(\bar{x}^{t+1}) - G_f(\bar{x}^{t+1}))$$

$$\leq f(x^t) - G_f(x^t) - \frac{1}{2} \alpha \|\nabla f(x^t) - \nabla G_f(x^t)\|^2 + \frac{3}{13} \gamma(1 + C_f) \alpha^4 \|\nabla f(x^t)\|^2$$

$$\leq f(x^t) - G_f(x^t) - \frac{1}{2} \alpha \|\nabla f(x^t) - \nabla G_f(x^t)\|^2,$$

$$\leq f(x^t) - G_f(x^t) - \frac{\nu}{2} \alpha (f(x^t) - \nabla G_f(x^t))^{\frac{2}{\theta}}$$

In the last two line, we apply eq. (47) and $\alpha \leq (\frac{13}{12(1 + C_f)})^{1/3} \frac{\|\nabla f(x^t) - \nabla G_f(x^t)\|}{\|\nabla f(x^t)\|}$. The result follows directly from Lemma 6 of Fatkhullin et al. (2022). $\qquad \square$

