# OpenReview forum: "Convergence Analysis of Gradient Descent under Coordinate-wise Gradient Dominance"
_ICLR.cc/2025/Conference — Submitted to ICLR 2025_

### Official Review · Reviewer_uFZD · 2024-10-29

**Soundness:** 2
**Presentation:** 3
**Contribution:** 2
**Rating:** 5
**Confidence:** 3

**Summary:**

This work explores the optimization of finding a Nash Equilibrium for nonconvex functions using first-order gradient-based algorithms and their variations, such as block coordinate descent. The authors introduce the n-sided PL condition, an extension of the PL condition. Then, under this condition, they analyze the convergence of various variants of gradient descent algorithms. They provide theoretical proofs of convergence rates for these algorithms and examine conditions under which linear convergence can be guaranteed.

**Strengths:**

The main strengths of this work are:
-The authors extended the PL condition to the n-sided PL condition.
-The authors proposed adapted variants of the gradient descent (GD) and block coordinate descent (BCD) algorithms and demonstrated the convergence to Nash Equilibrium of these methods.
-The authors provided theoretical proofs for the convergence rates of the proposed algorithms and established conditions under which linear convergence can be guaranteed.
-The authors provide some numerical tests to validate their claims.

**Weaknesses:**

The main weaknesses of this work are:
-Dependence on Strong Assumptions: I understand that the authors gave some toy examples and linear NN that satisfy the assumptions however in general these assumptions remain strong and hard to check.
-The empirical validation is focused on simple toy examples and linear NN. The performance of the proposed algorithms on large-scale or more complex problems is not deeply explored, making it uncertain how well the algorithms scale in practice. The authors may consider including some tests on bigger problems to assess the numerical efficiency of the proposed variants, even if the conditions to converge are not necessarily satisfied.
-The paper lacks a detailed computational complexity analysis of the proposed variants (the cost per iteration compared to the classical GD or BGD). The added steps (like approximating best responses in certain cases) introduce computational overhead, especially for high-dimensional data, and it’s unclear how these adaptations perform against standard or simpler gradient-based methods in terms of cost/runtime.
-The adaptive algorithms, particularly with the introduction of conditions for selecting step sizes and updating directions, could be challenging to implement efficiently in practice.

**Questions:**

-See the previous section.
-Lines 251-252 you mentioned "The above result ensures the convergence of BCD to the NE set, but it does not necessarily indicate whether the output converges to a point within the NE set." can you give an example where this is the case? i.e. we have the convergence to the NE set but not the convergence to a point in the set.
-Line 292: can you explain why f(x) = G_f(x) if and only if x is NE.
-Assumption 3.4. you require the condition (5) for a given set of points. But how can one know this set of points beforehand? you mentioned just after that for instance, for f_0 this assumption is valid for some domain, but the iterations of a given algorithm applied to this problem may leave this domain in the middle of the algorithm.
-Can you tell why Ass 3.4 for instance holds for strongly convex functions?
-In thm 3.6 in the linear rate of convergence, since \mu, \alpha and \kappa "do not depend" on "n" this rate can be negative for large "n"?
-Definition 3.8, do you really need the minimum to be zero?

---

> ### Author Response · Authors · 2024-11-19
> **Authors' Response 1/2**
>
> We appreciate your comprehensive review and valuable feedback of our paper. We would like to address each of your questions.
>
> $\textbf{Weakness 1:}$ As pointed out by the reviewer, our linear convergence rate results are established under additional assumptions, i.e., $n$-sided PL condition. It is worth noting that this assumption is weaker conditions than the strong convexity. Specifically, if a function is strongly convex with respect to $x_i$ while keeping the other coordinates fixed, it satisfies the $n$-sided PL condition. This indicates $n$-sided PL condition can be verified straightforwardly, as it requires the positive definiteness of the corresponding diagonal block of the Hessian matrix. Also, we would like to emphasize that achieving linear convergence is a significant result, as it is typically established under stronger assumptions in the literature, such as strong convexity or strong PL condition.
>
> $\textbf{Weakness 2:}$ Please note that this work is mainly theoretical and focuses on establishing the convergence rate for non-convex problems that satisfy the n-sided PL condition. We also tried to empirically verify our theoretical findings using simple but not that small size problems. For example, we compared our algorithms with the state-of-the-art algorithm (RBCD) with functions up to 20 dimensions.
> We expect similar results for higher dimensions, as our algorithms are only a constant factor slower than RBCD since they require a constant factor more gradient computations than RBCD. But our algorithm enjoy theoretical guarantees which is missing in the literature for non-convex objective functions.
>
> $\textbf{Weakness 3:}$ We propose IA-RBCD in Algorithm 2 and A-RBCD in Algorithm 3. Please note that both of them require only a constant factor of more gradient computations than BCD. This is because IA-RBCD needs to compute the gradient of $G_f$ in addition to the BCD.
> The A-RBCD algorithm requires additional gradient computations to approximate $G_f(x)$, denoted by $ T' $, which depends solely on the parameters of the objective function and is independent of the final precision of $ f - G_f $. This means that, in each iteration, A-RBCD incurs only a constant factor of additional gradient computations compared to IA-RBCD, with the factor determined by properties of the objective function such as smoothness and $\mu$. Compared with IA-RBCD, A-RBCD only cost $\mathcal{O}(\log(n))$ times more, where $n$ is the number of blocks. And we should also notice that our algorithm still works even stably if the function is not lower-bounded or the NE is saddle point.
>
> $\textbf{Weakness 4:}$ In fact, implementing the proposed algorithms are quite similar to standard BCD. The main difference is the subdivision into three cases which are quite straightforward to implement. Please note that we have uploaded the code along with the paper as well.
>
> $\textbf{Question 1:}$ Please note that such examples could be found in the literature. For example, please see the introduction of [1] in which the authors mentioned an example built by H. B. Curry in  "The method of steepest descent for non-linear minimization problems''. The example is "Let $G(x, y)=0$ on the unit circle and $G(x, y)>0$ elsewhere. Outside the unit circle let the surface have a spiral gully making infinitely many turns about the circle. The path $C$ will evidently follow the gully and have all points of the circle as limit points.''
>
> $\textbf{Question 2:}$ We would like to thank the reviewer for the constructive comments. It is important to emphasize that the primary goal of this paper is to find the NE, rather than the global minimizer.
> Furthermore, having $f(x^\star)-G_f(x^\star)=0$ is equivalent to have $x^\star$ is a NE. Notice that at NE, no player can improve its payoff by deviating from its current strategy. This implies that if $x^\star$ is a NE, $f(x^\star)=\min_{x_i}f(x_i,x_{-i}^\star)$, $\forall i$ or $f(x^\star)=G_f(x^\star)$. Conversely, if $f(x^\star)-G_f(x^\star)=0$, then $\sum_{i} [f(x^\star)-\min_{x_i}f(x_i,x_{-i}^\star)]=0$. So we have $f(x^\star)-\min_{x_i}f(x_i,x_{-i}^\star)=0$, $\forall i$. Therefore, we can conclude that $x^\star$ is a NE if and only if $f(x^\star)-G_f(x^\star)=0$.
> We will revisit the statement of Theorem 3.3 to provide more clarification in the final version of the paper.
>
> $\textbf{Question 3:}$ Please note that our algorithms do not require prior knowledge of these points. It is important to highlight that the primary reason for incorporating three different cases in both IA-RBCD and A-RBCD is to update the point based on the position of the current iterate. In particular, condition (5) corresponds to one of these three cases.

---

> > ### Author Response · Authors · 2024-11-19
> > **Authors' Response 2/2**
> >
> > $\textbf{Question 4:}$ Assumption 3.4 intuitively states that the gradients of $ f $ and $ G_f $ are well-aligned at the iterates $ \{x^t\} $. While this may appear to be an artificial assumption, it is expected to hold in the vicinity of isolated NE. Proving this rigorously is an interesting research problem and is left for future work.
> > To provide initial evidence supporting this assumption, we demonstrated in Appendix F that if the function is strongly convex and smooth, condition (5) holds on average over all iterations for gradient descent with continuous dynamics.
> >
> > $\textbf{Question 5:}$ Please note that it is impossible to have a negative rate. This can be explained from two perspectives. First, $\kappa$ depends on $n$. For example, consider the function $f(x_1,\cdots,x_n)=x_1^2+x_2^2+\cdots+x_n^2$, then, it is straightforward to see that $G_f(x)=\frac{n-1}{n}f(x)$, which implies $\kappa=\frac{n-1}{n}$.
> > Second, $f-G_f$ is non-negative by the definition, so this contraction term has to be non-negative.
> >
> > $\textbf{Question 6:}$ No, we do not require this assumption. It is included solely for simplicity. Alternatively, it could be assumed that the minimum is constant.
> >
> > $\textbf{Reference:}$
> >
> > [1] Absil, P.-A., Mahony, R., and Andrews, B. (2005). Convergence of the iterates of descent methods for analytic cost functions. SIAM Journal on Optimization, 16(2):531–547.

---

> > ### Comment · Reviewer_uFZD · 2024-11-24
> >
> > **Q-3.** I know that the algorithms do not require knowledge of the points. However, this assumption about the iterates is necessary for the theoretical guarantees. My question is: how can this type of assumption be ensured?
> >
> > **Q-4.** Demonstrating the assumption on the average does not imply that the assumption holds for every iteration. Can you clarify if this holds at least under strongly convex conditions near the Nash equilibrium points?
> >
> > **Q-5.** Can you say why all the time 1- n \mu \alpha (1-\kappa)/2 > 0. $f- G_f$ is non-negative by the definition is not a proof that this quantity can not be negative.
> >
> > **Q-6.** how the results might change in this case?

---

> ### Author Response · Authors · 2024-11-26
> **Authors' Response**
>
> Thank you for your comments.
>
> $\textbf{Q-3 and Q-4:}$ As noted in the paper and highlighted by the reviewer, we neither provide a theoretical guarantee for when the first case (i.e., when Assumption 3.5 holds) of the presented algorithms is satisfied, nor have we been able to disprove it by identifying a counterexample under the strongly convex condition near the NE points.
>
> However, we can argue the following points to support that Assumption 3.5 is an appropriate tool for analyzing the BCD algorithm. Furthermore, as $\gamma \to 1$ and $C\to 0$,  it becomes increasingly likely to eliminate case 3.
>
> To support this statement, we present a new theorem in Appendix F, which demonstrates that the measure of the set of points not satisfying case 1 or case 2 converges to zero as $\gamma\to 1$ and $C\to 0$.
> Moreover, the maximum of $f-G_f$ over this set of points also converges to 0.
> This indicates that even if we enter case 3 and experience a sublinear convergence rate, we can still ensure that $f-G_f$ is small at these points.
>
> In addition, even if the third case of the algorithms (e.g., when assumption 3.5 is violated) occurs at most $c_0 T$, where $c_0<1$, then we can still guarantee the linear convergence of $f-G_f$ using Theorems 3.10 and 3.11. Please note that $c_0 T$ is not negligible.
>
> Last but not least, we empirically verify how often the third case occurs (Fig. 4 (d) ) and show that it happens quite rarely, which again verifies our arguments.
>
> $\textbf{Q-5:}$ Please note that
> \begin{equation}
> f(x)-G_f(x)=\frac{1}{n}\sum_{i=1}^n\big(f(x)-f(x^\star_i(x),x_{-i})\big),
> \end{equation}
> and for every $i$, $f(x)-f(x^\star_i(x),x_{-i})\geq0$ because $x^\star_i(x)$ is the best response to $x_{-i}$.
> This already shows that the left hand side of the inequality in Theorem 3.7, i.e.,
> \begin{equation*}
>  f(x^{t+1})-G_f(x^{t+1})\leq \Big(1-\frac{n\mu\alpha(1-\kappa)}{2}\Big)(f(x^t)-G_f(x^t)).
> \end{equation*}
> is positive and therefore $(1-\frac{n\mu \alpha(1-\kappa)}{2})$ is also nonnegative.
> To see why the above inequality holds, please see the proof of Theorem 3.7 which uses the smoothness and n-sided PL condition of the function $f$.
>
> $\textbf{Q-6:}$ Please note that in this case, the function $f$ satisfies the $(\theta,\nu)$-PŁ condition iff there exists $\theta \in[1,2)$ and $\nu>0$ such that $\|\nabla f(x)\|^\theta \geq(2 \nu)^{\theta / 2} (f(x)-\min_x f(x)).$ This doesn't change the result of Theorem 3.10 and Theorem 3.11.

---

### Official Review · Reviewer_EUjj · 2024-11-02

**Soundness:** 2
**Presentation:** 3
**Contribution:** 2
**Rating:** 5
**Confidence:** 4

**Summary:**

This paper focuses on the convergence rates of block coordinate descent algorithms under an $n$-sided PŁ condition. Based on an adaptive, theoretical algorithm IA-RBCD that provably converges to the NE of a minimization problem of an $n$-sided PŁ objective, the paper proposes a novel algorithm A-RBCD that is implementable via computing the approximate best response and shows the same convergence speed as the ideal version. The paper also shows empirical results in various practical settings involving $n$-sided PŁ functions.

**Strengths:**

- I believe that extending ideas from convex/nonconvex or minimax optimization to multi-player games is a valuable direction to study and develop further. The $n$-PŁ condition might be useful in many cases, especially in handling stuff on nonconvex (and nonconcave) games.
- Experiments are well-aligned with the theorems. The results all demonstrate and support exactly what’s in the theoretical statements.
- The overall writing is clear and easy to follow.

**Weaknesses:**

- **W1.** I don’t get why and when could finding an NE for a *minimization* problem be an important thing. In Figure 1, the set of stationary points (which is equal to the set of Nashes for $n$-PŁ functions) might contain saddle points. Indeed we would want to find Nashes for minimax or multi-player game settings, but for (nonconvex) minimization problems it’s usually also important to *avoid* possible saddles among stationary points (Lee et al., 2016). I get that there might be cases when we won’t care about saddles and just want to find stationary points, maybe as in the example function with only strict saddles points in Section 4, but still, I think it’s not clear in the paper about why this search for NE’s and the $n$-PŁ assumption could be a really necessary or interesting thing.
- **W2.** In the theorems, everything related to Case 3 makes the theoretical contribution of this paper quite weak. While empirical results in Figure 5 suggest that Case 3 never really happens here, there should have been a better explanation than just saying ‘rigorously verifying these cases is intractable’. There are no lower bound results, i.e., which means that there could be pathological cases that get stuck in Case 3 for a long time, and the $n$-PŁ assumption isn’t powerful enough. In fact, we already need a stronger $(\nu, \theta)$-PŁ assumption to get non-asymptotic results for Case 3, which is even sublinear.
- **W3.** Continuing from **W2**, the main results all state no more than case-specific one-step contraction inequalities. These should have been conclusively rewritten in terms of iteration complexities for a complete understanding of the convergence rate of this algorithm. This will include quantifying how rare Case 3 happens (or maybe when exactly can we make it rare), and seeing what convergence rate we get after fixing a single step size (or possibly a schedule) considering all three cases.

**Questions:**

- Can you explain further about what I wrote in W1? When could it be important to find Nashes in minimization problems? Or, in a slightly different perspective, could these algorithms generalize to multi-player games with adversarial components?
- Have you checked if the proposed algorithms are capable of further avoiding saddles if possible either explicitly, or in some sort of an ‘implicit-bias’ sense? We don’t have any evidence proving/disproving this since, in the experiments in the paper, for the first quadratic case the only option is a saddle, while for the rest of the cases, we can only see $f(x) - G_f(x)$ from which we cannot distinguish saddles from minima.
- Case 3 happens when the vectors $\nabla f(x_t)$ and $\nabla G_f(x_t)$ are not well-aligned (or $\| \nabla G_f(x_t) \|$ is much larger than $\| \nabla f(x_t) \|$… which probably won’t happen). While by definition of $G$ it seems intuitive that the two gradients will eventually align together near the stationary points (while both converging to zero vectors), are there any ideas to quantify or theoretically analyze this part a bit further?

TYPO. Page 6, converges sublinearly for $f_1$ (instead of $f_2$)

TYPO. Algorithm 4, $x_{-j}^t \rightarrow x_{-j}$ (while we use $x = x^t$ in Algorithm 3)

---

> ### Author Response · Authors · 2024-11-19
> **Authors' Response**
>
> We thank for your reviews and valuable insights, which could be much helpful for improving and clarifying our paper. We would like to address your each concern in detail.
>
> $\textbf{Weakness 1 and Question 1:}$
> Finding Nash Equilibrium is important in the potential game literature[1], for example, in reinforcement learning[2] and network theory[3]. In such games, agents often aim at finding an NE using partial information about the full gradient vector of the potential function. To achieve such NE for the agents, we introduced the $n$-sided PL condition. This condition guarantees the existence of a solution for every agent's subproblem, i.e., $\min_{x_i} f(x_i,x_{-i})$, and also an efficient algorithm for finding the NE when only partial gradient is available. Secondly, it enables us to solve a broader set of problems as we do not need the objective function to be lower-bounded, unlike the previous methods. Please note that a function could satisfy the $n$-sided PL condition but not lower-bounded. Moreover, the process of finding an NE is not always guaranteed to be stable[4]. For instance, strategies may diverge when agents simply apply gradient descent to the potential function.
>
>
> It is noteworthy that games with adversarial players could also be included in our problem setting. More precisely, consider an objective function $f$ that depends on $x_1,\cdots,x_n$ and $z$, where $x_i$ is the coordinate corresponding to the $i$-th player and $z$ is the coordinate of an adversarial agent. The players' goal is to minimize the objective function while the adversarial agent aims at maximizing it. Thus, we have
>
> \begin{equation*}
> \begin{aligned}
>     &\min g(x_1,\cdots,x_n)=f(x_1,\cdots,x_n,z^\star(x_1,\cdots,x_n)),\\
>     &\mathrm{s.t.}\ z^\star(x_1,\cdots,x_n)\in\mathrm{argmax}_z f(x_1,\cdots,x_n,z).
> \end{aligned}
> \end{equation*}
>
> Note that the above problem is a minimization problem for the objective function $g$ and it is in the scope of our paper.
>
> $\textbf{Weakness 2:}$ Please note that the GD dynamic (iterates $\{x^t\}$ generated by the algorithm) of a smooth function could potentially visit case 3.
> However, by choosing hyper-parameters (in Algorithms 2 and 3) $\gamma$ close to one and $C$ close to zero, we can reduce the area for which case 3 occurs. Thus, we can reduce the number of times that the GD dynamic visit the third case.
> In this work, we did not characterize functions for which the GD dynamic never visit the third case as it is quite challenging and it is left for future investigations.
> But, we showed empirically (Fig. 5 (d) ) that such visits to case 3 is quite rare during the updates. Moreover, based on our analysis in Appendix F, the dynamic of GD remains outside or on the boundary of the area of case 3 in average by choosing $\gamma$ close to one.
>
> $\textbf{Weakness 3:}$ We thank the reviewer for the helpful suggestion. We avoid to preset the precise contraction inequality as a function of number of total rounds only for simplicity and the fact that we do not have any bound on how often the algorithm visits case 3.
> However, it is quite straightforward to obtain such inequality by assuming that the algorithm visits case 3 at most $T_3$ rounds out of total $T$ rounds. In this case, for $\alpha$ small enough, we have
> $$
> \mathbb{E}[f(x^{T})-G_f(x^{T})]\leq\Big(1-\frac{(L+L')\mu\alpha^2}{4}\Big)^{T-T_3}(f(x^0)-G_f(x^0)).
> $$
>
>
> $\textbf{Question 2:}$
> We would like to thank the reviewer for the valuable question. To this question, $\textbf{yes}$. Please note that it is possible that the conditions for case 1 and case 2 are both satisfied. If the case 1 of the algorithm occurs during the updates (the algorithm checks the first case before moving to the other cases), then we only use the partial gradient of $f$ and no information of the gradient of $G_f$. On the other hand, from Lee et al. (2016), we know that if only the information of the gradient of $f$ is used to find a stationary point, then the iterates almost surely escape the saddle point when the initialization is random.
>
> $\textbf{Question 3:}$
> Understanding when and how often case 3 occurs is an interesting but also challenging problem. In Appendix F, we further studied this case and provide some related results for the gradient dynamics. In short, we could show that for a smooth function, when the iterates are close enough to an isolated NE, $\langle \nabla f(x^t),\nabla G(x^t)\rangle\leq\|\nabla f(x^t)\|^2$ holds in average. Furthermore, when the function is strongly convex, then there exists $0<\gamma<1$ such that $\langle \nabla f(x^t),\nabla G(x^t)\rangle\leq\gamma\|\nabla f(x^t)\|^2$ holds in average.

---

> > ### Author Response · Authors · 2024-11-19
> > **References**
> >
> > [1] Monderer, D. and Shapley, L. S. (1996). Potential games. Games and economic behavior, 14(1):124–143.
> >
> > [2] Zhang, R., Mei, J., Dai, B., Schuurmans, D., & Li, N. (2022). On the global convergence rates of decentralized softmax gradient play in markov potential games. Neurips 2022.
> >
> > [3] Wang, B., Wu, Y., & Liu, K. R. (2010). Game theory for cognitive radio networks: An overview. Computer networks, 54(14), 2537-2561.
> >
> > [4] Carmona, G. (2013). Existence and stability of Nash equilibrium. World scientific.

---

> > > ### Comment · Reviewer_EUjj · 2024-11-24
> > >
> > > I thank the authors for the detailed response, and sorry for the late reply.
> > >
> > > I still think that the empirical observations that 'Case 3 is rare' need a bit more theoretical investigation for the completeness of the work, especially as the focus is on the theoretical parts.
> > >
> > > I will keep my score for now.

---

> ### Author Response · Authors · 2024-11-26
> **Authors' Response**
>
> Thank you for your comment.
>
> For your consideration, in Appendix F, we present our new theorem that as $\gamma\to 1$ and $C\to 0$, then the set of points doesn't satisfy case 1 and case 2 has its measure converge to 0. Also, the maximum of $f-G_f$ over this set converges to 0. This indicates the area that satisfies case 3 can be very small. What's more, even if we enter case 3 and have a sublinear convergence rate, we can still make sure $f-G_f$ is small enough at that time.

---

### Official Review · Reviewer_mncc · 2024-11-04

**Soundness:** 3
**Presentation:** 4
**Contribution:** 3
**Rating:** 6
**Confidence:** 2

**Summary:**

The paper focuses on the problem of finding Nash Equilibrium for function $f(x)$ having a block form $f(x)=f(x_1, …, x_n)$. The paper introduces $n$-sided PL condition which extends the notion of PL condition to functions with block structure. Under this weaker assumption the algorithm Block Coordinate Descent is analyzed and its convergence to set of NE points is proven. Under additional assumption on the alignment of gradients of $f$ and $G_f$, linear convergence of BCD and vanilla GD is proven. Furthermore, the algorithms Adaptive randomized Block Coordinate Descent and its oracle version are proposed and their convergence rate is analyzed without the additional assumption on gradient alignment.

**Strengths:**

Overall, the paper is well written and uses simple examples to provide intuition for important concepts. The strengths of the paper are as follows:
1. The assumption $n$-sided PL condition introduced in the paper is a weaker notion than two-sided PL condition making the results more general. The assumption is weaker in the sense that under this the stationary points are not guaranteed to be global minimums.
2. Linear convergence guarantee is proven for BCD and GD under $n$-sided PL condition and gradient alignment assumption.

**Weaknesses:**

1. Insufficient empirical evidence presented for demonstrating the benefits of the proposed variant: a. Comparison with BCD method is missing for the experiments with Linear Residual Network and Infinite Horizon $n$-player LQR game. It gives empirical comparison of convergence rate of BCD and A-RBCD which would be helpful. b. The variant A-RBCD requires significantly more gradient computation than BCD. It would be beneficial to have comparison keeping the gradient computation budget constant across different methods. c. Furthermore, adding SGD and Adam as baselines would make the experiments more persuasive.

**Questions:**

1. It would be helpful to highlight the differences between IA-RBCD and A-RBCD that arise due to using approximate best responses.
2. The Figure 5d is used to claim that the third case does not occur while executing the A-RBCD algorithm. I suppose the point being made is that $\rho\leq \gamma-\gamma\frac{\alpha^3}{13}$. A clarification regarding the same would be appreciated. If the graph could indicate the line $y=\gamma-\gamma\frac{\alpha^3}{13}$ that would make it much easier to understand.

---

> ### Author Response · Authors · 2024-11-19
> **Authors' Response**
>
> We appreciate your comprehensive review and valuable feedback of our paper. We would like to address each of your questions.
>
> $\textbf{Weakness 1.a:}$  We thank the reviewer for the valuable suggestion and we have the comparision between these methods in the new version. Indeed, this work is primarily theoretical, focusing on providing convergence guarantees and analyzing the convergence rates of various block coordinate descent (BCD) algorithms under the n-sided PL condition. This contribution represents a novel addition to the field.
> In our experiments, we observed that our algorithms exhibit comparable convergence rates to state-of-the-art approaches in LRN and n-player LQR games. However, it is important to highlight a key distinction: unlike the state-of-the-art methods for these types of games, our BCD-based approaches can find NE even when the objective function is not lower bounded or agents do not share their strategies with one another. Specifically, our methods allow agents to individually update their own coordinates using partial gradients, rather than relying on the full gradient vector.
>
> $\textbf{Weakness 1.b:}$ We propose A-RBCD in Algorithm 3 and IA-RBCD in Algorithm 2. Please note that both of them require only a constant factor of more gradient computations than BCD, and are independent of the final precision of $f-G_f$. This is because IA-RBCD needs to compute the gradient of $G_f$ in addition to the BCD.
> The A-RBCD algorithm requires additional gradient computations to approximate $G_f(x)$, denoted by $ T' $, which depends solely on the parameters of the objective function and is independent of the final precision of $ f - G_f $. This means that, in each iteration, A-RBCD incurs only a constant factor of additional gradient computations compared to IA-RBCD, with the factor determined by properties of the objective function such as smoothness and the number of blocks $n$. Compared with IA-RBCD, A-RBCD only cost $\mathcal{O}(\log(n))$times more, where $n$ is the number of blocks. We should also notice that our algorithm still works even if the function is not lower-bounded or the Nash Equilibrium is a saddle point from the example in the Figure 3.
>
> $\textbf{Weakness 1.c:}$ In line with such comparisons to baseline methods, we selected a more relevant algorithm: random block coordinate descent, which is an SGD-based method. Please note that in expectation, random block coordinate descent behaves similarly to SGD. We then compared our algorithm against this baseline to provide a meaningful evaluation.
>
> $\textbf{Question 1:}$ We thank the reviewer for the valuable suggestion. We have added couple of sentences after algorithm 3 to highlight the differences between IA-RBCD and A-RBCD.
>
> $\textbf{Question 2:}$ We thank the reviewer for the valuable suggestion. We have added the line $y=\gamma-\gamma\frac{\alpha^3}{13}$ to the figure for more clarification.

---

> > ### Comment · Reviewer_mncc · 2024-11-25
> >
> > I thank the authors for the elaborate response.
> > As mentioned in author’s response to another reviewer if we assume that the algorithm visits case 3 at most $T_3$ rounds out of total $T$ rounds then for $\alpha$ small enough, we get $$ \mathbb{E}[f(x^{T})-G_f(x^{T})]\leq\Big(1-\frac{(L+L')\mu\alpha^2}{4}\Big)^{T-T_3}(f(x^0)-G_f(x^0)). $$ However no upper bound has been provided for $T_3$ which makes the guarantee provided weak.

---

> ### Author Response · Authors · 2024-11-26
> **Author's response**
>
> Thank you for your comments.
>
> However, we can argue the following points to support that Assumption 3.5 is an appropriate tool for analyzing the BCD algorithm. Furthermore, as $\gamma \to 1$ and $C\to 0$,  it becomes increasingly likely to eliminate case 3.
>
> To support this statement, we present a new theorem in Appendix F, which demonstrates that the measure of the set of points not satisfying case 1 or case 2 converges to zero as $\gamma\to 1$ and $C\to 0$.
> Moreover, the maximum of $f-G_f$ over this set of points also converges to 0.
> This indicates that even if we enter case 3 and experience a sublinear convergence rate, we can still ensure that $f-G_f$ is small at these points.
>
> In addition, even if the third case of the algorithms (e.g., when assumption 3.5 is violated) occurs at most $c_0 T$, where $c_0<1$, then we can still guarantee the linear convergence of $f-G_f$ using Theorems 3.10 and 3.11. Please note that $c_0 T$ is not negligible.
>
> Last but not least, we empirically verify how often the third case occurs (Fig. 4 (d) ) and show that it happens quite rarely, which again verifies our arguments.

---

### Official Review · Reviewer_Cw3t · 2024-11-06

**Soundness:** 3
**Presentation:** 3
**Contribution:** 2
**Rating:** 5
**Confidence:** 4

**Summary:**

The paper focuses on solving non-convex problems and, in particular, aims to design methods for finding Nash Equilibrium. The paper proposes the n-sided PL condition and provides convergence guarantees for different block coordinate descent methods (BCD, IA-RBCS,  and A-RBCD) under this new assumption. Applications of where the assumption is satisfied and some preliminary experiments are presented.

**Strengths:**

The paper is well-written, and the idea is easy to follow. The authors did a very nice job in the narrative of the paper. The explanation of their new assumption, applications of where this assumption appears, and the theoretical analysis are clearly presented.

**Weaknesses:**

I believe there are a few issues in terms of using specific terminology that are not standard, and the author might need to revise.

1. The notion of Nash Equilibrium (NE) is very popular in min-max problems and multi-agent games. However, the paper focuses on pure minimization problems and defines NE as a standard concept, which is not typically the case. How is their NE equilibrium related to the minimizer of the function they are trying to optimize? This i believe was not clearly presented in the paper.

2. I believe the notion G_f(x) should be more carefully presented and explain why this is a valid measure of convergence. Why is showing that f(x^t) - G_f(x^t) reduces enough to guarantee convergence to global minima (again, this is related to the notion of NE)? I would appreciate it if the authors provide more details on this.

3. Why does Assumption 3.4 make sense? This looks like a very artificial condition to have just to make the proof work. This shows that one example (in Fig 1) satisfies, but that does not mean that this is a relaxed condition. More explanation is needed

4. The paper is clearly theoretical, but I would appreciate it if the authors compared their proposed method with state-of-the-art approaches for training Linear Residual Networks and Infinite Horizon n-player Linear-quadratic (LQR) Game. In my opinion, this is very important for the paper as then it would convey the message that through the new assumption, we not only provide new convergence guarantees but that via the new analysis and algorithms, we can more efficiently solve these practical problems.

**Questions:**

See Weaknesses above

---

> ### Author Response · Authors · 2024-11-19
> **Authors' Response**
>
> Thank you for taking the time to review this paper. We appreciate the constructive feedback and would like to address the comments below:
>
> $\textbf{Weakness 1}:$
> Finding Nash equilibria (NE) is a central topic in the literature on multi-agent potential games [1] such as multi-agent reinforcement learning[2]. In potential games, each agent seeks to minimize a unified potential function while having limited knowledge of the strategies of other agents. The NE in such games represents a point of strategic stability, where no agent can improve their payoff by unilaterally deviating from their current strategy. It is important to note that a NE does not necessarily coincide with the global minimum of the potential function. Moreover, the process of finding an NE is not always guaranteed to be stable [3]. For instance, strategies may diverge when agents simply apply gradient descent to the potential function.
> As discussed on pages 4 and 5 and illustrated in Figure 1, there can also be multiple NEs. In such cases, it becomes impossible to guarantee that the identified NE corresponds to the global minimum of the potential function.
> We will provide a more detailed explanation of these points in the introduction and Section 2 of the final version.
>
> $\textbf{Weakness 2}:$
> We would like to thank the reviewer for the constructive comments. It is important to emphasize that the primary goal of this paper is to find the NE, rather than the global minimizer.
> Furthermore, having $f(x^\star)-G_f(x^\star)=0$ is equivalent to have $x^\star$ is a NE. Notice that at NE, no player can improve its payoff by deviating from its current strategy. This implies that if $x^\star$ is a NE, $f(x^\star)=\min_{x_i}f(x_i,x_{-i}^\star)$, $\forall i$ or $f(x^\star)=G_f(x^\star)$. Conversely, if $f(x^\star)-G_f(x^\star)=0$, then $\sum_{i} [f(x^\star)-\min_{x_i}f(x_i,x_{-i}^\star)]=0$. So we have $f(x^\star)-\min_{x_i}f(x_i,x_{-i}^\star)=0$, $\forall i$. Therefore, we can conclude that $x^\star$ is a NE if and only if $f(x^\star)-G_f(x^\star)=0$.
> We will revisit the statement of Theorem 3.3 to provide more clarification in the final version of the paper.
>
> $\textbf{Weakness 3}:$
> Although the statement of Assumption 3.4(the assumption 3.5 in the new version) may seem artificial, it essentially asserts that the vectors $ \nabla f(x_t) $ and $ \nabla G_f(x_t) $ are well-aligned. Intuitively, this alignment is expected based on the definition of $ G_f $ in the vicinity of isolated NEs.
> This intuition has been rigorously justified on average in Appendix F and mentioned before Lemma 3.7. Specifically, we demonstrated in Appendix F that around every isolated minimum of smooth functions, this assumption holds on average for all iterates of the GD algorithm. It is important to note, however, that this assumption does not hold globally. This limitation is the primary reason we have further divided our algorithm into three distinct cases.
>
> $\textbf{Weakness 4}:$
> We thank the reviewer for the valuable suggestion. Indeed, this work is primarily theoretical, focusing on providing convergence guarantees and analyzing the convergence rates of various block coordinate descent (BCD) algorithms under the n-sided PL condition. This contribution represents a novel addition to the field.
> In our experiments, we observed that our algorithms exhibit comparable convergence rates to state-of-the-art approaches in LRN and n-player LQR games. However, it is important to highlight a key distinction: unlike the state-of-the-art methods for these types of games, our BCD-based approaches can find NE even when the objective function is not lower bounded or agents do not share their strategies with one another. Specifically, our methods allow agents to individually update their own coordinates using partial gradients, rather than relying on the full gradient vector.
>
> $\textbf{References}$
>
> [1] Monderer, D. and Shapley, L. S. (1996). Potential games. Games and economic behavior, 14(1):124–143.
>
> [2] Leonardos, S., Overman, W., Panageas, I., and Piliouras, G. (2021). Global convergence of multi-agent policy gradient in markov
> potential games. ICLR 2022.
>
> [3] Carmona, G. (2013). Existence and stability of Nash equilibrium. World scientific.

---

> ### Author Response · Authors · 2024-11-27
> **Official Comment by the Authors**
>
> Thank you for your reviews and suggestions on this paper. We wanted to kindly remind you that the deadline for authors to make revisions to the manuscript is approaching soon (Nov. 27 AoE). While the discussion period has been generously extended by the ICLR committee until Dec. 2 AoE, revisions to the manuscript can only be made before Nov. 27 AoE.
>
> We believe your initial feedback has been very beneficial in clarifying and improving upon our paper. We have already made some revisions under the suggestions from your review and the other reviewers alike, if there are any remaining concerns or suggestions that you feel would benefit from additional updates to the manuscript, we would be very grateful to hear from you before Nov. 27 AoE.
>
> After that date, we are happy to patiently continue the discussion until the end of the discussion period to clarify questions and concerns. Thank you so much for your time and effort in reviewing our work, we appreciate it.

---

### Author Response · Authors · 2024-11-20
**General Response 1/2**

We thank the reviewers deeply for their thoughtful and in-depth reviews of our draft. To efficiently address all concerns, we provide a General Response section, which serves as an additional discussion and clarification to points raised by multiple reviewers that overlap in terms of topic. We hope that the additional discussion will improve our understanding of our work further and resolve outstanding questions.

We have also revised the manuscript with relatively minor updates addressing typos, minor adjustments to diagrams, and adjustments to theorems.

$\textbf{Question 1:}$ The importance of $n$-sided PL condition.

This paper focuses on finding NE for a potential function, which is the main topic in multi-agent potential games [1]. In the potential games, the incentive of all agents to change their strategy can be expressed in one global function. It has many applications, such as reinforcement learning [2,3], power control [4], and network theory [5], etc. The existing work, however, has strict assumptions on the potential function, for example, the strong convexity and the PL condition to provide a convergence rate. These conditions, however, can be problematic for games with multiple NEs with different function values [6]. The $n$-sided PL condition, on the one hand, guarantees the existence of a solution for every best-response problem and, on the other hand, accepts the condition when multiple NE exists.

$\textbf{Question 2:}$ The reason $f-G_f$ is a reasonable measure.

The idea of the best response function and its related dynamics exist in the literature [7,8,9]. From the definition of $G_f$, it is straightforward to see $f-G_f\geq 0$. In the new version of our paper, we prove the equivalence of $f(x^\star)-G_f(x^\star)=0$ and $x^\star$ being a NE point without any further assumption. This is helpful for the cases where multiple NE exists since they might have different function values on $f$, but their value on $f-G_f$ is the same.

$\textbf{Question 3:}$ The idea of assumption 3.4 (assumption 3.5 in the new version) and its illustration.

Please note that our algorithms do not solely rely on this assumption, as this assumption is only one of the three cases in our algorithm. This condition, however, explains why gradient descent on $f$ doesn't converge to NE when the NE is a saddle [10]. Moreover, in Appendix F, we further show that, on average, this assumption is satisfied for strongly convex functions.

$\textbf{References:}$

[1] Monderer, D. and Shapley, L. S. (1996). Potential games. Games and economic behavior, 14(1):124–143.

[2] Ding, D., Wei, C. Y., Zhang, K., Jovanovic, M. (2022). Independent policy gradient for large-scale markov potential games: Sharper rates, function approximation, and game-agnostic convergence. In International Conference on Machine Learning (pp. 5166-5220). PMLR.

[3]Fox, R., Mcaleer, S. M., Overman, W., Panageas, I. (2022). Independent natural policy gradient always converges in markov potential games. In International Conference on Artificial Intelligence and Statistics (pp. 4414-4425).

[4] Liu, W., Xu, H., Wang, X., Zhang, S., Hu, T. (2022). Optimal dispatch strategy of virtual power plants using potential game theory. Energy Reports, 8, 1069-1079.

[5] Moragrega, A., Closas, P., Ibars, C. (2015). Potential game for energy-efficient RSS-based positioning in wireless sensor networks. IEEE Journal on Selected Areas in Communications, 33(7), 1394-1406.

[6] Pozo, D., Contreras, J. (2011). Finding multiple nash equilibria in pool-based markets: A stochastic EPEC approach. IEEE Transactions on Power Systems, 26(3), 1744-1752.

[7] Caruso, F., Ceparano, M. C., Morgan, J. (2020). An inverse-adjusted best response algorithm for Nash equilibria. SIAM Journal on Optimization, 30(2), 1638-1663.

[8] Amiet, B., Collevecchio, A., Scarsini, M., Zhong, Z. (2021). Pure Nash equilibria and best-response dynamics in random games. Mathematics of Operations Research, 46(4), 1552-1572.

[9] Laraki, R., Baudin, L. (2022). Smooth Fictitious Play in Stochastic Games with Perturbed Payoffs and Unknown Transitions.  NeurIPS 2022.

[10] Lee, J. D., Simchowitz, M., Jordan, M. I., Recht, B. (2016). Gradient descent converges to minimizers. COLT 2016

---

### Author Response · Authors · 2024-11-26
**General Response 2/2**

Once again, we thank the reviewers for their thoughtful feedback on our rebuttal. Since Assumption 3.5, and particularly the theoretical guarantees regarding the frequency of case 3, were questioned, we would like to provide the following clarification.

$\textbf{In the newly added Theorem F.2}$ in Appendix F, we further provide a bound for the measure of the set of points satisfying case 3 and an upper bound for $f-G_f$ over this set.
In particular, we show that the measure of the set of points not satisfying case 1 or case 2 converges to zero as $\gamma\to 1$ and $C\to 0$. Moreover, the maximum of $f-G_f$ over this set of points also converges to 0.
This indicates that the size of the set of points satisfying case 3 shrinks as $\gamma\to 1$ and $C\to 0$, and even if the algorithms enter case 3 and experience a sublinear convergence rate, we can ensure that $f-G_f$ is small at these points.

In addition, even if the third case of the algorithms (e.g., when assumption 3.5 is violated) occurs at most $c_0 T$, where $c_0<1$, then we can still guarantee the linear convergence of $f-G_f$ using Theorems 3.10 and 3.11. Please note that $c_0 T$ is not negligible.

---

### Author Response · Authors · 2024-12-03
**Official Comment by Authors**

We sincerely thank the reviewers for their efforts and valuable feedback. The reviewers provided excellent constructive comments and insightful discussions.
To highlight our main contributions: this paper introduces an extension of the PL condition in nonconvex optimization. Under this extended notion, we provide theoretical guarantees for convergence toward Nash Equilibrium using the proposed first-order gradient-based algorithms. These algorithms adjust the update rule based on three possible scenarios. We demonstrate linear convergence in the first two scenarios and sublinear convergence in the third one.

As noted by the reviewers, we have added a new theorem to analyze the rarity of case 3 and illustrated the frequency of case 1 for strongly convex functions. We also included a performance comparison in the experimental section. The manuscript has been revised with minor updates, including corrections of typos, improvements to diagrams, adjustments to definitions and lemmas.

Kind regards, Authors

---

### Meta-Review · Area_Chair_Kz39 · 2024-12-20

**Metareview:**

This paper addresses the optimization of finding Nash Equilibria for non-convex functions with a block structure, introducing the novel n-sided PL condition, which generalizes the PL condition to such settings. The authors analyze the convergence of gradient-based algorithms, including Block Coordinate Descent (BCD) and its adaptive variants (IA-RBCD and A-RBCD), proving convergence rates and linear convergence under specific gradient alignment assumptions. They further propose A-RBCD, a practical algorithm that approximates the best response and achieves the same convergence speed as its theoretical counterpart. Empirical results demonstrate the effectiveness of these methods in practical scenarios satisfying the n-sided PL condition.

**Additional Comments On Reviewer Discussion:**

The reviewers have concerns about the paper's clarity, rigor, and overall contribution, categorizing it as borderline. In reviewer discussions, they find the authors' responses to raised questions unconvincing, with key issues remaining unresolved. Concerns include weak asymptotic results that fail to provide meaningful improvements in convergence rates, insufficient justification and clarity around the motivation for finding Nash Equilibria in minimization problems, and a lack of robust theoretical analysis for certain cases (e.g., Case 3) that undermine confidence in the proposed algorithms. They agree that the paper requires substantial refinement and polishing before it can meet the standards of a major machine learning conference.

---

### Decision · Program_Chairs · 2025-01-22

Reject